# Arf1 coordinates fatty acid metabolism and mitochondrial homeostasis

Ludovic Enkler [1], Viktoria Szentgyörgyi[1], Mirjam Pennauer[1], Cristina Prescianotto-Baschong[1], Isabelle Riezman[2], Aneta Wiesyk[3], Reut Ester Avraham[4], Martin Spiess [1], Einat Zalckvar[4], Roza Kucharczyk[3], Howard Riezman [2] & Anne Spang [1] ✉

Lipid mobilization through fatty acid β-oxidation is a central process essential for energy production during nutrient shortage. In yeast, this catabolic process starts in the peroxisome from where β-oxidation products enter mitochondria and fuel the tricarboxylic acid cycle. Little is known about the physical and metabolic cooperation between these organelles. Here we found that expression of fatty acid transporters and of the rate-limiting enzyme involved in β-oxidation is decreased in cells expressing a hyperactive mutant of the small GTPase Arf1, leading to an accumulation of fatty acids in lipid droplets. Consequently, mitochondria became fragmented and ATP synthesis decreased. Genetic and pharmacological depletion of fatty acids phenocopied the *arf1* mutant mitochondrial phenotype. Although β-oxidation occurs in both mitochondria and peroxisomes in mammals, Arf1's role in fatty acid metabolism is conserved. Together, our results indicate that Arf1 integrates metabolism into energy production by regulating fatty acid storage and utilization, and presumably organelle contact sites.

Intracellular compartmentalization of metabolic processes involves deep and well-orchestrated inter-organelle communications to coordinate cellular functions. This requires homeostatic control of lipid, ion and metabolite transfer between organelles, and between organelles and the plasma membrane[1–6]. Exchanges are established through vesicular transport by means of kissing and fusing[1], and membrane contact sites[5,7].

Mitochondria form contacts with almost every organelle in the cell[2,3,8]. They establish functional interactions with peroxisomes[9,10] and with lipid droplets (LDs[11,12]) to ensure fatty acid (FA) metabolism and ATP production. Lipids are stored in LDs in the form of triacylglycerol (TAG) and sterol esters (SE). Under nutrient shortage, FAs are released from LDs by lipolysis and metabolized by β-oxidation solely in peroxisomes in yeast, or in both peroxisomes and mitochondria in mammalian cells. Subsequently, shortened acyl-CoA (or acetylcarnitine/citrate)

is transferred from peroxisomes to mitochondria by an unknown mechanism[9,10], where it will fuel the tricarboxylic acid (TCA) cycle and the respiratory chain (RC) complexes for oxidative phosphorylation (OXPHOS). Hence, LDs stay in close proximity to peroxisomes and mitochondria for efficient transfer of metabolites[13–17]. Perturbed contact sites between these organelles and mitochondria are correlated with metabolic syndromes, liver disease and cancers, highlighting their central role in cellular homeostasis[8,18,19]. Nevertheless, how contact sites are organized and regulated, and which proteins are involved in metabolite transfer allowing proper lipid flux between organelles to ensure effective energy production, remains enigmatic.

Arf1 is a master regulator of vesicle formation at the Golgi[20,21], and its activity is modulated by ArfGAPs (GTPase activating proteins) and ArfGEFs (guanine nucleotide exchange factors) stimulating GTP hydrolysis and GDP-to-GTP exchange, respectively. Over the past years,

[1]Biozentrum, University of Basel, Basel, Switzerland. [2]Department of Biochemistry, NCCR Chemical Biology, University of Geneva, Geneva, Switzerland. [3]Institute of Biochemistry and Biophysics, Polish Academy of Sciences, Warsaw, Poland. [4]Department of Molecular Genetics, Weizmann Institute of Science, Rehovot, Israel. ✉e-mail: anne.spang@unibas.ch

additional functions of Arf1 have been identified. We and others have shown that Arf1 regulates messenger RNA transport[22,23], mTORC1 activity[24,25], and mitochondrial dynamics and transport[26–28]. However, it still remains unclear how Arf1 specifically regulates mitochondrial dynamics. While we and others have observed that eliminating *ARF1* in *Caenorhabditis elegans* or HeLa cells leads to mitochondrial hyperconnectivity[26,28], mitochondria were fragmented and globular in the yeast *arf1-11* mutant[26,29], indicating that Arf1 might play additional roles at mitochondria.

The Arf1/COPI machinery has also been implicated in lipid metabolism by governing lipolysis, LD morphology, protein recruitment, phospholipid removal and the formation of endoplasmic reticulum (ER)–LD bridges[30–36]. Furthermore, Arf1 and COPI could be recruited onto peroxisomes, and Arf1 might be involved in peroxisome proliferation[37–39]. However, Arf1 function on peroxisomes and in FA metabolism remains elusive.

In this Article, we show that Arf1 couples FA β-oxidation to mitochondrial ATP synthesis. We demonstrate that Arf1 activity regulates expression of long-chain FA transporters Pxa1/Pxa2 and of the first and rate-limiting enzyme involved in β-oxidation, Pox1 in yeast. Arf1 modulates FA availability on LDs by promoting TAG synthesis and hydrolysis. Hyperactive Arf1 leads to an increased level of TAGs in LDs, and to reduced lipid transfer to mitochondria. This conserved mechanism is essential to sustain endomembrane homeostasis and mitochondrial ATP synthesis. Moreover, Arf1 activity drives both mitochondrial fusion and fission in yeast, thereby consolidating previous results[26,28]. Thus, Arf1 appears to be required for the regulation of mitochondrial dynamics and for FA metabolism and acetyl-CoA transfer to mitochondria.

## Results

### Arf1 regulates mitochondrial fusion and fission

To better understand the discrepancies between the hyperconnectivity of mitochondria observed in metazoans[26,28] and the globular, fragmented mitochondria in the yeast temperature-sensitive *arf1-11* mutant (Fig. 1a), we measured mitochondrial fission and fusion activity in the *ARF1* and *arf1-11* strains. For clarity, we will use the prefix 'y' for all yeast and 'm' for all mammalian genes and proteins. On the basis of growth curves and cell morphology (Fig. 1b–d), we shifted strains for 30 min to 37 °C before imaging[29].

We could readily detect yArf1–GFP at mitochondrial fission and fusion sites at both 23 °C and 37 °C (Fig. 1e,g,j, Extended Data Fig. 1a,c and Supplementary Videos 1 and 2). Surprisingly, in y*arf1-11* cells, the number of fission and fusion events was higher than in y*ARF1* cells at 23 °C (Fig. 1f,i, Extended Data Fig. 1b and Supplementary Video 3). In contrast, mitochondrial dynamics was greatly reduced at 37 °C (Fig. 1h,i, Extended Data Fig. 1d and Supplementary Video 4), even though only mildly affecting yArf1-11–GFP localization at fusion and fission sites (Fig. 1h,j and Extended Data Fig. 1d). Our data suggest that yArf1 is required for both mitochondrial fusion and fission, reconciling the findings in mammalian cells, *C. elegans* and yeast.

To test whether Arf1 exerts its activity directly on mitochondria, we anchored yArf1, yArf1-11 or the dominant negative version of yArf1T31N (yArf1-DN) on mitochondria via the mitochondrial translocase Tom20 (Fig. 2a,b). Irrespective of which yArf1 variant we anchored, almost no

mitochondrial tubular network could be observed (Fig. 2c). Thus, the constant presence of yArf1 on mitochondria impairs mitochondrial dynamics. Deletion of y*ARF1*, however, also impacted mitochondrial morphology, while mitochondrial dynamics were only mildly affected, presumably due to the presence of Arf2 (Fig. 2d,e and Supplementary Videos 5–8). Interestingly, yArf1-DN but not yArf1-11 or the constitutively active yArf1Q71L (yArf1-CA) had a dominant phenotype on mitochondria morphology (Fig. 2f,g), indicating that active Arf1 is necessary to maintain mitochondria morphology. Taken together, our data suggest that Arf1 cycling between GTP- and GDP-bound states might be important to sustain mitochondria homeostasis.

### yArf1-11 is a hyperactive mutant present on the ER and LDs

Although Arf1 activity affects mitochondrial morphology, we were puzzled by the difference in mitochondrial morphology in yeast, *C. elegans* and mammalian cells. A main difference between the experiments was that in yeast we used a mutant, while Arf1 was eliminated in metazoans[26,28]. Since two of the three mutations of y*arf1-11* are located within or in close proximity to the GTP-binding domain, we asked whether GTP binding was impaired in yArf1-11. Thus, we incubated soluble (S100) and pellet (P100) yArf1-11 lysate fractions with the GAT domain of the Arf effector Gga2 (Gga2$^{GAT}$), which specifically binds yArf1 in its GTP-bound form[40]. yArf1-11 in the P100 fraction was more efficiently retained by Gga2$^{GAT}$ compared with yArf1, regardless of the temperature and whether tagged or untagged Arf1 variants were used (Fig. 3a,b). At 23 °C, yArf1-11 in the S100 fraction also bound Gga2$^{GAT}$ (Fig. 3a,b). Our results indicate that yArf1-11 is mostly in the active conformation already at 23 °C, that GTP binding does not change upon shift to the restrictive temperature, and that hence y*arf1-11* is a gain-of-function mutant. This finally explains the different observations; loss of Arf1 function yields hyperfused mitochondria, while a gain-of-function mutation results in globular mitochondria.

When we recorded the movies on mitochondrial dynamics, we noticed that the yArf1-11 localization pattern was different to that of yArf1. This could either be due to a difference in Golgi morphology in y*arf1-11*, where the bulk of yArf1 is localized, or yArf1-11 might localize to different organelles. While yArf1–GFP mainly localized to the Golgi (Fig. 3c and Extended Data Fig. 2a,c), only a minor fraction of yArf1-11–GFP was present at the Golgi (Extended Data Fig. 2b,d). Most yArf1-11–GFP was in a pattern conspicuously similar to the ER at 23 °C (Fig. 3c), which we confirmed with the ER-marker Sec61–mCherry (Fig. 3d) and by immuno-electron microscopy (EM) (Fig. 3e–h). Since y*arf1-11* cells do not have a growth defect at 23 °C (Fig. 1b), Arf1-11's ER localization does not seem to be detrimental. At 37 °C, however, yArf1-11 was massively relocated to puncta, which did not correspond to Golgi compartments (Fig. 3c and Extended Data Fig. 2b–d). Arf1 has been reported to be localized also to LDs and mitochondria[30,31,33,36,41,42], which we also observed irrespective of growth temperature (Fig. 3e,f). Indeed, these yArf1-11 puncta at 37 °C corresponded to LDs (Fig. 3i and Extended Data Fig. 2e,f). Of note, the number of LDs appeared to be increased and clustered in y*arf1-11* compared with wild type (WT) (Fig. 3g–i and Extended Data Fig. 2f). The presence of yArf1-11 on LDs did not prompt an increase in COPI coat components on LDs (Extended Data Fig. 2g,h), suggesting a COPI-independent function of Arf1-11 on

**Fig. 1 | yArf1 regulates mitochondria fusion and fission. a**, Schematic of the thermo-sensitive mutant Arf1-11 in yeast (Yahara et al.[29]). Amino acid coordinates are indicated in bold below the protein and corresponding mutated amino acids in red. **b**, Growth test of y*ARF1* and y*arf1-11* strains on rich medium (YPD) and incubated at 23 °C, 30 °C and 37 °C. **c**, Cell viability assay of y*ARF1* and y*arf1-11* strains performed after shifting cells to 37 °C. ODs were measured at regular timepoints. Mean and standard deviation are shown; *n* = 3 biological replicates. **d**, y*ARF1* and y*arf1-11* strains phenotypes followed by microscopy after 0, 30, 60 and 120 min incubation time at 37 °C. Scale bar, 5 μm. **e**–**h**, Single timepoint images of movies done with strains expressing yArf1–GFP (**e,g**) or yArf1-11–GFP

(**f,h**) together with the mitochondrial protein Tom70 fused to mCherry at 23 °C (**e,f**) or shifted to 37 °C (**g,h**). White arrows indicate sites of fission and yellow arrows fusion. Asterisk indicates a fusion event independent of Arf1 in **h**. Scale bar, 5 μm. Scale bar inlays, 2.5 μm. **i,j**, Measurements of mitochondrial fusion and fission events per cell (**i**) and the frequency of events where yArf1 is involved (**j**). Mean and standard deviation are shown; y*ARF1* 23 °C = 271 cells, y*ARF1* 37 °C = 231 cells, y*arf1-11* 23 °C = 379 cells and y*arf1-11* 37 °C = 186 cells from *n* = 3 biological replicates. Source numerical data are available in source data. See also Extended Data Fig. 1.

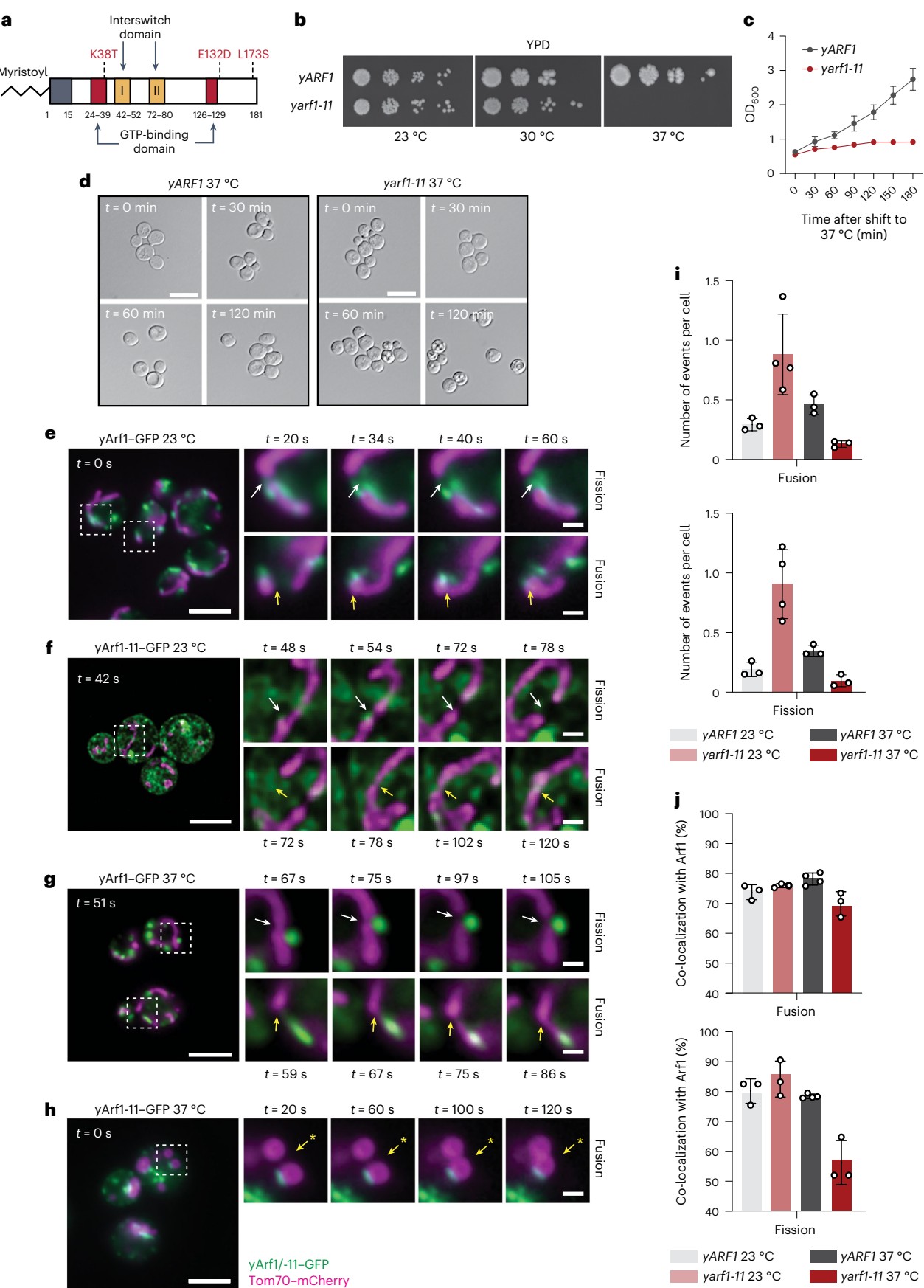

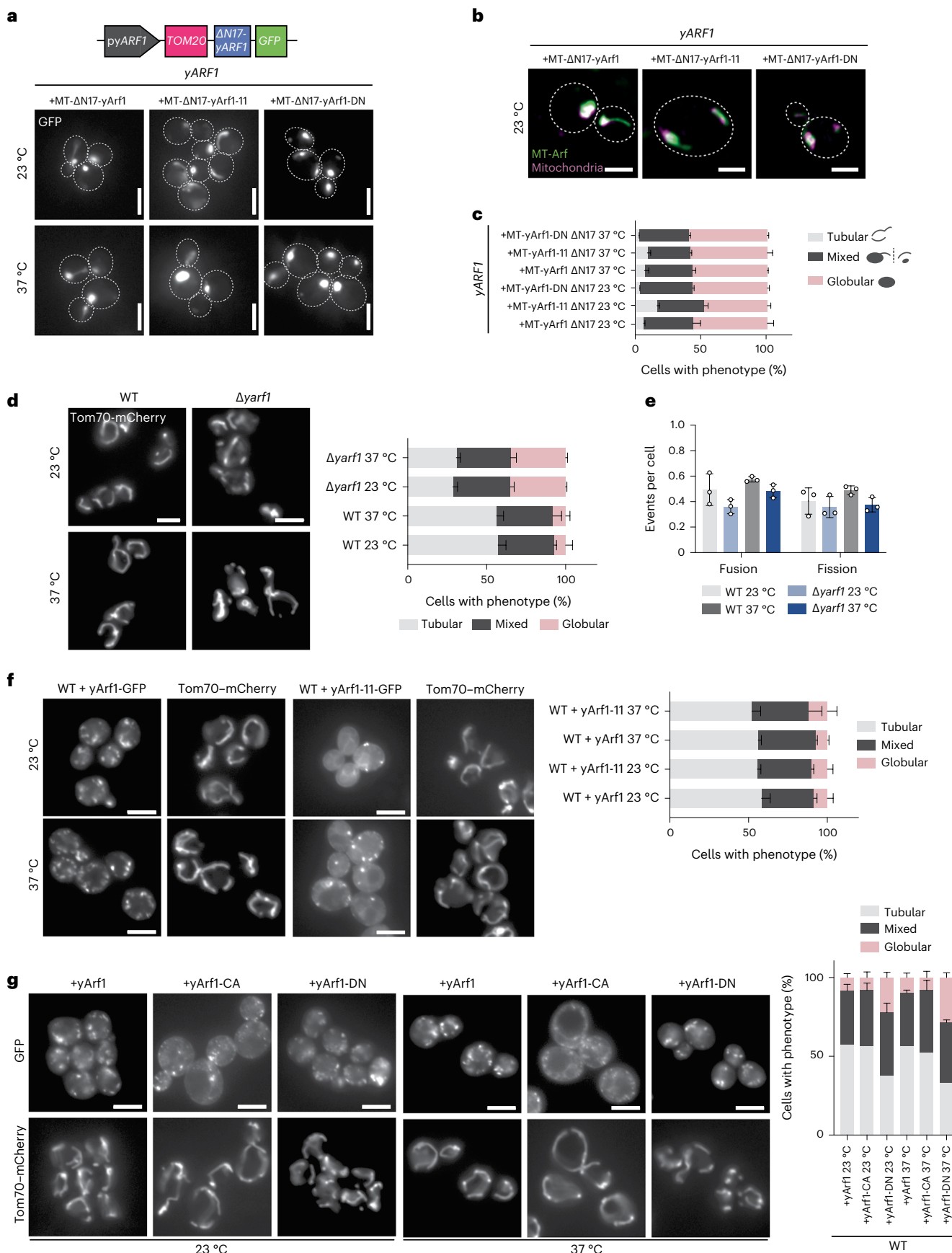

**Fig. 2 | Control of Arf1 activity is needed for mitochondria dynamics.**
**a**, Schematic of the construct designed to anchor yArf1 on mitochondria (MT) via Tom20. y*ARF1* deleted in its myristoylation sequence (ΔN17) was expressed from its endogenous promoter and fused to GFP on its 3′ end. Localization of MT-anchored ΔN17-yArf1–GFP, the dominant negative yArf1-DN or yArf1 bearing yArf1-11–GFP variant in y*ARF1* cells grown at 23 °C or shifted to 37 °C. Scale bar, 5 μm. **b**, High-resolution co-localization of MT-anchored ΔN17-yArf1–GFP, yArf1-DN and Arf1-11–GFP with mitochondria stained with MitoTracker Deep Red FM. A single focal plane of 0.2 μm is shown. Scale bar, 2 μm. **c**, Measurements of mitochondria phenotypes (tubular, mixed or globular) based on images taken in **a**. Mean and standard deviation are shown; Δ*yarf1* + MT-Arf1 23 °C = 419 cells, Δ*yarf1* + MT-Arf1-11 23 °C = 544 cells, Δ*yarf1* + MT-Arf1-DN 23 °C = 606 cells, Δ*yarf1* + MT-Arf1 37 °C = 509 cells, Δ*yarf1* + MT-Arf1-11 37 °C = 453 cells, Δ*yarf1* + MT-Arf1-DN 37 °C = 462 cells from *n* = 3 biological replicates. **d**, Mitochondria morphology were imaged in WT and Δ*yarf1* strains grown at 23 °C or shifted to 37 °C using Tom70–mCherry as mitochondrial marker. Mitochondria phenotypes (tubular, mixed or globular) were measured. Mean and standard deviation are shown; WT 23 °C = 355 cells, WT 37 °C = 582 cells, Δ*yarf1* 23 °C = 419 cells, Δ*yarf1-11* 23 °C = 571 cells from *n* = 3 biological replicates.

Scale bar, 5 μm. **e**, Mitochondrial fusion and fission events were measured on the basis of Supplementary Videos 5–8. Mean and standard deviation are shown; WT 23 °C = 320 cells, WT 37 °C = 279 cells, Δ*yarf1* 23 °C = 319 cells, Δ*yarf1-11* 37 °C = 321 cells from *n* = 3 biological replicates. **f**, Mitochondria morphology were imaged in WT cells expressing yArf1–GFP or yArf1-11–GFP grown at 23 °C or shifted to 37 °C using Tom70–mCherry as mitochondrial marker. For each strain the tubular, mixed and globular phenotypes were measured. Mean and standard deviation are shown. Scale bar, 5 μm. +yArf1 23 °C = 251 cells, +yArf1 37 °C = 273 cells, +yArf1-11 23 °C = 266 cells, +yArf1-11 37 °C = 273 cells from *n* = 3 biological replicates. **g**, Mitochondria morphology were imaged in WT cells expressing yArf1-, the constitutively active mutant yArf1-CA- or the dominant negative yArf1-DN–GFP grown at 23 °C or shifted to 37 °C using Tom70–mCherry as mitochondrial marker. For each strain the tubular, mixed and globular phenotypes were measured. Mean and standard deviation are shown. +yArf1 23 °C = 269 cells, +yArf1 37 °C = 275 cells, +yArf1-CA 23 °C = 268 cells, +yArf1-CA 37 °C = 287 cells, +yArf1-DN 23 °C = 243 cells, +yArf1-DN 37 °C = 411 cells from *n* = 3 biological replicates. Scale bar, 5 μm. Source numerical data are available in source data.

LDs. We conclude that yArf1-11 mainly localizes to the ER at the permissive temperature and to LDs at the restrictive temperature.

### LD-localized yArf1 induces mitochondria fragmentation

Since yArf1-11 is a gain-of-function mutant, we hypothesized that yArf1-CA might localize in a similar fashion. Indeed, yArf1-CA localized also to the ER and in smaller puncta that were quite distinct from the Golgi localization observed with yArf1 and yArf1-DN (Fig. 4a). Therefore, the active form of yArf1 can be found on the ER and most likely also on LDs.

Next, we asked which mutation, or combination of mutations, is responsible for the localization of yArf1-11 to the ER and LDs by re-introducing yArf1-11 mutations in WT yArf1 (Supplementary Tables 1 and 2). The single mutations K38T and L173S as well as the K38T–E132D pair perturbed yArf1 localization, but failed to localize yArf1 to the ER at 23 °C (Fig. 4b). Only the reconstitution of all three mutations (K38T–E132D–L173S) caused yArf1 to be on the ER at 23 °C and on LDs at 37 °C (Fig. 4b), but none of the combinations tested elicited a dominant phenotype (Fig. 4c).

We wondered whether the localization of yArf1-11 on LDs was the cause of mitochondrial fragmentation. Therefore, we anchored yArf1 on LDs by replacing the N-terminal amphipatic helix of Arf1 with the PAT domain of the LD protein Pln1 (ref. 43) in a Δ*arf1* strain. As shown above, loss of yArf1 already impacts mitochondria morphology (Fig. 4d). However, this phenotype was exacerbated irrespective of which yArf1 version was targeted to LDs (Fig. 4D), suggesting that the continuous presence of yArf1 on LDs increases the level of globular mitochondria. To corroborate our findings, we prevented yArf1 to localize on LDs by targeting yArf1 and yArf1-11 to the ER using the transmembrane domain of Sec66 (Fig. 4e). Their sequestration at the ER did not affect growth even in the absence of WT y*ARF1* (Fig. 4f). Under these conditions, the mitochondrial network remained

tubular in cells expressing ER-anchored yArf1-11 even at 37 °C (Fig. 4g,h). Moreover, mitochondria fusion was similar between yArf1 and yArf1-11 irrespective of the temperature (Fig. 4i and Supplementary Videos 9–12). Thus, localization of yArf1-11 on the ER prevents mitochondria fragmentation at 37 °C.

### Functional conservation of Arf1-11 in mammalian cells

Since Arf1 also plays a role in mitochondrial dynamics in mammalian cells, we wondered whether Arf1-11 localization and function would be conserved in mammalian cells. Thus, we mutated mammalian Arf1 at the corresponding positions (mArf1-11), and C-terminally fused WT mammalian Arf1 (mArf1) and mArf1-11 to GFP (Extended Data Fig. 3a). These proteins were expressed in HeLa *ARF1* knockout cells (*ARF1* KO; Extended Data Fig. 3b,c)[44]. mArf1-11 expression in these cells led to drastic cell death after 3 days (Extended Data Fig. 3d). Thus, like in yeast, the hyperactive form of mArf1 has severe effects on cell survival.

As expected, mArf1–GFP was present on the Golgi and on vesicles (Fig. 5a and Extended Data Fig. 3e). In contrast, mArf1-11 only modestly localized to the Golgi (Fig. 5b and Extended Data Fig. 3f). Instead mArf1-11 decorated tubular and large round structures (Extended Data Fig. 3g,h). The tubular structures were positive for the mitochondrial marker TOM20, but not for the ER marker CLIMP63 (Fig. 5c–f). mArf1-positive vesicles were sometimes juxtaposed to mitochondria in agreement with mArf1 function in mitochondria division or transport[26–28]. As expected, mArf1 knockout (KO) cells had reduced levels of mitochondria fission and fusion, which were restored to normal levels upon expression of either mArf1 or mArf1-11 (Extended Data Fig. 3i and Supplementary Videos 13–16), supporting the notion that Arf1's functions on mitochondria are conserved from yeast to mammals. Likewise, the localization of mArf1-11 to LDs was conserved (Fig. 5g,h), encouraging us to determine the function of Arf1-11 on LDs.

**Fig. 3 | yArf1-11 is a hyperactive mutant present on the ER and LDs. a,b**, Active yArf1 pull-down and detection experiments done with strains expressing yArf1 and yArf1-11 fused to GFP (**a**) or endogenous untagged yArf1 and yArf1-11 (**b**). Protein extracts from soluble (S100) or pellet (P100) fractions from y*ARF1* and y*arf1-11* cells grown at 23 °C or shifted to 37 °C were incubated with equal amount of purified GST-tagged GAT domain of Gga2 (Gga2^GAT). Sec61 and Anp1 were used as membrane marker and Pgk1 as cytosolic marker. s.e., short exposure; l.e., long exposure; PD, pull-down. **c**, Localization of WT yArf1 and yArf1-11 C-terminally fused to GFP. Cells were incubated either at 23 °C or shifted at 37 °C for 30 min. Mean and standard deviation are shown. Scale bar 5 μm. **d**, Co-localization of yArf1–GFP and yArf1-11–GFP with the ER marker Sec61 tagged with mCherry grown at 23 °C and 37 °C. Cells highlighted by dotted squares depict GFP

and mCherry co-localization. Fluorescence intensities of each channel were measured on a circle drawn around the perinuclear ER and are shown here as arbitrary units (a.u.). Scale bar, 5 μm. **e–h**, TEM of y*ARF1* (**e,f**) and y*arf1-11* (**g,h**) strains grown either at 23 °C (**e,g**) or shifted to 37 °C (**f,h**) for 30 min. yArf1 and yArf1-11 localizations were detected by immunogold labeling, and dotted squares show enlargements of specific Arf1 localizations. Scale bar, 500 nm. Scale bar magnification, 200 nm. **i**, Co-localization of yArf1–GFP and yArf1-11–GFP with the LD marker Erg6 tagged with mCherry grown at 23 °C or shifted to 37 °C for 30 min. Arrows indicate sites of co-localization between the yArf1/yArf1-11 and LD. Scale bar, 5 μm. Unprocessed blots are available in source data. See also Extended Data Fig. 2.

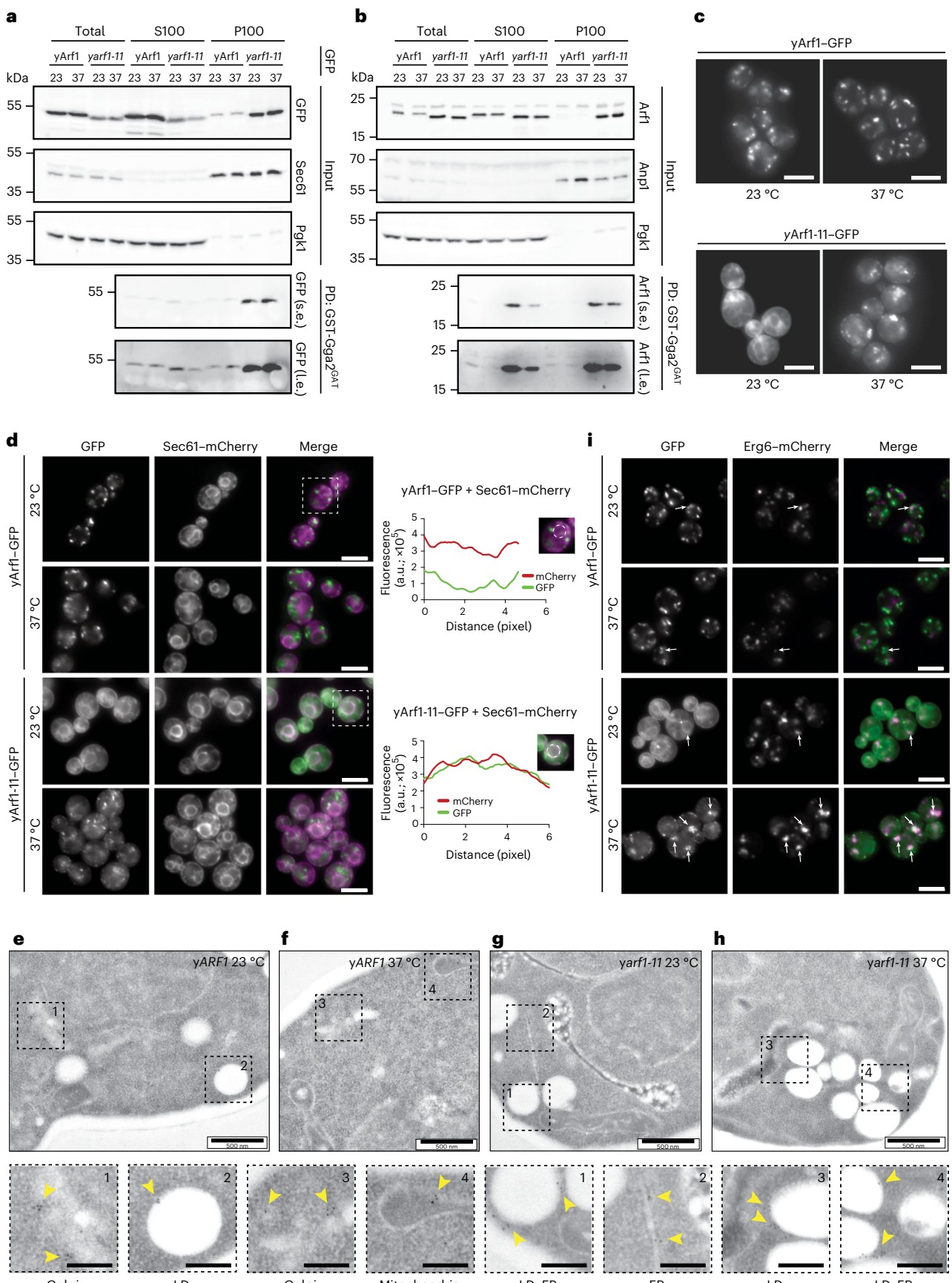

## Hyperactive Arf1 induces TAG accumulation

When we analysed the yeast *arf1-11* mutant phenotype, we noticed in our transmission electron microscopy (TEM) pictures an increase in LD number and dilated ER (Figs. 3e–h and 6a and Extended Data Fig. 4a), which we confirmed by staining LDs with LipidTox in yeast (Fig. 6b) and with Nile Red in mammalian cells (Fig. 6c). LDs are specialized organelles primarily known for their role in energy storage in the form of neutral lipids, mainly TAG and SE. Although lipidomic analysis of y*ARF1* and y*arf1-11* strains did not reveal any differences in the levels of SE (Extended Data Fig. 4b), we found a strong increase in TAG levels in y*arf1-11* compared with y*ARF1* (Fig. 6d). Strikingly, these elevated TAG levels were not matched by changes in the overall phospholipid composition (Extended Data Fig. 4c).

We have previously shown that the ER stress response is elevated in y*arf1-11* (ref. 23). Therefore, we asked whether the increase in LD biogenesis was due to ER stress. We deleted the two mammalian *FIT2* homologues *SCS3* and *YFT2* known to connect ER stress response and LD biogenesis[45], and to maintain cellular proteostasis and membrane lipid homeostasis at the ER in WT yeast (Fig. 6e)[46]. In these strains, LD accumulation and mitochondrial morphology remained unaffected and did not phenocopy y*arf1-11*. Thus, TAG accumulation and increased LD biogenesis were not a secondary effect due to ER stress in the y*arf1-11* strain.

It has been previously reported that the number of LDs is increased also in a Δ*arf1* mutant[47], a phenotype we confirmed (Fig. 6f and Extended Data Fig. 4d,e). Similar to the y*arf1-11* strain, the increase in LD number correlated with an increase in TAG levels (Fig. 6g) but not of SE (Extended Data Fig. 4f). Since both the absence of yArf1 and presence of a hyperactive yArf1 mutant induced LD formation, we asked whether Arf1 cycling between active and inactive states is needed to control LD number. LD number was significantly increased when we expressed either yArf1-CA or yArf1-DN (Extended Data Fig. 4g). This function of yArf1 on LD is independent of its role in conjunction with COPI components (Extended Data Fig. 4h). Thus, tightly controlled yArf1 activity on LD regulates TAG levels and LD number.

## yArf1 regulates LD-associated functions and β-oxidation

The TAG accumulation under dysregulated yArf1 activity could either be due to alteration in LD proliferation or to perturbations in FA efflux from LDs, or both. First, we tested whether yArf1-11 was involved in LD proliferation and TAG synthesis. To induce LD proliferation, we grew y*ARF1* and y*arf1-11* cells in the presence of saturated FAs (+SFA), which resulted in a growth defect for y*arf1-11* cells already at 23 °C. This phenotype was exacerbated when we blocked endogenous FA synthesis by cerulenin (+SFA+Cer) or by deleting the TAG synthases (Δ*lro1*Δ*dga1*) (Extended Data Fig. 5a). Moreover, Dga1 co-immunoprecipitated with both yArf1 and yArf1-11 (Fig. 7a,b). Consistently, yArf1 occasionally co-localized with or was juxtaposed to Dga1 and Lro1, while yArf1-11 co-localized with the ER pool of Dga1 and Lro1 at 23 °C and the LD pool of Dga1 at 37 °C (Fig. 7c and Extended Data Fig. 5b). These data suggest that yArf1 could positively influence TAG synthesis.

Next, we asked whether yArf1 could also be involved in FA efflux from LDs. TAGs in LD are converted into free FA by triglyceride lipases, activated by Faa1/Faa4 on LDs, and then activated free FA are imported into peroxisomes for β-oxidation (Fig. 7a). Both the triglyceride lipase Tgl4 and the acyl-CoA synthase Faa1 were co-immunoprecipitated with yArf1-11 and to a lesser extent with yArf1 (Fig. 7d,e), suggesting that yArf1 negatively regulates FA efflux from LDs. Thus, we measured TAG mobilization in y*ARF1* and y*arf1-11* cells. Because of the y*arf1-11* temperature sensitivity (Fig. 1c), we were unable to perform long kinetics. Still, we observed a slight impairment in TAG mobilization from LDs in y*arf1-11* compared with y*ARF1* (Extended Data Fig. 5c,d), consistent with the possibility that Arf1 negatively affects TAG mobilization.

Alterations in FA efflux might also affect peroxisomes. We could, however, not detect any difference in peroxisome number (Fig. 7f and Extended Data Fig. 5e). yArf1 and mArf1 can bind to peroxisomes in vitro[37,48] and Pex35 functionally interacts with yArf1 (ref. 39). Likewise, we detected interactions between yArf1 and yArf1-11 with the peroxisomal protein Pex13 (Extended Data Fig. 5f). However, neither yArf1–GFP nor yArf1-11–GFP co-localized with peroxisomes, but were rather found on juxtaposed structures (Extended Data Fig. 5g).

Next, we determined whether FA import into peroxisomes or β-oxidation was defective in y*arf1-11* (Fig. 7g). The two subunits of the obligate heterodimeric FA transporter Pxa1 and Pxa2 were affected in y*arf1-11*. Pxa1 levels were strongly reduced at 37 °C and Pxa2 was virtually absent, both at 23 °C and 37 °C (Fig. 7h,i). Likewise, the first and rate-limiting enzyme involved in β-oxidation, the acyl-CoA oxidase Pox1, was almost undetectable at 37 °C in y*arf1-11*, while its levels were increased in y*ARF1* under the same condition. Consistent with the functional peroxisome biogenesis, not all peroxisomal proteins were affected in y*arf1-11*. The level of the very-long-chain FA transporter Fat1 and two other enzymes of the β-oxidation cascade, Fox2 and Pot1, were

---

**Fig. 4 | LD-localized yArf1 induces mitochondria fragmentation.**
**a**, Localization of yArf1, constitutively active (CA) or dominant negative (DN) forms of yArf1 fused to GFP grown at 23 °C or shifted to 37 °C for 30 min. Constructs were expressed from the centromeric low copy number plasmid pGFP33. Scale bar, 5 μm. **b**, Localization of yArf1, or yArf1 bearing single (K38T, L173S), double (K38T–E132D) or triple (K38T–E132D–L173S) substitution y*arf1-11* mutations fused to GFP in *Saccharomyces cerevisiae* (YPH500) grown at 23 °C or shifted to 37 °C for 30 min. Constructs were expressed from the centromeric low-copy-number plasmid pGFP33. Scale bar, 5 μm. **c**, Growth assay of the WT strain bearing the empty pGFP3 vector (+EV), single (K38T, L173S), double (K38T–E132D) or triple (K38T–E132D–L173S) y*arf1-11* mutations fused to GFP on rich YPD plates incubated at 23 °C, 30 °C or 37 °C. **d**, Schematic of the construct designed to anchor yArf1 on the LD via the PAT domain of the perilipin *PLN1*. y*ARF1* deleted in its myristoylation sequence (ΔN17) was expressed from its endogenous promoter and fused to GFP on its 3′ end. Localization of LD-anchored ΔN17-yArf1–GFP, the constitutively active mutant yArf1-CA, or yArf1 bearing yArf1-11–GFP variant in cells depleted of *ARF1* grown at 23 °C and shifted to 37 °C. Tom70–mCherry was used as a mitochondrial marker. Mitochondria phenotypes (tubular, mixed or globular) were measured. Mean and standard deviation are shown. At 23 °C, Δ*yarf1* = 406 cells, Δ*yarf1* + LD-Arf1 = 477 cells, Δ*yarf1* + LD-Arf1-11 = 483 cells, Δ*yarf1* + LD-Arf1-CA = 403 cells; At 37 °C, Δ*yarf1* = 443 cells, Δ*yarf1* + LD-Arf1 = 480 cells, Δ*yarf1* + LD-Arf1-11 = 523 cells, Δ*yarf1* + LD-Arf1-CA = 529 cells from *n* = 3 biological replicates. Scale bar, 5 μm. **e**, Schematic of the construct designed to anchor yArf1 on the ER via Sec66. y*ARF1* deleted in its myristoylation sequence (ΔN17) was expressed from its endogenous promoter and fused to GFP on its 3′ end. Localization of ER-anchored ΔN17-yArf1–GFP or yArf1 bearing yArf1-11–GFP variant in Δ*yarf1* cells grown at 23 °C and shifted to 37 °C. Scale bar, 5 μm. **f**, Growth assay of the ER-anchored ΔN17-yArf1–GFP or Arf1 strains bearing y*arf1-11* mutations (Arf1^K38T–E132D–L173S) on rich YPD plates or synthetic medium lacking uracil (HC-Ura) incubated at 23 °C, 30 °C or 37 °C, and of the ER-anchored ΔN17-yArf1–GFP or yArf1 bearing y*arf1-11* mutations (Arf1^K38T–E132D–L173S) in YPH500 cells lacking y*ARF1* (Δ*yarf1*) on rich YPD plates or synthetic media lacking uracil (HC-Ura) incubated at 23 °C, 30 °C or 37 °C. **g**, Cells expressing yArf1/11 fused to GFP, or expressing ER-ΔN17-yArf1/11–GFP were grown at 37 °C for 30 min and mitochondria were imaged with Tom70–mCherry by high-resolution microscopy followed by deconvolution. A *z*-projection of maximum intensities is shown for each panel. Scale bar, 2 μm. **h,i**, Measurements of mitochondria phenotypes (tubular, mixed or globular) based on images taken in **g** (**h**), and mitochondrial fusion and fission events based on Supplementary Videos 9–12 (**i**). Mean and standard deviation are shown. Δ*yarf1* + yArf1–GFP = 364 cells, Δ*yarf1* + yArf1-11–GFP = 800 cells, Δ*yarf1* + ER-yArf1–GFP = 608 cells, Δ*yarf1* + ER-yArf1-11–GFP = 542 cells from *n* = 3 biological replicates (**h**); at 23 °C Δ*yarf1* + ER-yArf1–GFP = 210 cells, Δ*yarf1* + ER-yArf1-11–GFP = 208 cells and at 37 °C Δ*yarf1* + ER-yArf1–GFP = 195 cells, Δ*yarf1* + ER-yArf1-11–GFP = 190 cells from *n* = 3 biological replicates (**i**). Source numerical data are available in source data.

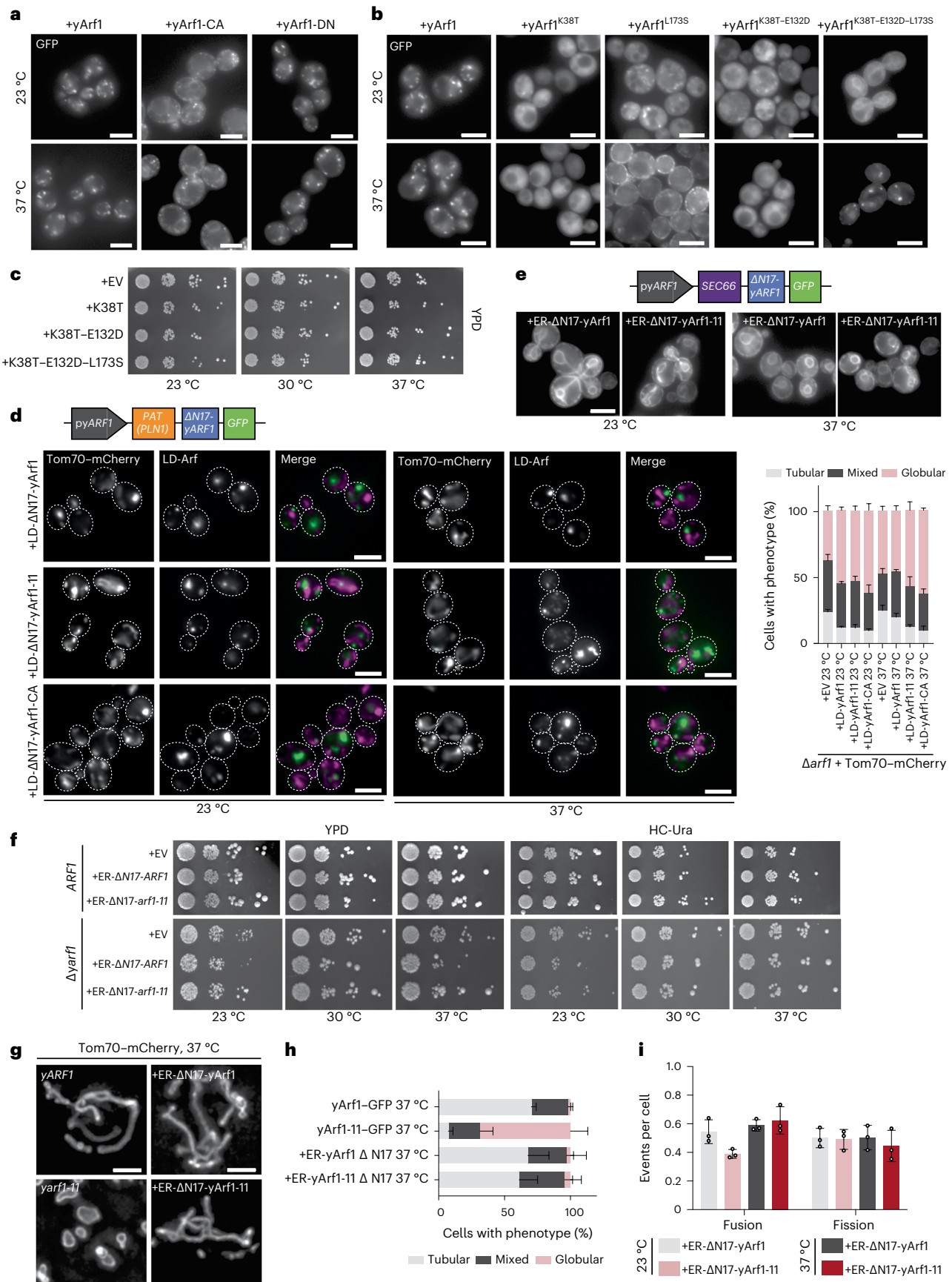

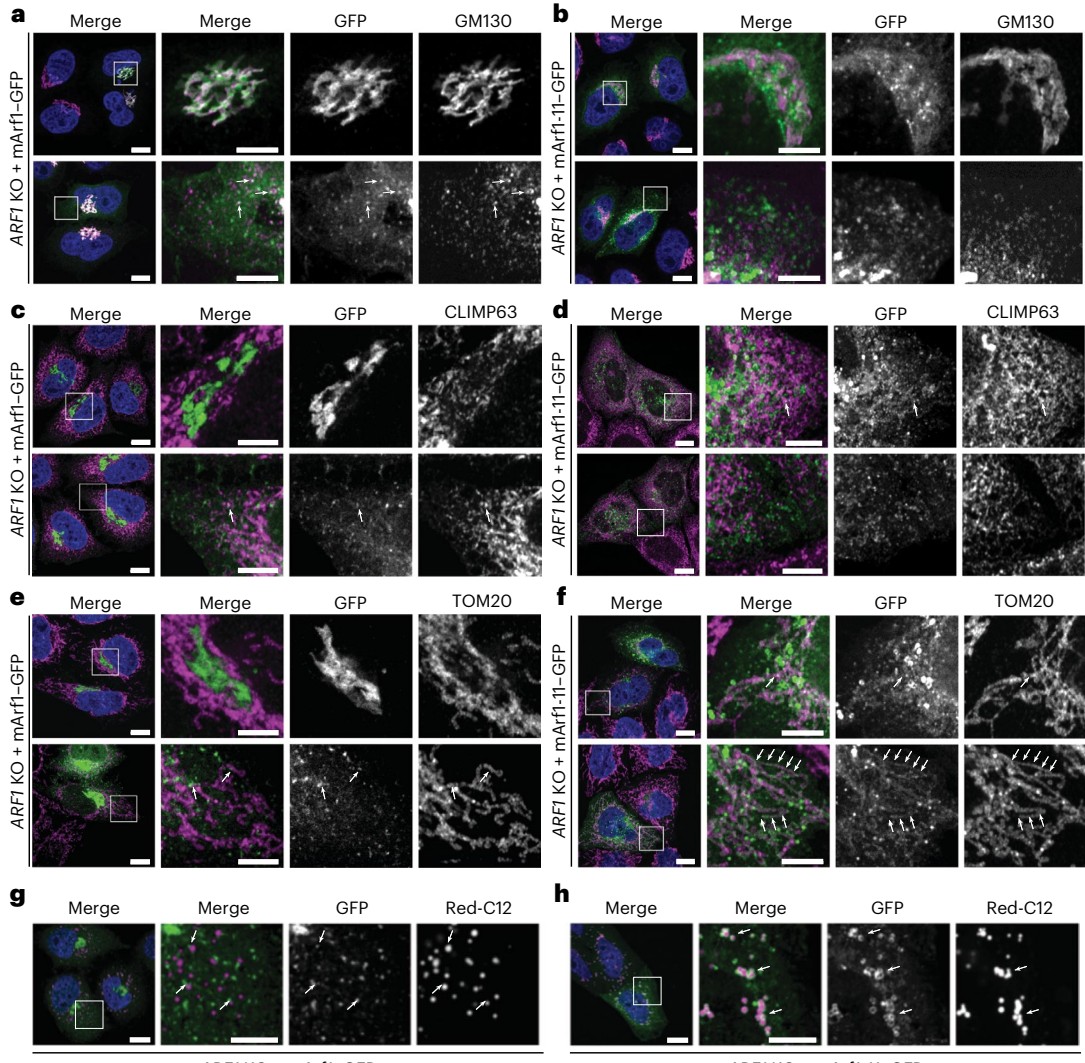

**Fig. 5 | Functional conservation of Arf1-11 in mammalian cells. a,b,** Mammalian Arf1 (mArf1; **a**) or mArf1-11 (**b**) fused to GFP were expressed in CRISPR/Cas9-mediated *ARF1* knockout HeLa cells (*ARF1* KO). Co-localization with the Golgi was done by immunostaining against the marker GM130. Squares show magnification of a perinuclear and distal portion of the cell. Scale bars, 10 μm and 5 μm (inlays). **c,d,** Co-localization of mArf1–GFP (**c**) and mArf1-11–GFP (**d**) expressed in the *ARF1* KO cell line with the ER was determined by immunostaining against the marker CLIMP63. Squares show magnification of a perinuclear and distal portion of the cell. Scale bars, 10 μm and 5 μm (inlays). **e,f,** Co-localization of mArf1–GFP (**e**)

and mArf1-11–GFP (**f**) expressed in the *ARF1* KO cell line with mitochondria was determined by immunostaining against the translocase of mitochondrial outer membrane TOM20. Squares show magnification of a perinuclear and distal portion of the cell. Scale bars, 10 μm and 5 μm (inlays). **g,h,** Co-localization of mArf1–GFP (**g**) and mArf1-11–GFP (**h**) expressed in the *ARF1* KO cell line with LDs was determined by incubation with the fluorescent fatty-acid BODIPY Red-C12. Scale bars, 10 μm and 5 μm (inlays). Squares show magnification of distal portion of the cell. All images were acquired 24 h after transfection. See also Extended Data Fig. 3.

not altered in y*arf1-11*. The reduction of Pxa1, Pxa2 and Pox1 levels was not due to transcriptional regulation as we did not observe any decrease in the mRNA levels of all three genes in *yarf1-11* at 37 °C (Extended Data Fig. 5h), indicating that the regulation of Pxa1, Pxa2 and Pox1 occurs post-transcriptionally.

The product of β-oxidation, acetyl-CoA is transferred to mitochondria for ATP production. Our data indicate that *yarf1-11* might be defective in acetyl-CoA synthesis using FA as substrate. Besides FAs, acetate can also be metabolized by yeast cells to produce acetyl-CoA. In the presence of 0.3 M sodium acetate, none of the y*arf1-11* mutant strains grew at the semi-permissive temperature 30 °C (Extended Data Fig. 5i). Thus, our data indicate that y*arf1-11* is defective in acetyl-CoA synthesis. Taken together, our results suggest a function for Arf1 in either directly or indirectly regulating TAG synthesis and metabolism, peroxisome function and thereby FA flux into mitochondria.

**Acetyl-CoA transfer loss leads to mitochondria fragmentation**

We hypothesized that disruption of FA flux into mitochondria would affect mitochondria function and morphology. To test this hypothesis, we first deleted the two TAG synthases *LRO1* and *DGA1* and determined mitochondrial morphology. As expected, the LD marker Erg6 remained in the ER resulting from a lack of TAG and LD biogenesis (Extended Data Fig. 6a). Consistently, the proportion of cells harbouring globular mitochondria was increased (Extended Data Fig. 6b,c). Next, we investigated the impact of FA deprivation on mitochondrial morphology by treating cells with cerulenin[49] (Fig. 8a–c). Cerulenin treatment efficiently reduced the levels of FAs and LDs (Extended Data Fig. 6d), and slowed down growth (Extended Data Fig. 6e). Under these conditions, the fraction of cells with globular mitochondria increased drastically (Fig. 8a,b). Moreover, abolishing β-oxidation *(Δpox1* or *Δpot1)* likewise resulted in globular mitochondria (Extended Data Fig. 6f). Thus, disruption of FA metabolism impairs mitochondria morphology.

To corroborate the results above and to show that FA flux into mitochondria is impaired in y*arf1-11*, we followed the transport of the red-fluorescent FA derivative Bodipy C12 (Red-C12) to mitochondria (Fig. 8d). This fluorescent FA has been shown to be incorporated into LD-specific neutral lipids[14,50–53], and to be metabolized by β-oxidation[14,16]. After 30 min, Red-C12 efficiently reached mitochondria in y*ARF1* (Fig. 8e), while it mainly remained in structures reminiscent of ER and LD, and was rarely transferred to mitochondria in y*arf1-11* (Fig. 8e). Interestingly, Red-C12 transport into mitochondria was impaired in cells in which yArf1, yArf1-11 or yArf1-CA were locked on LDs (Extended Data Fig. 7a), but efficiently transported when yArf1-11 was anchored to the ER, consistent with the notion that Arf1 negatively regulates FA efflux from LDs (Extended Data Fig. 7b). However, abolishing peroxisome biogenesis *(Δpex3Δpex19* (ref. 54)) did not affect the mitochondria phenotype (Extended Data Fig. 6f), and did not reduce the flow of Red-C12 to mitochondria (Extended Data Fig. 7c). These data suggest that loss of β-oxidation is more detrimental than the loss of peroxisomes altogether for efficient FA transfer. Taken together, our data provide evidence that Arf1 on LDs negatively regulates FA efflux from LDs and thereby contribute to the fraction of globular mitochondria.

FA transport between LDs, peroxisomes and mitochondria occurs at organellar contact sites[10]. Impairing peroxisome–mitochondria contacts by deletion of the *PEX34* tether in y*ARF1* and y*arf1-11* led only to a small increase of cells with globular mitochondria (Extended Data Fig. 7d), much weaker than the *yarf1-11* phenotype. Additional tethers might be involved, which is also supported by previous findings[10,55]. Moreover, our EM data revealed an increase of ER–mitochondria contacts both in number and length, and also between LDs and mitochondria in *yarf1-11* (Extended Data Fig. 7e–h). We surmise that this increase in organellar contacts might represent a compensatory mechanism. Taken together, our data indicate that β-oxidation is impaired in y*arf1-11* at the restrictive temperature and hence no acetyl-CoA can be transferred to mitochondria, which ultimately leads to mitochondrial fragmentation.

### OXPHOS activity and ATP synthesis are impaired in y*arf1-11*

Mitochondria fragmentation has been described as a general mechanism in response to various types of stress[56], such as ATP synthase inhibition[57], oxidative stress[58], or loss of mitochondrial membrane potential ($\Delta\psi_m$)[59,60]. We therefore hypothesized that the lack of metabolite transfer from peroxisomes to mitochondria leads to decreased ATP synthesis.

y*arf1-11* did not grow on plates containing a non-fermentable carbon source (glycerol) with or without the ATP synthase inhibitor oligomycin (+Oligo), suggesting defects in RC function (Extended Data Fig. 8a). Moreover, y*arf1-11* exhibited reduced $\Delta\psi_m$ (Extended Data Fig. 8b), together with impaired ATP synthesis, but not hydrolysis (Extended Data Fig. 8b,c). We confirmed these observations by directly measuring ATP synthesis and hydrolysis rates on purified mitochondria (Extended Data Fig. 8d,e), at the cellular level by using a

fluorescence resonance energy transfer (FRET)-based ATP nanosensor[61] (Extended Data Fig. 8f) and after 30, 120 and 360 min incubation at 37 °C (Extended Data Fig. 8g). In all cases, the outcome was a lower ATP level in y*arf1-11*. Conversely, ATP synthesis was not affected in *Δarf1* (Extended Data Fig. 8h), most likely due to the fact that only 40% of the mitochondria were globular in this strain (Fig. 2d). The inability of the y*arf1-11* to synthesize ATP was attributed to a decrease in oxygen consumption (that is, lower respiratory rate; Extended Data Fig. 8i) and not due to uncoupled oxidative phosphorylation (P/O; Extended Data Fig. 8j). Our data indicate that the low ATP synthesis rate in y*arf1-11* is a direct consequence of the low respiratory rate.

### Arf1 controls acetyl-CoA flux into mitochondria in mammals

We tested next whether Arf1's role in FA metabolism was conserved in mammalian cells. We performed a Red-C12 pulse-chase assay using *ARF1-KO* cells expressing mArf1–GFP or mArf1-11–GFP as reported previously[14]. Cells were pulsed for 16 h in complete medium (CM), and chased in nutrient-deprived media (Hanks' Balanced Salt Solution, HBSS) for 0 or 9 h (Fig. 8f). Under these conditions, FAs were present in LDs both in cells expressing mArf1 or mArf1-11 at the 0 h timepoint (Fig. 8g). While Red-C12 was efficiently transferred from LDs to mitochondria in mArf1 expressing cells after 9 h of starvation, the dye persisted in LDs in mArf1-11 cells (Fig. 8h,i). Thus, Arf1 plays an evolutionarily conserved role in acetyl-CoA transfer to mitochondria.

Taken together our results provide evidence that TAG accumulation in LDs in yeast and mammalian cells expressing the hyperactive Arf1-11 is a consequence of reduced FA metabolism and acetyl-CoA transfer into mitochondria. As a consequence, energy production is impaired, leading to mitochondrial fragmentation.

### Arf1 is present at organellar contact sites

The flux of the lipid metabolites from LDs to peroxisomes and mitochondria happens through contact sites between these organelles[10]. Arf1 appears to regulate FA efflux from LDs, and therefore might be present at the contact sites. To test this possibility, we simultaneously labelled LDs, peroxisomes, mitochondria and yArf1 (Extended Data Fig. 9a–c). yArf1 was present at contacts between LDs and mitochondria, LDs and peroxisomes, and peroxisomes and mitochondria. More importantly, we also detected yArf1 at tripartite organellar contacts. Therefore, yArf1 is at the right location to regulate FA and acetyl-CoA flux from LDs to mitochondria. This localization was also observed in mammalian cells. mArf1 was present at mitochondria–LD, mitochondria–peroxisomes and mitochondria–LD–peroxisomes contact sites under normal (fed) conditions, nutrient starvation or in the presence of oleate (Extended Data Fig. 10a–c). Thus, Arf1 plays a conserved role in FA metabolism at organellar contact sites (Extended Data Fig. 10d).

## Discussion

Here we provide evidence that Arf1 regulates mitochondria functions via two independent mechanisms. First, yArf1 is required for both fission and fusion of mitochondria, and we establish that yArf1 is present

---

**Fig. 6 | Hyperactive Arf1 induces TAG accumulation. a**, TEM of y*ARF1* and y*arf1-11* strains grown at 37 °C for 30 min. Scale bars, 2,000 nm. **b**, LipidTox staining of LDs in y*ARF1* and y*arf1-11* strains grown at 23 °C or shifted to 37 °C. Scale bar, 5 µm. Mean and standard deviation are shown; y*ARF1* 23 °C = 408 cells, y*ARF1* 37 °C = 408 cells, y*arf1-11* 23 °C = 421 cells, y*arf1-11* 37 °C = 422 cells from *n* = 3 biological replicates; two-way ANOVA using Sidak's multiple comparison, ***$P$ = 0.0009, **$P$ = 0.0032. **c**, Nile Red staining of LDs in parental HeLa cells (control), *ARF1* KO HeLa cells, and *ARF1* KO HeLa cells expressing mArf1 or mArf1-11. For each cell line, the numbers of LD were quantified. Images were acquired 24 h after transfection. Mean and standard deviation are shown; HeLa control = 182 cells, *ARF1* KO = 181 cells, *ARF1* KO +mArf1 = 163 cells, *ARF1* KO +mArf1-11 = 151 cells from *n* = 3 biological replicates; unpaired two-tailed *t*-test, **$P$ = 0.0096. Scale bar, 5 µm. **d**, Measurements of TAG in the y*ARF1* and y*arf1-11* strains grown at 23 °C or shifted to 37 °C. Mean and standard deviation are shown from *n* = 3 biological replicates; unpaired two-tailed *t*-test, *$P$ = 0.0349; **$P$ = 0.002. **e**, LDs (Erg6) and mitochondria (Tom70) morphologies imaged in the y*ARF1* and y*arf1-11* parental strains and in strains deprived of *SCS3* and *YFT2* (Δ*scs3* Δ*yft2*) grown at 23 °C or shifted to 37 °C. **f**, LipidTox staining of LDs in WT and Δ*arf1* strains grown at 23 °C or shifted to 37 °C. Mean and standard deviation are shown; WT 23 °C = 669 cells, WT 37 °C = 662, Δ*arf1* 23 °C = 673 cells, Δ*arf1* 37 °C = 659 cells from *n* = 3 biological replicates Two-way ANOVA using Sidak's multiple comparison, WT versus Δ*arf1* 23 °C *$P$ = 0.0032, WT versus Δ*arf1* 37 °C **$P$ = 0.001. **g**, Measurements of TAG in the WT and Δ*yarf1* strains grown at 23 °C or shifted to 37 °C. Mean and standard deviation are shown from *n* = 3 biological replicates; unpaired two-tailed *t*-test, ***$P$ = 0.0002; ****$P$ = 0.00000396. Source numerical data are available in source data. See also Extended Data Fig. 4.

at sites on mitochondria where either fusion or fission occurs. Second, Arf1 regulates mitochondrial function by controlling the flow of FAs and metabolites from LDs to peroxisomes/mitochondria in yeast and in mammalian cells (Fig. 8j).

Previously, roles for Arf1 and the COPI coat in peroxisome biogenesis and function and on LDs have been established[36–39,48,62]. However, the Arf1 function we reveal here is independent of COPI. For example, coatomer mutants did not accumulate LDs. Likewise, peroxisome biogenesis appeared unaffected, but they were non-functional in

*yarf1-11*. This functional defect is due to (1) strongly reduced acyl-CoA import into peroxisomes and (2) the almost complete absence of the acyl-CoA oxidase Pox1. We propose that FA remain stored in LDs because they cannot be metabolized in peroxisomes. This scenario is supported by the finding that deletion of Pox1 or Pxa1 in yeast leads to increased TAG levels in LDs[63]. Strikingly, abolishing peroxisome biosynthesis had less severe consequences than blocking β-oxidation. It is conceivable that, in the absence of peroxisomes, an alternative pathway through the ER could be activated, and that FA would flow

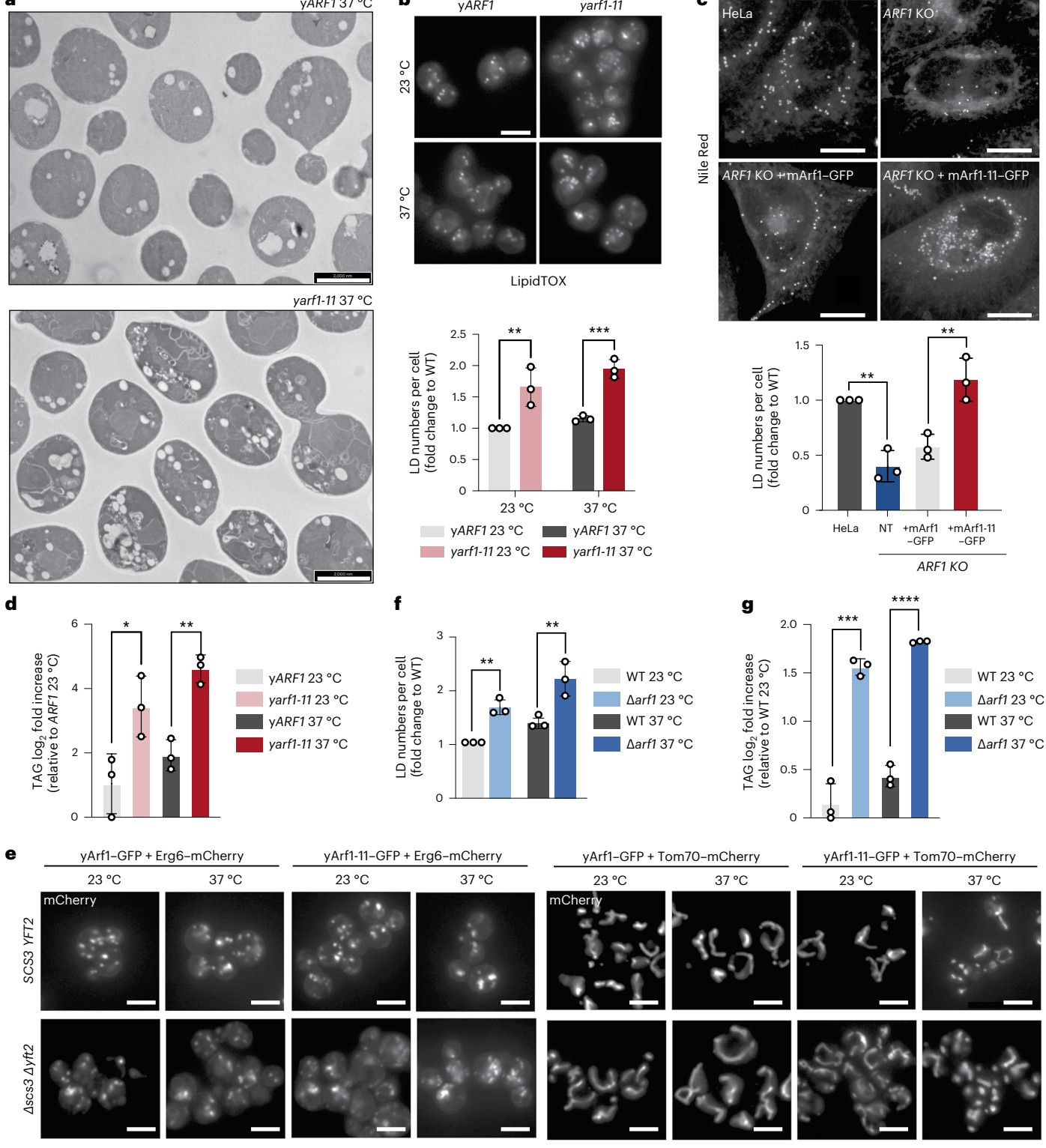

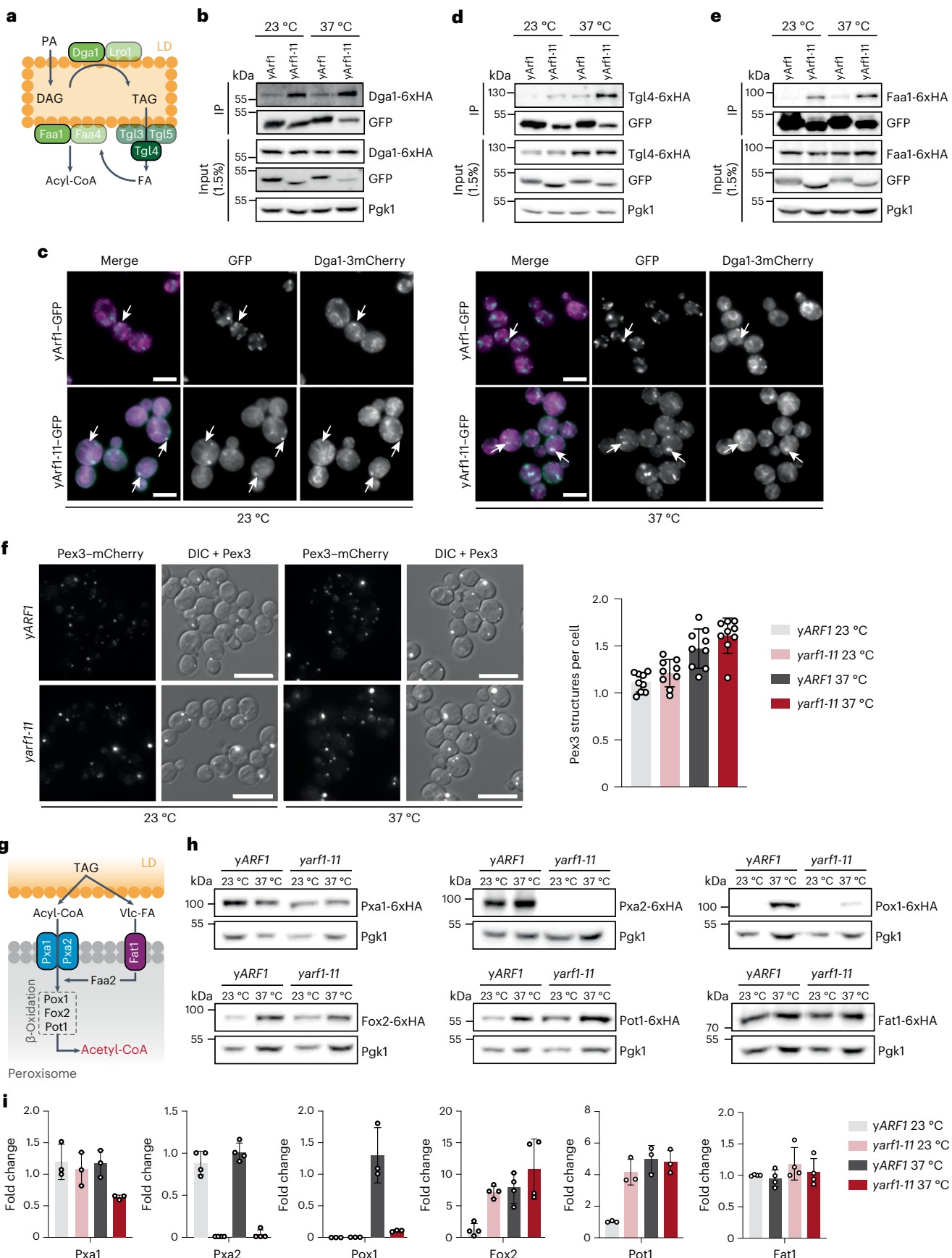

**Fig. 7 | yArf1 regulates LD-associated functions and β-oxidation. a,** Schematic of TAG synthesis and breakdown on LD. Key enzymes involved in TAG synthesis (Dga1), hydrolysis (Tgl4), and FA activation (Faa1) used in our co-IP experiments are shown in green. **b,d,e,** Co-IP of yArf1–GFP and yArf1-11–GFP with Dga1-6xHA (**b**), Tgl4-6xHA (**d**) and Faa1-6xHA (**e**). Strains were grown at 23 °C or shifted to 37 °C. **c,** Co-localization of yArf1–GFP and yArf1-11–GFP with the diacylglycerol acyltransferase Dga1 tagged with 3xmCherry grown at 23 °C or shifted to 37 °C for 30 min. Arrows indicate sites of co-localization or juxtaposition between the yArf1/yArf1-11 and Dga1 on the ER or on LD. Scale bars, 5 μm. **f,** Peroxisome biogenesis followed by microscopy using the peroxisomal marker Pex3 fused to mCherry in the y*ARF1* and y*arf1-11* strains grown at 23 °C or shifted to 37 °C (left). Quantification of peroxisomes per cell in each strain and condition (right). Mean

and standard deviation are shown; y*ARF1* + Pex3–mCherry 23 °C = 1,496 cells, y*ARF1* + Pex3–mCherry 37 °C = 1,070 cells, y*arf1-11* + Pex3–mCherry 23 °C = 1,025 cells, y*arf1-11* + Pex3–mCherry 23 °C = 970 cells from n = 3 biological replicates. Scale bars, 5 μm. **g,** Schematic of TAG mobilization to synthesize acetyl-CoA by peroxisomal β-oxidation in yeast. Relevant proteins monitored in **h** are shown. Vlc-FA, very-long-chain FAs. **h,** Immunoblot analysis of all β-oxidation proteins, both acyl-CoA transporters and the Vlc-FA transporter genomically fused to 6xHA in the y*ARF1* and y*arf1-11* strains grown at 23 °C or shifted to 37 °C. Pgk1 was used as loading control. **i,** Relative fold changes in protein levels from immunodetections done in **h**. Mean and standard deviation are shown from n = 3 biological replicates. Source numerical data and unprocessed blots are available in source data. See also Extended Data Fig. 5.

from LDs to the ER and from there to mitochondria. This pathway would only be activated in the absence of peroxisomes and since peroxisomes, albeit non-functional, are still present in *arf1-11*, FA transfer is blocked entirely. In the presence of peroxisomes, neutral lipids accumulate in the ER, if transfer into LDs is abolished[64–66]. At any rate, as a consequence of reduced acyl-CoA import into peroxisomes and the absence of Pox1, the transfer of acetyl-CoA to mitochondria is reduced, which leads to substrate depletion for the TCA cycle and ultimately to reduced ATP production, which in turn causes mitochondrial fragmentation. We propose that the two ways by which Arf1-11 interferes with mitochondrial morphology and function drives cell death. Remarkably, the roles of Arf1 in acyl-CoA flow from LDs to mitochondria is conserved from yeast to mammals. While β-oxidation occurs in peroxisomes in yeast, it is also a mitochondrial function in mammalian cells.

mArf1 and yArf1 are found at contact sites between LDs and mitochondria, LDs and peroxisomes and at tripartite contacts, suggesting that Arf1 locally regulates organellar contacts. Moreover, the hyperactive yArf1-11 increased contacts between mitochondria, LDs and the ER. We and others have indicated that Arf1 and Arf1-11 are present on LDs[36]. Here we show that yArf1 co-precipitates with Dga1, which also participates in ER-LD tethering in mammalian cells[67]. Thus, even though co-immunoprecipitation (co-IP) does not allow us to distinguish between direct and indirect interactions, Arf1 could be involved in the regulation of tethering or TAG synthesis, or both. Likewise, Arf1 might function in LD–peroxisome tethering. The tethers between LDs and peroxisomes remain elusive. It has been speculated that proteins required for lipolysis and acyl-CoA production could act as potential tethers on LDs[68]. Consistent with this possibility, yArf1 co-precipitated with the triglyceride lipase Tgl4 and the acyl-CoA synthetase Faa1. In addition, yArf1-11 and mArf1-11 had a negative effect on Red-C12 transfer

into peroxisomes/mitochondria and on TAG mobilization from LDs. We also detected Arf1 at contacts between LDs and mitochondria. Perlipins PLIN1 and PLIN5, MIGA2 and the mitofusin MFN2 were implicated as tethers in LD–mitochondria contacts in adipose tissue[69–71]. The perilipin/mitofusin tether has been proposed to be conserved between yeast and mammals[72]. Interestingly, in yeast *arf1-11* cells the mitofusin Fzo1 formed aggregates, a phenotype that was alleviated by overexpression of Cdc48 (ref. [26]).

We provide evidence that Arf1 is present at bi- and tripartite organellar contacts involving mitochondria, peroxisomes and LDs, in both yeast and mammalian cells. In y*arf1-11*, the contacts between mitochondria, LDs, and the ER are increased, often involving all three organelles. We cannot detect peroxisomes in our EM method. It is tempting to speculate, however, that contact sites might exist in which all four organelles come together. We envisage Arf1 to be a regulator of all these contact sites. How Arf1 regulates proteins at these contacts sites or whether Arf1 acts directly or indirectly on the TAG synthase Dga1 and the lipase Tgl4 or how it can assert its effects on peroxisomal proteins in *trans* remain to be established.

Strikingly, Arf1 appears to be a Jack of all trades. Historically, Arf1 has been mostly implicated in vesicle transport pathways. We have previously shown that Arf1 is also involved in mRNA transport and metabolism[23]. Moreover, Arf1 is also present on the ER, peroxisomes, LDs and mitochondria[26–28,36,48,62]. While most small GTPases have mostly rather precise set of effector molecules that they recruit, Arf1 could be much more promiscuous, since some of the processes it is involved in may not require either COPI or clathrin or its adaptors. At least, Arf1's roles in mitochondria dynamics[26] and FA efflux from LDs appear to be independent of the COPI vesicle coat. A recent study found that COPI does affect mitochondrial function and LD size in a process that is apparently not linked to TAG synthesis but rather a feedback from ROS

**Fig. 8 | Acetyl-CoA transfer loss leads to mitochondria fragmentation.**
**a,** Mitochondrial morphology imaged in the y*ARF1* and y*arf1-11* strains grown at 23 °C or shifted to 37 °C and treated with either DMSO or the FA synthesis inhibitor cerulenin. Scale bars, 5 μm. **b,** Quantification of the mitochondrial phenotypes observed in **a.** Mean and standard deviation are shown; y*ARF1* + DMSO 23 °C = 751 cells, y*ARF1* + DMSO 37 °C = 935 cells, y*ARF1* + cerulenin 23 °C = 501 cells, y*ARF1* + cerulenin 37 °C = 654 cells, y*arf1-11* + DMSO 23 °C = 616 cells, y *arf1-11* + DMSO 37 °C = 740 cells, y *arf1-11* + cerulenin 23 °C = 471 cells, y *arf1-11* + cerulenin 37 °C = 667 cells from at least n = 3 biological replicates. **c,** Metabolic pathway leading to TAG synthesis. FAs are used to produce phosphatidic acid (PA), which can be further converted to diacylglycerol (DAG) and to TAG inside LDs by the Lro1 and Dga1 enzymes. The FA synthesis inhibitor cerulenin inhibits TAG synthesis. **d,** Schematic of the experiment done in **e.** Yeast cells are first grown for 30 min at 37 °C, treated with BODIPY Red-C12 for another 30 min at 37 °C, washed and imaged. **e,** Acetyl-CoA transfer to mitochondria monitored in the yeast *ARF1* and *arf1-11* strains grown at 37 °C using the fluorescent FA BODIPY Red-C12. Co-localization of GFP signal (mitochondria) over Red-C12 one was measured using Mander's co-localization index. Mean and minimum to maximum are shown, box ranges from the first (Q1–25th percentiles) to the third quartile (Q3–75th percentiles) of the distribution; y*ARF1*

37 °C = 115 cells, y*arf1-11* 37 °C = 123 cells from n = 3 biological replicates; Unpaired two-tailed *t*-test, ****$P = 0.000000000000001$. **f,** Schematic representation of the FA pulse-chase assay. Cells were stained with BODIPY Red-C12 for 16 h in CM, washed and chased for 9 h in nutrient-depleted medium (HBSS). Then before imaging, cells were stained for 30 min with the MitoView dye. **g,h,** *ARF1 KO* cells expressing mArf1–GFP or mArf1-11–GFP were pulsed with BODIPY Red-C12 for 16 h, incubated 1 h in CM, transferred in HBSS (0 h; **g**) and chased HBSS for 9 h (**h**). BODIPY Red-C12 was initiated 24 h after mArf1 or mArf1-11 transfection. Scale bar, 10 μm. Scale bar inlays, 2 μm. **i,** Relative BODIPY Red-C12 localization measured by Pearson's co-localization index. Mean and minimum to maximum are shown, box ranges from the first (Q1–25th percentiles) to the third quartile (Q3–75th percentiles) of the distribution; *ARF1 KO* + mArf1 0 h = 114 cells, *ARF1 KO* + mArf1-11 0 h = 122 cells, *ARF1 KO* + mArf1 9 h = 145 cells, *ARF1 KO* + mArf1-11 9 h = 153 cells from n = 3 biological replicates; two-way ANOVA using Sidak's multiple comparison test, ****$P = 0.000000000000001$. NS, not significant. **j,** Schematic of the model we propose for Arf1 role in FA metabolization and how this affects maintenance of mitochondria morphology. For more details, see Discussion. Source numerical data are available in source data. See also Extended Data Figs. 6–10.

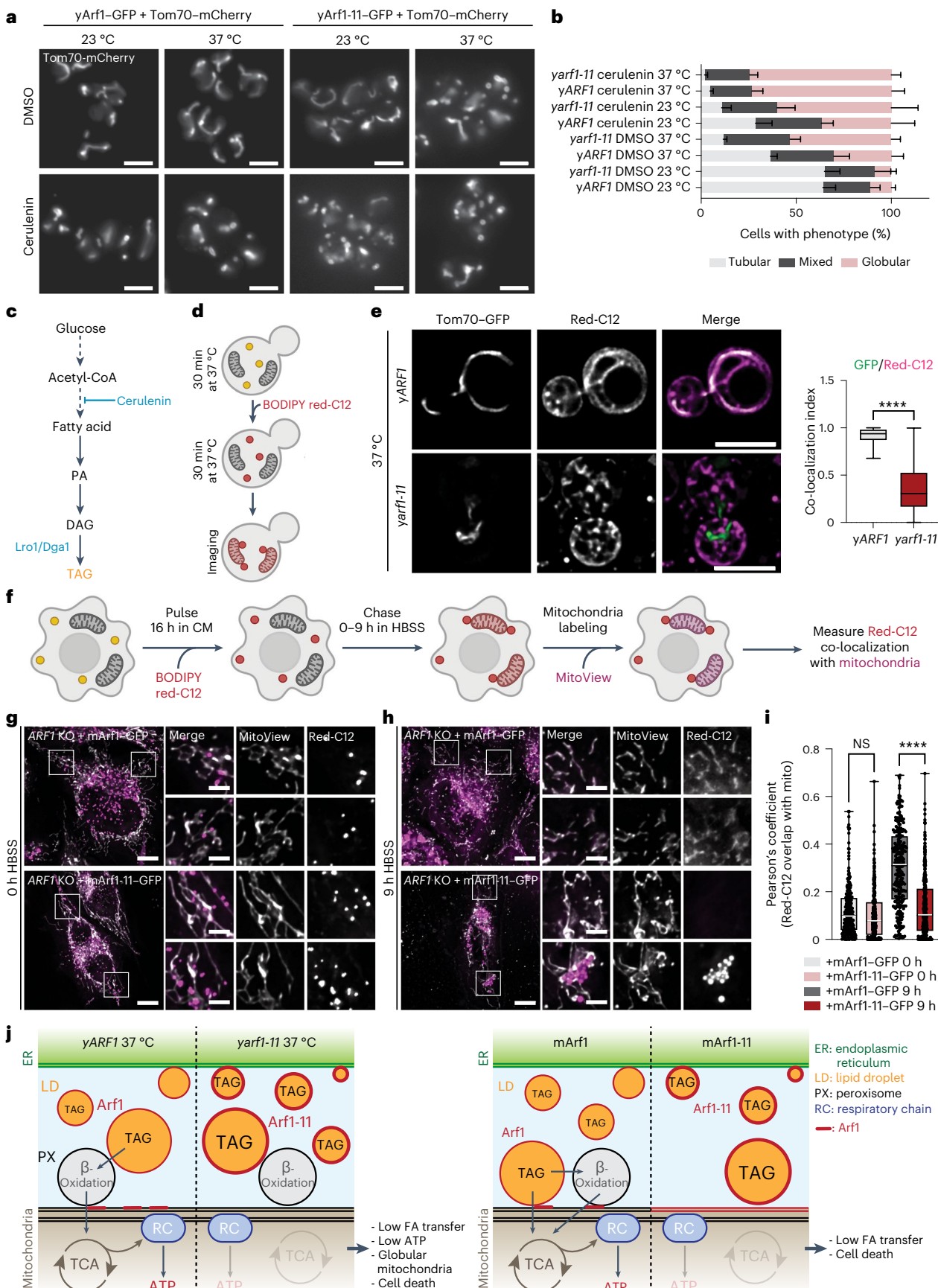

production[73]. Thus, Arf1 may affect mitochondrial morphology and function in COPI-dependent and COPI-independent ways. Certainly, more studies are required to reveal the full repertoire of Arf1 function.

## Online content

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

## Methods

### Strains, media and plasmids

Yeast strains were either grown in rich media composed of 1% w/v yeast extract, 1% (w/v) peptone, 40 mg l⁻¹ adenine, 2% (w/v) glucose (YPD) or 2% glycerol (YPGly), or in synthetic complete medium (HC) composed of 0.17% (w/v) yeast nitrogen base with ammonium sulfate and without amino acids, 2% (w/v) glucose and mixtures of amino acids (MP Biomedicals) depending on the auxotrophies used for selection. Unless otherwise indicated, cells were grown at 23 °C and a subset was shifted to 37 °C for 30 min before analysis. Inhibition of FA synthesis was done by treating cells with cerulenin (10 µg ml⁻¹; Alexis Biochemicals, Lausen, Switzerland) for 6 h, or with equal volume of dimethyl sulfoxide (DMSO) as a control. Solid media contained 2% (w/v) agar. YPD plates with SFAs contained 1% Brij58 (Fluka), 0.5 mM palmitic acid and 0.5 mM stearic acid (Sigma). Non-supplemented plates contained only YPD and 1% Brij58. YPD plates containing 0.33 M acetate pH 6 were prepared by adding 3 M sodium acetate (Ambion) to 1 litre sterilized YPD. ATP synthase activity was inhibited by adding 0.5 µg µl⁻¹ oligomycin to YPGly plates. Cloning into the pGFP33 plasmid was performed using the Gibson assembly kit (NEB). *ARF1* point mutations were generated with the Quick-change Mutagenesis kit (NEB) and KLD (Kinase, Ligase and DpnI enzymes) reactions, primers were generated with the NEBaseChanger website (http://nebasechanger.neb.com). For LD anchoring of Arf1, we subcloned the first 143 amino acids (PAT domain) of the yeast perilipin Pln1 (ref. [43]).

### Cell culture

*ARF1* KO HeLaα cells were established, mycoplasma tested and described elsewhere[44]. HeLa cells were grown in high-glucose Dulbecco's modified Eagle's medium (Sigma-Aldrich) with 10% foetal bovine serum (FBS, Biowest), 2 mM L-glutamine, 100 U ml⁻¹ penicillin G and 100 ng ml⁻¹ streptomycin, 1 mM sodium pyruvate at 37 °C and 7.5% CO₂. For transient cell transfections, cells were plated into six-well plates to reach 70% confluency the following day and transfected with 1 µg plasmid DNA complexed with Helix-IN transfection reagent (OZ Biosciences).

### Yeast transformation

Three units of $OD_{600}$ (optical density at 600 nm) of yeast cells were grown in appropriate YPD or HC media to mid-log phase. Cells were spun down and washed in 1 volume of 1× TE and 10 mM LiAc. The pellet was then resuspended in 350 µl of transformation mix (1× TE, 100 mM LiAc, 8.5% (v/v) single-stranded DNA and 70% (v/v) PEG3000), incubated with DNA (PCR product or plasmid) for 1 h at 42 °C, spun down (30 s at 10,000*g* at room temperature) and resuspended in 100 µl of YPD or HC media, and cells were plated onto selective media and incubated at 23 °C or 30 °C. Genomic tagging was done according to standard procedures[74].

### HeLa cell lines survival assay

Cells were seeded in 12-well plates at a density of 55,000 cells per well, which was confirmed by re-counting. Every 24 h for 6 consecutive days, cells from one well for each cell line were trypsinized, and resuspended in phosphate-buffered saline (PBS) complemented with 2% FCS, and GFP fluorescence from 100,000 cells per sample was measured by a Fortessa flow cytometer. After 3 days, all cell lines were trypsinized, diluted to 1:10 and transferred into fresh media.

### Microscopy

Fluorescence and DIC images were acquired with an ORCA-Flash 4.0 camera (Hamamatsu) mounted on an Axio Imager.M2 fluorescence microscope with a 63× Plan-Apochromat objective (Carl Zeiss) and an HXP 120C light source with ZEN 2.6 software. Image processing was performed using OMERO.insight client, and analysed with Fiji software. Measurement of number and length of contact sites on TEM images was done with Fiji software.

High-resolution images were acquired with an ORCA-Flash 4.0 camera (Hamamatsu) mounted on a FEI-MORE microscope with a 100× U Plan-S-Apochromat objective (Olympus).

To image Pex11-mScarlet strains, high-resolution imaging was performed at 23 °C and 37 °C using an automated inverted fluorescence microscope system (Olympus) harbouring a spinning disk high-resolution module (Yokogawa CSU-W1 SoRa confocal scanner with double microlenses and 50 µm pinholes). Images of cells in 96-well plates were made using a 60× oil lens (numerical aperture 1.42) and a Hamamatsu ORCA-Flash 4.0 camera. All images were taken in a *Z*-stack using cellSens software. Best focal planes were deconvoluted using cellSens software, and single planes or *Z*-projections of maximum intensity images were processed with the Fiji software.

In HeLa cell lines, mitochondria were stained by immunofluorescence using TOM20 antibody (1:200, Santa Cruz sc-17764) as marker, GM130 antibody (1:1,000, Cell Signalling 12480S) was used as Golgi marker, β-COP (1:500, gift from the Wieland lab) as COPI vesicles marker and CLIMP63 (1:1,000, gift from the Hauri lab) as ER marker. Secondary mouse (1:500) and rabbit (1:500) Alexa-Fluor 568 (A10037 and 110042, respectively, Invitrogen) antibodies were used and mounted with Fluoromount-G mounting medium (Thermo Fischer) containing 4′,6-diamidino-2-phenylindole. Images were acquired using a LSM700 Upright confocal laser-scanning microscope with the Zen 2.6 software (Zeiss) equipped with a Plan-Apochromat 63×/1.4 oil-immersion objective lens and two photomultiplier tubes.

To image mArf1 at LD–mitochondria and peroxisomes–mitochondria contact sites in HeLa cells, Arf1 KO cells were transfected with Arf1-EGFP and mPlum-PTS1. LDs and mitochondria were stained with Lipi-Blue (Dojindo Laboratories) staining and MitoTracker Deep Red (Invitrogen) dye respectively. Cells were imaged at 37 °C with a Zeiss Axio Observer wide-field microscope with a Plan-Apochromat N 63×/1.40 oil DIC M27 objective and a Photometrics Prime 95B camera. Filters with standard specifications for GFP, TexasRed and Cy5 were used to image Lipi-Blue, Arf1-EGFP, mPlum-PTS1 and MitoTracker Deep Red dye, respectively. *Z*-stack images were deconvolved with Huygens Professional software using the standard deconvolution method.

### Protein extraction and immunoblot analysis

For yeast cells, 10 ml of mid-log grown cultures were lysed at 4 °C in breaking buffer containing 50 mM Tris–HCl pH 8, 300 mM NaCl, 0.6% Triton X-100, 1 mM dithiothreitol (DTT) and 9 M urea, supplemented with half-volume of glass beads (0.25–0.5 mm; ROTH). Cell debris and unbroken cells were pelleted by centrifugation 3,000*g* for 5 min at room temperature. Equal protein concentration were loaded on 12% or 15% SDS–PAGE and transferred onto 0.45 µm nitrocellulose membranes (Amersham). Membranes were blocked with TBST (20 mM Tris, 150 mM NaCl, pH 7.6 and 0.1% Tween20) with 5% non-fat dry milk for 30 min and incubated with anti-HA primary antibody (1:5,000, Eurogentec 16B12) or anti-Pgk1 primary antibody (1:5,000, Invitrogen clone 22C5D8) overnight at 4 °C, followed by 2 h incubation with horseradish peroxidase (HRP)-conjugated secondary antibody (1:10,000; anti-mouse, Invitrogen 31430) in TBST. Chemiluminescence signals were detected using Immobilon Western HRP Substrate (Millipore) and imaged using a FusionFX (Vilber Lourmat).

Alternatively, before Gga2^GAT interaction, yeast cells were resuspended in 1 ml of 0.2 M sorbitol, 25 mM KPO₄ pH 7, 2 mM ethylenediaminetetraacetic acid (EDTA), 0.6% Triton X-100, 1× Halt proteases inhibitor cocktail (Thermo Scientific), transferred to Corex glass tubes filled with 500 µl glass beads (0.25–0.5 mm; ROTH) and broken 15 min by vortexing at 4 °C with 30 s intervals on ice. Unbroken cells and debris were pelleted at 3,000*g* for 5 min at 4 °C and supernatants (SNs) were transferred into new 1.5 ml Eppendorf tubes and pelleted at 100,000*g* for 30 min at 4 °C in a TLA 100-3 rotor. One millilitre of SN (S100) was saved for each sample, and the pellets (P100) were resuspended in 500 µl of Lysis buffer.

HeLa cell were lysed in 20 mM Tris–HCl pH 7.5, 150 mM NaCl, 10 mM MgCl$_2$, 1% Triton X-100, 2 mM phenylmethylsulfonyl fluoride (PMSF) and protease inhibitors, separated by 15% SDS–PAGE and transferred to Immobilon-P polyvinylidene difluoride membranes (Millipore). Membranes were blocked with TBST (20 mM Tris, 150 mM NaCl, pH 7.6 and 0.1% Tween20) with 5% non-fat dry milk for 1 h and incubated with primary antibody in TBST with 1% milk overnight at 4 °C: anti-Arf1 (1:2,500, Abnova MAB10011), and anti-actin (1:100,000, Sigma-Aldrich MAB1501). After washing, the membranes were incubated with HRP-conjugated secondary antibody (1:10,000; anti-rabbit, Sigma-Aldrich A0545 or anti-mouse, Sigma-Aldrich A0168) in TBST with 1% milk. Chemiluminescence signals were detected using Immobilon Western HRP Substrate (Millipore) and imaged using a FusionFX (Vilber Lourmat).

## Co-IP

Yeast cells were grown to mid-log phase, and $4 \times 10^8$ cells were lysed at 4 °C in IP buffer containing 25 mM Tris–HCl pH 7.5; 150 mM NaCl, 2 mM EDTA, 0.6% Triton X-100, 1 mM DTT, 1× protease inhibitor, supplemented with 100 µl glass beads (0.25–0.5 mm; ROTH). Cell debris and unbroken cells were pelleted by centrifugation 3,000$g$ for 5 min at 4 °C. The SN was then supplemented with 25 µl anti-GFP magnetic beads (Chromotek) and incubated 1 h at 4 °C under rotation. Beads were then washed four times in IP buffer, and proteins were eluted with 2× Laemmli buffer at 95 °C for 5 min.

## Arf1-Gga2 recruitment

**Gga2$^{GAT}$ expression and purification.** One litre of *Escherichia coli* BL21 strains harbouring the pGEX4-GST-GGA2$^{GAT}$ were induced with 0.5 mM isopropyl β-ᴅ-1-thiogalactopyranoside and cells were transferred from 37 °C to 30 °C for 3.5 h. Cells were then pelleted at 4,000$g$ at room temperature for 10 min, resuspended in 20 ml ice-cold 1× PBS/5 mM EDTA buffer (2.7 mM KCl, 1.5 mM KH$_2$PO$_4$, 137 mM NaCl, 5.6 mM Na$_2$HPO$_4$, 1.4 mM NaH$_2$PO$_4$ and 5 mM EDTA/NaOH pH 8.0) containing 1 mM PMSF and 1× Halt protease inhibitors. The resuspended cells were lysed by sonication seven times for 10 s (50% duty) on ice, cleared at 6,000$g$ at 4 °C for 30 min. The SN was transferred to ultracentrifuge tubes and further cleared at 100,000$g$ for 1 h at 4 °C. To isolate GST-fusion proteins the SN was added to 500 µl glutathione Sepharose magnetic beads and incubated for 1 h at 4 °C under rotation. Glutathione Sepharose beads were spun down at 500$g$ for 5 min and washed three times with 15 ml ice-cold 1× PBS/5 mM EDTA and twice with 1 ml 1× PBS/5 mM EDTA on a magnetic stand. Bound proteins were eluted by three consecutive treatments with 250 µl reduced glutathione buffer (20 mM reduced glutathione and 100 mM Tris–HCl pH 8.0) at 4 °C for 10 min incubation each time. SNs were dialysed in 2.5 litre dialysis buffer (10 mM HEPES/NaOH pH 7.8, 1 mM MgCl$_2$, 1 mM DTT and 0.2 mM PMSF) with slow stirring overnight at 4 °C. The next day, samples were centrifuged for 1 min at 20,000$g$ at 4 °C to remove precipitates. The SNs were frozen in liquid nitrogen in aliquots of 80 µg protein and stored at −80 °C.

**Gga2$^{GAT}$ pre-loading on glutathione magnetic beads.** Purified GST-GGA2$^{GAT}$ was loaded onto glutathione (80 µg per tube). To do so, 200 µl of resuspended glutathione magnetic beads were taken, vortexed for 10 s and washed twice in 500 and 400 µl 1× PBS, 0.5 mM EDTA on a magnetic stand. Beads were then resuspended in 200 µl lysis buffer (0.2 M sorbitol, 25 mM KPO$_4$ pH 7, 2 mM EDTA, 0.6% Triton X-100 and 1× Halt proteases inhibitor cocktail), and pure GST-GGA2$^{GAT}$ was added. Binding was done for 30 min on a rotating stand at 4 °C, followed by one wash in 300 µl lysis buffer, and final resuspension in 400 µl lysis buffer.

**Binding and elution.** S100 and P100 fractions of yeast cells were incubated with pre-bound GGA2$^{GAT}$ on beads for 1 h on a rotating wheel at 4 °C. Washes were done three times in 50 µl lysis buffer, and elution was done by adding 30 µl of Laemmli 2× to the beads and 5 min incubation at 95 °C. For each gel, 10 µl per lane was used for the lysis, flow-through and elution samples.

## TEM

Cells were grown to mid-log phase and fixed into the YPD media with 0.2% glutaraldehyde and 3% formaldehyde final concentration overnight at 4 °C. The next day, cells were pelleted and washed three times with 0.1 M HEPES buffer pH 7 and incubated for 30 min in 1% NaJO$_4$ in HEPES buffer, washed three times, and free aldehydes were quenched with 50 mM NH$_4$Cl in HEPES buffer for 30 min. After that, pellets were dehydrated through a series of methanol (50%, 70%, 90%) then infiltrated with LR Gold resin (Polysciences) according to the manufacturer's instructions and allowed to polymerize at −10 °C under an ultraviolet lamp for 24 h. Sections of 60–70 nm were collected on carbon-coated Formvar-Ni grids. Sections were blocked with PBST (PBS + 0.05% Tween20) + 2% bovine serum albumin (BSA) for 15 min and incubated 3 h at room temperature with anti-GFP antibody (1:100, 6556 Abcam) in PBST/BSA. Sections were washed 5 × 5 min with PBS and incubated with goat anti-rabbit secondary antibody (BBI) coupled to 10 nm gold 1:100 in PBST/BSA for 2 h, washed 5 × 5 min with PBS and 3 × 2 min H$_2$O, and stained for 10 min in 2% Ur acetate and 1 min in Pb citrate (Reynold's solution). Sections were viewed with a Philips CM100 electron microscope.

## Mitochondria fusion and fission dynamics

Yeast cells were grown to mid-log phase in YPD media, and switched to 37 °C for 30 min when indicated. Movies were acquired either with an Axio Imager.M2 (*ARF1* 23–37 °C and *arf1-11* at 37 °C) or the FEI-MORE (*arf1-11* at 23 °C), over a period of 2 min with sequential image acquisition. Images from the FEI-MORE were further deconvolved in standard mode using the Huygens Pro software. Movies were assembled with the Fiji software and single images prepared on the OMERO.insight client.

To analyse mitochondria dynamics in WT and *ARF1* KO HeLa cells parental and *ARF1* KO cells were seeded on a six-well plate. For rescuing the absence of Arf1 phenotype, we transfected the KO cells with 1 µg Arf1-EGFP or Arf1-11-EGFP constructs using Helix IN transfection reagent. After overnight incubation, cells were reseeded on Ibidi four-well imaging chambers and imaged the next day. Just before imaging, mitochondria were stained with 250 nM MitoTracker Deep Red for 30 min following the provider's instructions. Staining solution was replaced with imaging medium containing OxyFluor anti-phototoxicity agent.

Cells were imaged with Olympus Spinning Disk confocal SpinSR microscope equipped with Yokogawa CSU-W1 confocal scan head using UPL Apo 100×/1.5 oil immersion objective. To avoid phototoxicity, we used 640 nm laser with 1% laser power. During aquisition, seven *Z*-stacks with 120 nm optical steps were taken for 2 min in every 3 s. Time-lapse movies were deconvolved with Huygens Remote Manager using Good's Roughness Maximum Likelihood Estimation, 50 iterations, quality control criteria 0.0002. Fusion and fission events were quantified in 10.01 × 10.01-sized region of interest in each cell for 1 min using FIJI software.

## Fluorescent FA pulse-chase

Yeast cells were grown to mid-log phase in HC complete medium at 23 °C, then switched to 37 °C for 30 min. Following this, $1.5 \times 10^6$ cells were incubated with 50 µM BODIPY 558/568 C12 (Invitrogen) for 30 min at 37 °C, washed twice in 1× PBS, 7.5 µM BSA and resuspended in 20 µl 1× PBS, 7.5 µM BSA supplemented with 20 µl 0.4% Trypan blue in 1× PBS. Images were taken with the FEI-MORE microscope, and deconvolved in standard mode using the Huygens Pro software.

FA pulse-chase experiments in HeLa cells were performed according to Rambold et al.[14]. In short, transfected cells were plated into eight-well imaging chambers (Miltenyi) and incubated for at least 6 h

to let the cells adhere. Then their CM was replaced with CM containing 1 µM BODIPY 558/568 C12 (Invitrogen) and incubated overnight. Cells were washed three times with CM, incubated for 1 h in CM and then chased for 9 h in CM or in HBSS. Mitochondria were labelled with 200 nM MitoView Fix 640 (Biotinum) for 20 min before imaging. Just before imaging, CM or HBSS was replaced with imaging buffer containing 25 mM dextrose supplemented with 10% FBS or 5 mM dextrose, supplemented with 0.2% FBS, respectively.

Cells were imaged at 37 °C using an inverted Axio Observer microscope (Zeiss) with a Plan-Apochromat N 63×/1.40 oil DIC M27 objective and a Photometrics Prime 95B camera. Filters with standard specifications for GFP, dsRed and Cy5 were used to image EGFP, Bodipy 558/568 and Mitoview 640, respectively.

## LD staining
Neutral lipids of mid-log grown cells in YPD media were stained with 1 µl of the lipophilic fluorophore ReadiStain Lipid Green (InVivo Biosystems) for 30 min at 23 °C and 37 °C. Cells were then washed three times in HC complete medium and prepared for microscopy.

Transfected HeLa cells were plated into eight-well imaging chambers (Ibidi, ibiTreat µ-Slide) the day before imaging to reach 50–70% confluency the following day. Just before imaging, cells were rinsed with pre-warmed PBS and replaced with imaging buffer (4.5 g l$^{-1}$ dextrose, 1 mM CaCl$_2$, 2.7 mM KCl and 0.5 mM MgCl$_2$ in PBS supplemented with 0.2% FBS) containing 400 ng ml$^{-1}$ Nile Red (Sigma) and incubated for 10 min before starting imaging. The dye was present during imaging. Confocal images were acquired at 37 °C with Olympus Fluoview FV3000 system, using an UPLSAPO 60×/1.30 objective with silicone oil, resulting in an $xy$ pixel size of 0.1 µm. Laser intensities were at 0.5–3% for both 488 (GFP) and 561 (DsRed) wavelengths. Sampling speed was 8.0 µs per pixel with a zoom factor of 2.0. All images for corresponding experiments were processed with the same settings to ensure comparable results.

## Total RNA isolation and qRT–PCR
Total RNA were extracted from 25 ml of mid-log grown cultures. Cell pellets were mixed with 300 µl AE buffer (50 mM NaOAC and 10 mM EDTA, pH 6) and 50 µl of 20% SDS. Then 300 µl of phenol–chloroform isoamyl alcohol pH 4.5 was added to the mixture, and tubes were vortexed for 15 s. Then samples were incubated at 65 °C for 5 min and flash-frozen in liquid nitrogen. Samples were centrifuged for 10 min at 20,000$g$ at 4 °C. The upper aqueous phase was recovered and put in a new low-binding tube with 200 µl phenol–chloroform isoamyl alcohol. Tubes were again vortexed for 10 s, centrifuged for 10 min at 20,000$g$ at 4 °C and the aqueous layer recovered. To that phase, 20 µl 3 M sodium acetate and 600 µl ethanol was added, and the mixture was chilled at −80 °C for 2 h. Samples were then centrifuged 20,000$g$ for 30 min at 4 °C, and RNA pellets washed with 75% ethanol. Air-dried RNAs were dissolved in 50 µl water.

Four micrograms of total RNA was subjected to reverse transcription (200 ng µl$^{-1}$ final RNA concentration) with the MegaScript reverse transcription mix using random (0.5 µg) and oligo-dT primers (0.5 µg) (Promega). Reactions were then incubated 5 min at 25 °C, followed by 1 h at 62 °C and 15 min at 75 °C. Using 100 ng of complementary DNA, qPCR was carried out using the GoTaq master mix. Oligonuceotides sequences are available as Supplementary Table 3.

## Lipid extraction from yeast cell pellets
Lipid extraction and analysis were performed in triplicates. Frozen cell pellets (5 × 10$^8$ cells) were taken from −80 °C and 500 µl glass beads were added, then 20 µl of yeast internal standard mix and 20 µl cholesterol were added (see below), followed by 600 µl H$_2$O, 1.5 ml methanol, then vortexed for 1 min. Then 0.75 ml chloroform was added, and the samples were vortexed at high speed for 6 min. The solution was transferred to a 13 × 100 mm screw-capped glass tube, and the beads were

washed with 0.6 ml chloroform:methanol (1:2). Following vortexing, the solution was combined with the first extraction. Then 0.4 ml H$_2$O was added and mixed by vortexing. Samples were centrifuged for 10 min at 4,000 r.p.m. (3,220$g$) and most of the aqueous phase was removed, followed by transfer of the organic phase to fresh 13 × 100 tubes without taking interface. Then 0.4 ml of artificial upper phase (chloroform:methanol:water (3:48:47, v/v/v)) was added and mixed by vortexing. After centrifugation for 10 min at 4,000 r.p.m. and removal of most of the aqueous phase, the organic phase was transferred to mass spectrometry (MS) vials. Then 0.3–0.4 ml was transferred to a vial insert (filled up) for analysis using the Thermo Q-Exactive Plus. The extracts in inserts were dried in the Centrivap (Labconco) and the extracts in vials under flow of N$_2$. Internal standards used were PC31:1 (7.5 nmol per sample), PE31:1 (7.5 nmol per sample), PI31:1 (6 nmol per sample), PS31:1 (4 nmol per sample), CL56:0 (4 nmol per sample), C17Cer (1.2 nmol per sample), C8GC (2 nmol per sample) and cholesterol (20 nmol per sample).

## Sterol analysis by gas chromatography–MS
Lipid extracts from approximately 40% of the total sample was dissolved in 0.3 ml chloroform:methanol (1:1), and 5 µl was loaded onto a gas chromatograph–mass spectrometer (Varian 320 MS) fitted with a fused silica capillary column (15 m × 0.32 mm inner diameter, Macherey Nagel ref. 726206.15). The gas chromatography program started at 45 °C for 4 min, then a gradient to 195 °C at 20 °C min$^{-1}$, to 230 °C at 4 °C min$^{-1}$, to 320 °C at 10 °C min$^{-1}$, to 350 °C at 6 °C min$^{-1}$ followed by a return to 45 °C at 100 °C min$^{-1}$. Data were collected in the centroid mode. The different sterol species were identified by their retention times and mass spectra using the NIST database and ergosterol and cholesterol standards. Peaks were integrated and amounts calculated with correction for the yield of the internal standard, cholesterol, and using standard curves of cholesterol and ergosterol concentrations.

## Lipid analysis by electrospray MS using the TSQ Vantage
Lipid analysis using nanoflow infusion (Advion Nanomate) and multiple reaction monitoring (TSQ Vantage, Thermo) was performed as described[75]. All runs were performed in duplicates with all transitions measured three times for a total of at least six measurements for each lipid species.

## TAG analysis using the UHPLC–MS using the Q-Exactive Plus
Dried samples were resuspended by sonicating in 100 µL of liquid chromatography–MS-grade chloroform:methanol (1:1, v/v). Reversed-phase ultrahigh-performance liquid chromatography (UHPLC)–high-resolution MS analyses were performed using a Q-Exactive Plus Hybrid Quadrupole-Orbitrap mass spectrometer coupled to an UltiMate 3000 UHPLC system (Thermo Fisher Scientific) equipped with an Accucore C30 column (150 × 2.1 mm, 2.6 µm) and its 20 mm guard (Thermo Fisher Scientific). Samples were kept at 8 °C in the autosampler, 10 µl was injected and eluted with a gradient starting at 10% B for 1 min, 10–70% B in 4 min, 70–100% B in 10 min, washed in 100% B for 5 min and column equilibration for an additional 3 min. Eluents were made of 5 mM ammonium acetate and 0.1% formic acid in water (solvent A) or in isopropanol/acetonitrile (2:1, v/v) (solvent B). Flow rate and column oven temperature were respectively at 350 µl min$^{-1}$ and 40 °C. The mass spectrometer was operated using a heated electrospray-ionization source in positive polarity with the following settings: electrospray voltage: 3.9 kV (+); sheath gas: 51; auxiliary gas: 13; sweep gas: 3; vapourizer temperature: 431 °C; ion transfer capillary temperature: 320 °C; S-lens: 50; resolution: 140,000; $m/z$ range: 200–1,000; automatic gain control: 1 × 10$^6$; maximum injection time: 50 ms. The following setting was used in Higher-energy C-trap dissociation fragmentation: automatic gain control: 2.5 × 10$^5$; maximum injection time: 120 ms; resolution: 35,000;

(N)CE: 30. Xcaliburv.4.2 (Thermo Fisher Scientific) was used for data acquisition and processing. Major TAG species were quantified after removing background and normalization, and was presented as ratio to sample no. 1, respectively.

## TAG analysis by thin liquid chromatography

*ARF1* and *arf1-11* strains were grown overnight at 23 °C in YPUAD media, and diluted to 0.2 optical density (OD) in the morning. When cells reached OD 0.7, cells were treated either with cerulenin (10 µg ml$^{-1}$) or with the corresponding volume of DMSO. After 30 min of incubation at 23 °C, $1.5 \times 10^8$ cells were collected. Then cells were shifted to 37 °C and $1.5 \times 10^8$ cells were collected after 30 and 60 min of incubation. Lipids were extracted as described above. Dried lipid extracts were thawed and resuspended in 20 µl chloroform. Triolein (18:1 TG; Avanti Polar Lipids) was used as an internal marker. Samples were then spotted onto thin-liquid chromatography silica plates (aluminium sheets gel 60; Sigma-Aldrich), and neutral lipids separated by running in two successive chambers containing petroleum ether/diethyl ether (1:1, v/v) and petroleum ether/diethyl ether (49:1, v/v). For detection, plates were dipped into a solution containing $MnCl_2$, methanol and sulfuric acid (0.63 g $MnCl_2 \cdot 4H_2O$, 60 ml water, 60 ml methanol, 4 ml concentrated sulfuric acid) for 1 min and heated at 110 °C for 3 min. Plates were then scanned and analysed using the Fiji software.

## ATP measurements

**FRET measurement.** ATP levels were measured in single cells grown to mid-log phase using a FRET-based nanosensor expressed from a cen/ars plasmid, pDR-GW AT1.03YEMK, (Addgene no. 28004). Images were acquired with the Axio Imager.M2, using dedicated CFP, YFP and FRET CFP-YFP filters. Regions of interest were measured with the Fiji software.

**Biochemical assay.** ATP quantitation was performed using the BacTiter-Glo Cell viability assay (Promega). Cells were resuspended in TE buffer pH 7 containing 0.7 M sorbitol and $10^6$ cells were used for each assay. Luminescence was measured right after mixing cells with BacTiter-Glo buffer in 96× flat-white plates (Greiner) on a Tecan Infinite M1000Pro with 10 s orbital shaking (2 mm wide, 350 r.p.m.).

## Mitochondrial activity measurements

Mitochondria were isolated from WT and mutant strains grown for five to six generations in 2 litres of YPGly at 23 °C or 28 °C by enzymatic method[76]. The cultures contained 2–5% of ρ$^-$/ρ$^o$ cells. The values reported are averages of two biological repetitions. Respiratory, ATP synthesis activities and the variations of inner membrane potential were measured using freshly isolated, osmotically protected mitochondria buffered at pH 6.8. The oxygen consumption was measured in an oxygraph with Clarck electrode (Heito, France) at 28 °C in thermo-stabilized cuvette. Reaction mixes for assays contained 0.15 mg ml$^{-1}$ of mitochondria, 4 mM NADH, 150 µM ADP, 12.5 mM ascorbate, 1.4 mM *N,N,N,N*,-tetramethyl-*p*-phenylenediamine and 4 µM carbonyl cyanide m-chlorophenyl hydrazone (CCCP). The rates of ATP synthesis were determined under the same experimental conditions in the presence of 750 µM ADP; aliquots were withdrawn from the oxygraph cuvette every 15 s and the reaction was stopped with 3.5% (w/v) perchloric acid, 12.5 mM EDTA. The ATP in samples was quantified using the Kinase-Glo Max Luminescence Kinase Assay (Promega) and a Beckman Coulter's Paradigm Plate Reader. Variations in transmembrane potential ($\Delta\psi$) were evaluated in the respiration buffer containing 0.150 mg ml$^{-1}$ of mitochondria and the Rhodamine 123 (0.5 µg ml$^{-1}$), with $\lambda_{exc}$ of 485 nm and $\lambda_{em}$ of 533 nm under constant stirring using a Cary Eclipse Fluorescence Spectrophotometer (Agilent Technologies)[77]. For the ATPase assays, mitochondria kept at −80 °C were thawed and the reaction was performed in absence of osmotic protection and at pH 8.4 (ref. 78).

## Statistics and reproducibility

All experiments were performed at least in three independent biological replicates. Unpaired two-tailed *t*-test or two-way analysis of variance (ANOVA) were calculated for each experiment using GraphPad Prism9. For image analysis, pictures were taken at random places on coverslips, analysed in a blinded manner whenever possible with Omero webclient. All images are representative images from at least three independent biological experiments. For western blot, loading controls were run on a separate gel when the protein of interest had a similar molecular weight.

## Reporting summary

Further information on research design is available in the Nature Portfolio Reporting Summary linked to this article.

## Data availability

The authors declare that the main data supporting the findings of this study are available within the article and its supplementary information files. Source data are provided with this paper. All other data supporting the findings of this study are available from the corresponding author upon request.

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

## Acknowledgements

We thank A. Gil and H. Wechlin, S. Begum and S. Feng for excellent technical assistance with some of the experiments and F. Wieland (BZH, Heidelberg) for β-COP antibodies. This work was supported by grants of the Swiss National Science Foundation (310030B_163480 and 310030_185127) to A.S. and the University of Basel, Swiss National Science Foundation, NCCR Chemical Biology and the Leducq Foundation to H.R., Swiss National Science Foundation (31003A-182519) to M.S., and the National Science Center of Poland (2018-31-B-NZ3-01117) to R.K.

## Author contributions

L.E. and A.S. conceived the project. L.E. performed all yeast cell biology experiments, except for the Pex11–scarlet/Arf1–GFP imaging, which was performed by R.E.A. and E.Z. V.S. and M.P. performed the experiments in mammalian cells. I.R. and H.R. did the lipidomics analyses. A.W. and R.K. carried out the biochemical analyses on yeast mitochondria. The EM experiments were executed by C.P.-B. Data were analysed by L.E., V.S., R.K., H.R., M.S. and A.S. L.E. and A.S. wrote the manuscript with input from all authors.

## Competing interests

The authors declare no competing interests.

## Additional information

**Extended data** is available for this paper at https://doi.org/10.1038/s41556-023-01180-2.

**Correspondence and requests for materials** should be addressed to Anne Spang.

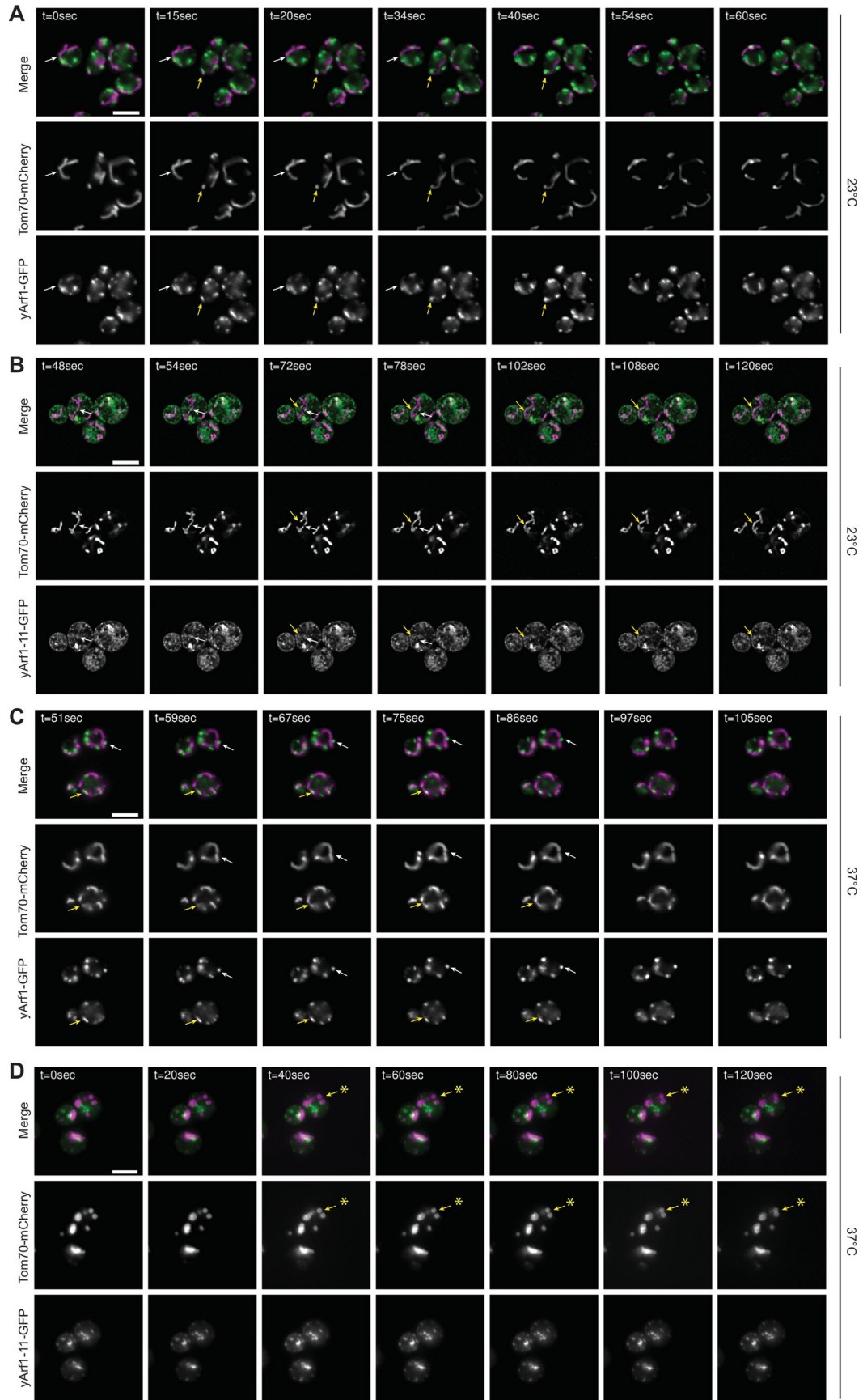

**Extended Data Fig. 1 | yArf1 regulates mitochondria fusion and fission.**
(**A**–**D**) Single time-point images of movies done with strains expressing yArf1-GFP
(**A**) or yArf1-11-GFP (**B**) and the mitochondrial protein Tom70-mCherry at 23 °C,
or shifted 30 min at 37 °C for yArf1-GFP (**C**) and yArf1-11-GFP (**D**). Merge and
individual GFP and mCherry channels are shown. Arrows indicate mitochondrial
fusion (yellow) and fission (white). Asterisk indicates mitochondrial fusion
independent of yArf1-11. Scale bars: 5 µm.

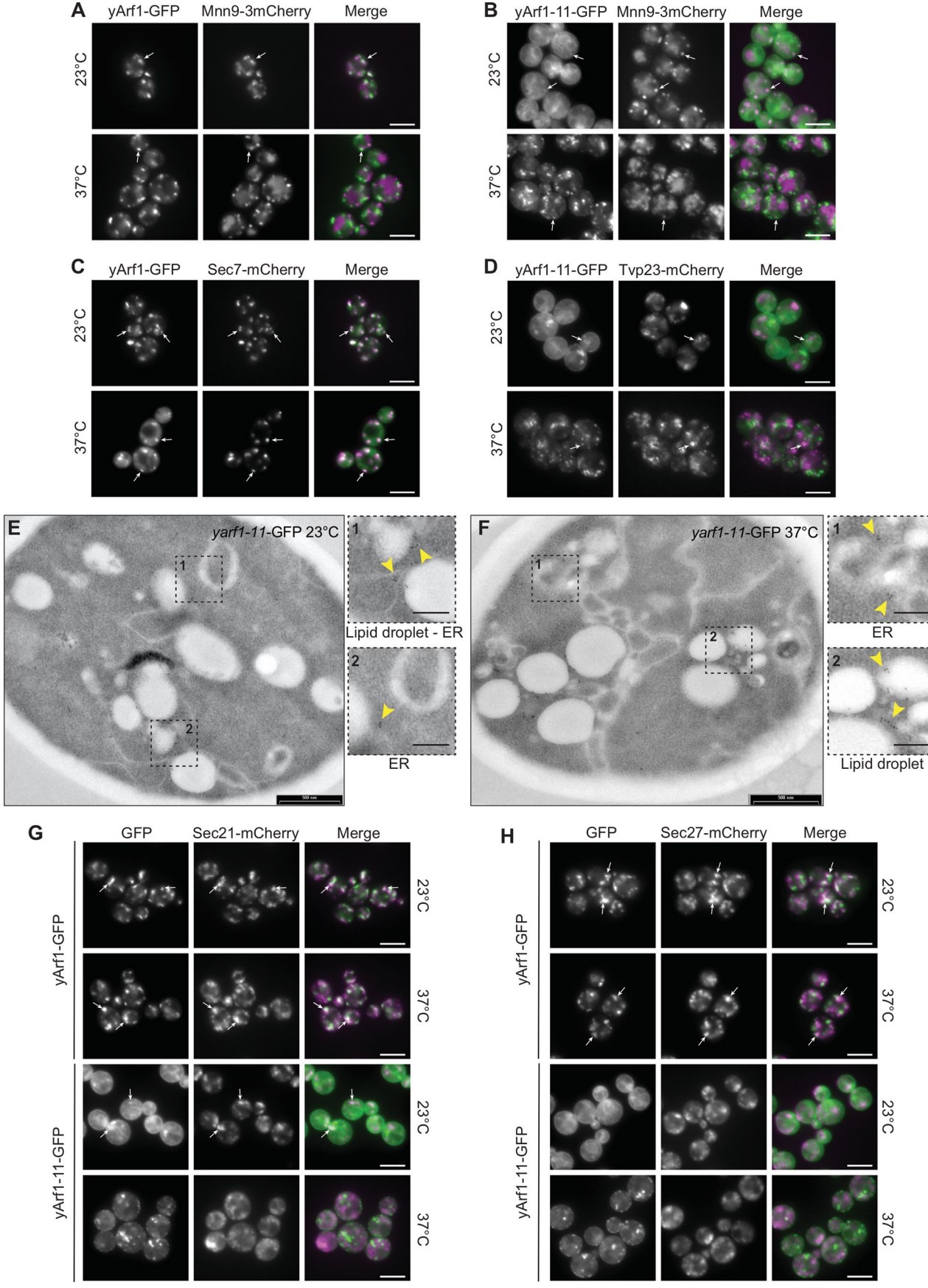

**Extended Data Fig. 2 | See next page for caption.**

**Extended Data Fig. 2 | yArf1-11 is a hyperactive mutant present on the ER and LDs.** (**A**, **B**) Co-localization of yArf1-GFP (**A**) and yArf1-11-GFP (**B**) with the *cis*-Golgi marker Mnn9-mCherry grown at 23 °C or shifted to 37 °C. Arrows indicate sites of co-localization between yArf1/yArf1-11 and Mnn9. Scale bar: 5 µm. (**C**, **D**) Co-localization of yArf1-GFP (**C**) and yArf1-11-GFP (**D**) with the *trans*-Golgi marker Sec7-mCherry and Tvp23-mCherry grown at 23 °C or shifted to 37 °C. Arrows indicate sites of co-localization between yArf1/yArf1-11 and the corresponding markers. (**E**, **F**) Transmission electron microscopy of the yArf1-11-GFP strain grown either at 23 °C (**E**) or shifted to 37 °C for 30 min (**F**). yArf1-11-GFP localizations were highlighted by immunogold labelling, and dotted squares shows enlargements of yArf-11 specific localizations (arrows). Scale bar: 500 nm, scale bar magnification: 200 nm. (**G**, **H**) Co-localization of yArf1-GFP and yArf1-11-GFP with the COPI markers Sec21-mCherry (**G**) and Sec27-mCherry (**H**) grown at 23 °C or shifted to 37 °C. Arrows indicate sites of co-localization between the yArf1 or yArf1-11 and COPI components. Scale bar: 5 µm.

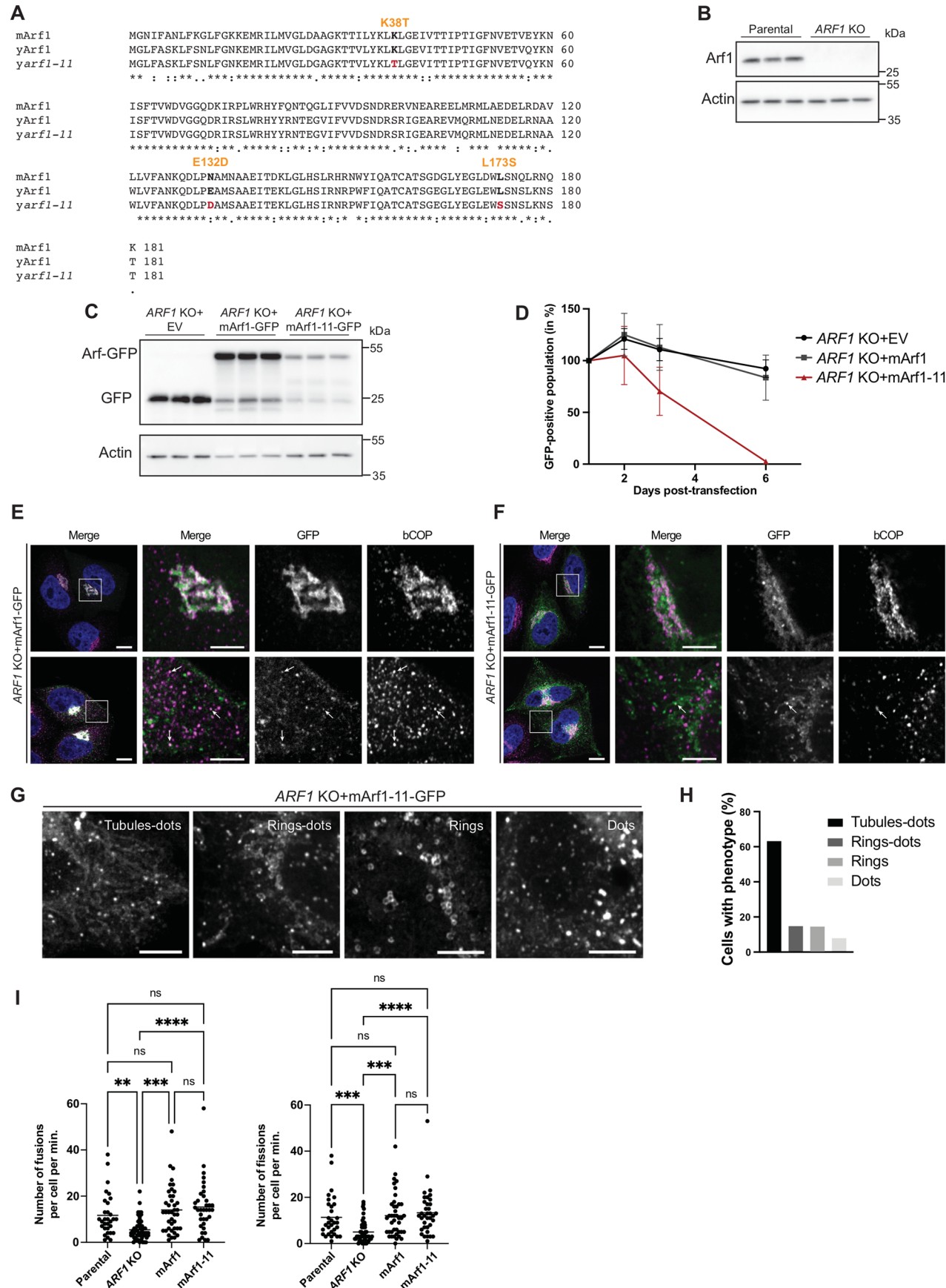

**Extended Data Fig. 3 | See next page for caption.**

**Extended Data Fig. 3 | Functional conservation of Arf1-11 in mammalian cells.**
(**A**) Alignment of mammalian Arf1 (mArf1), yeast Arf1 (yArf1) and the yeast *arf1-11* (yArf1-11) amino acids sequences. Arf1-11 mutation's and their corresponding amino acids in mArf1 are shown. (**B**) Immunoblot analysis of Arf1 presence in parental HeLa cells and the CRISPR-Cas9 ARF1 KO cells. For each cell line, three independent biological replicates were analyzed on the same gel. (**C**) Immunoblot analysis of Arf1-GFP presence in *ARF1* KO HeLa cells transfected either with an empty vector (EV), with mArf1-GFP or with mArf-11-GFP. For each cell line, three independent biological replicates were analyzed on the same gel. Actin was used as internal control. (**D**) Cell viability assay as percent of GFP-positive cells in the total population after transfection of EV, mArf1-GFP or mArf-11-GFP. Mean and standard deviation are shown from n = 3 biological replicates. (**E, F**) Co-localization of mArf1- (**E**) and mArf1-11-GFP (**F**) expressed in the *ARF1* KO cell line with COPI vesicles done by immunostaining against the coatomer subunit beta (bCOP). Squares show magnification of a perinuclear and distal portion of the cell. Scale bars: 10 µm and 5 µm (inlays). Images were acquired 24 h after transfection. (**G**) Different localizations of mArf1-11 observed in *ARF1* KO HeLa cell line expressing mArf1-11-GFP. (**H**) From the images in (**G**), four phenotypes were identified and their occurrences quantified. 380 cells were quantified. (**I**) Mitochondrial fusion and fission events measured in Parental cell lines, *ARF1 KO*, *ARF1 KO* expressing mArf1- or mArf1-11-GFP. Events were scored 48 h after transfection. Mean and standard deviation are shown, Parental = 31 cells, *ARF1 KO* = 53 cells, *ARF1 KO*+mArf1 = 41 cells, *ARF1 KO*+mArf1-11 = 38 cells from n = 3 biological replicates; Statistical analysis were done using a two-way ANOVA test with Turkey's multiple comparison test, ****$p$ = 0.000000000048; ***$p$ = 0.001; **$p$ = 0.0047. Source numerical data and unprocessed blots are available in source data. See also Supplementary data 1.

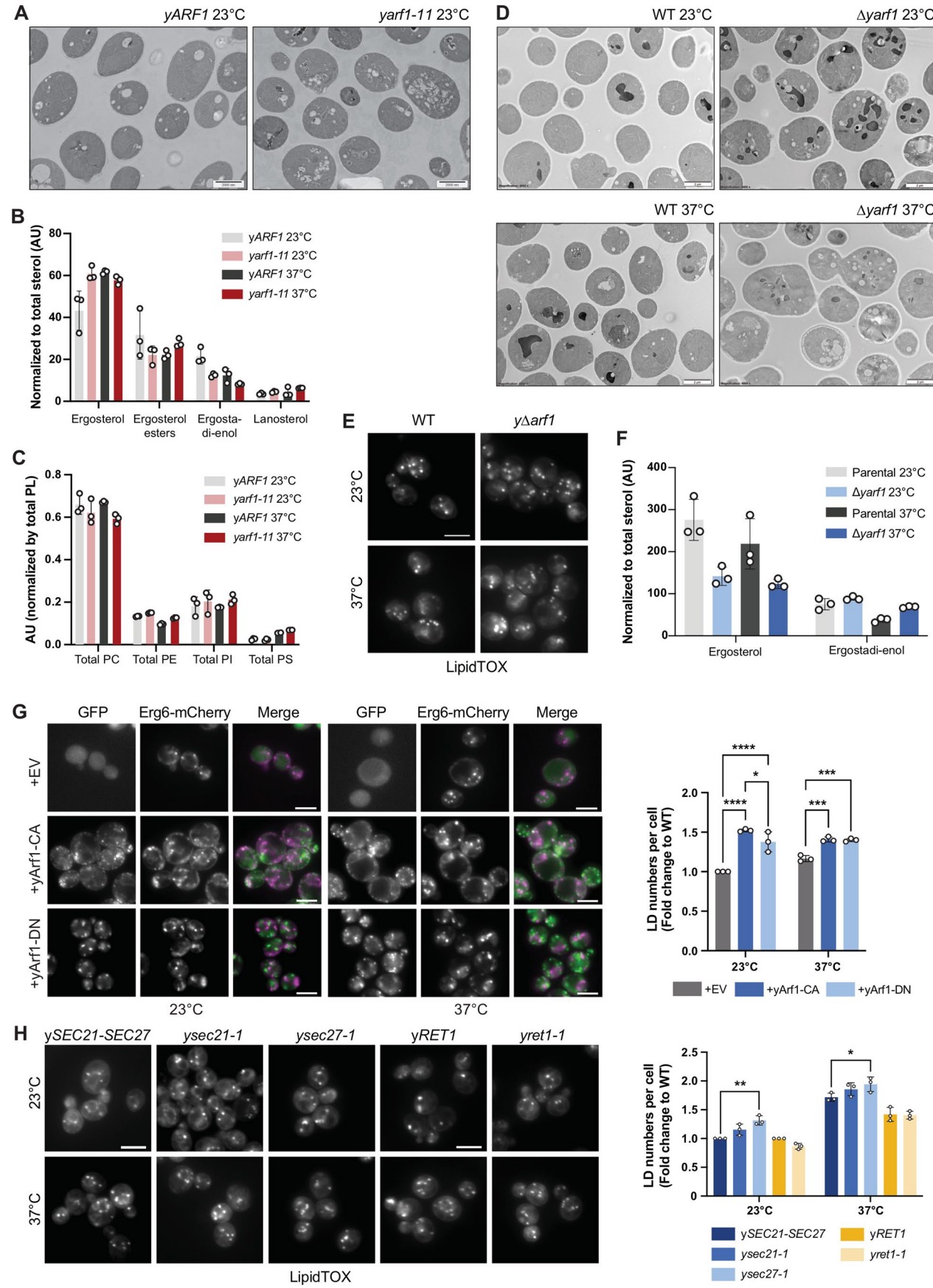

**Extended Data Fig. 4 | See next page for caption.**

**Extended Data Fig. 4 | Hyperactive Arf1 induces triacylglycerol accumulation.** (**A**) Transmission electron microscopy of y*ARF1* and y*arf1-11* strains grown at 23 °C. Scale bars: 2000 nm. (**B, C**) Measurements of sterols (**B**) and phospholipids (**C**) in y*ARF1* and y*arf1-11* strains grown at 23 °C or shifted to 37 °C. PC: phosphatidylcholine, PE: phosphatidylethanolamine, PI: phosphatidylinositol, PS: phosphatidylserine. Mean and standard deviation are shown from n = 3 biological replicates. (**D**) Transmission electron microscopy of WT and Δy*arf1* strains grown at 23 °C or shifted to 37 °C. Scale bars: 2000 nm. (**E**) LipidTox staining of LDs in WT and Δy*arf1* strains grown at 23 °C or shifted to 37 °C. Scale bar: 5 μm. (**F**) Measurements of sterols in WT and Δy*arf1* strains grown at 23 °C or shifted to 37 °C. Mean and standard deviation are shown from n = 3 biological replicates. (**G**) Measurements of LD numbers in WT cells expressing constitutively active yArf1-CA- or dominant negative yArf1-DN-GFP using Erg6-mCherry as marker. Mean and standard deviation are shown; WT + EV 23 °C = 303 cells, WT+yArf1-CA 23 °C = 298 cells, WT+yArf1-DN 23 °C = 289 cells, WT + EV 37 °C = 291 cells, WT+yArf1-CA 37 °C = 316 cells, WT+yArf1-DN 37 °C = 300 cells from n = 3 biological replicates. Statistical analysis were done using a two-way ANOVA test with Turkey's multiple comparison test, +EV vs +CA 23 °C ****$p$ = 0.000000271; +EV vs +DN 23 °C ****$p$ = 0.00000871; ***$p$ = 0.0006; *$p$ = 0.022 from n = 3 biological replicates. Scale bar: 5 μm. (**H**) Measurements of LD numbers in COPI *ts*-mutants *sec21-1, sec27-1, ret1-1* and their corresponding parental strains by LipidTox. Mean and standard deviation are shown; y*SEC21-27* 23 °C = 559 cells, y*sec21-1* 23 °C = 563 cells, y*sec27-1* 23 °C = 560 cells, y*SEC21-27* 37 °C = 557 cells, y*sec21-1* 37 °C = 527 cells, y*sec27-1* 37 °C = 572 cells, y*RET1* 23 °C = 377 cells, y*ret1-1* 23 °C = 553, y*RET1* 37 °C = 427 cells, y*ret1-1* 37 °C = 486 cells; Statistical analysis were done using a two-way ANOVA test with Turkey's multiple comparison test, **$p$ = 0.0034; *$p$ = 0.0215 from n = 3 biological replicates. Scale bar: 5 μm. Source numerical data are available in source data.

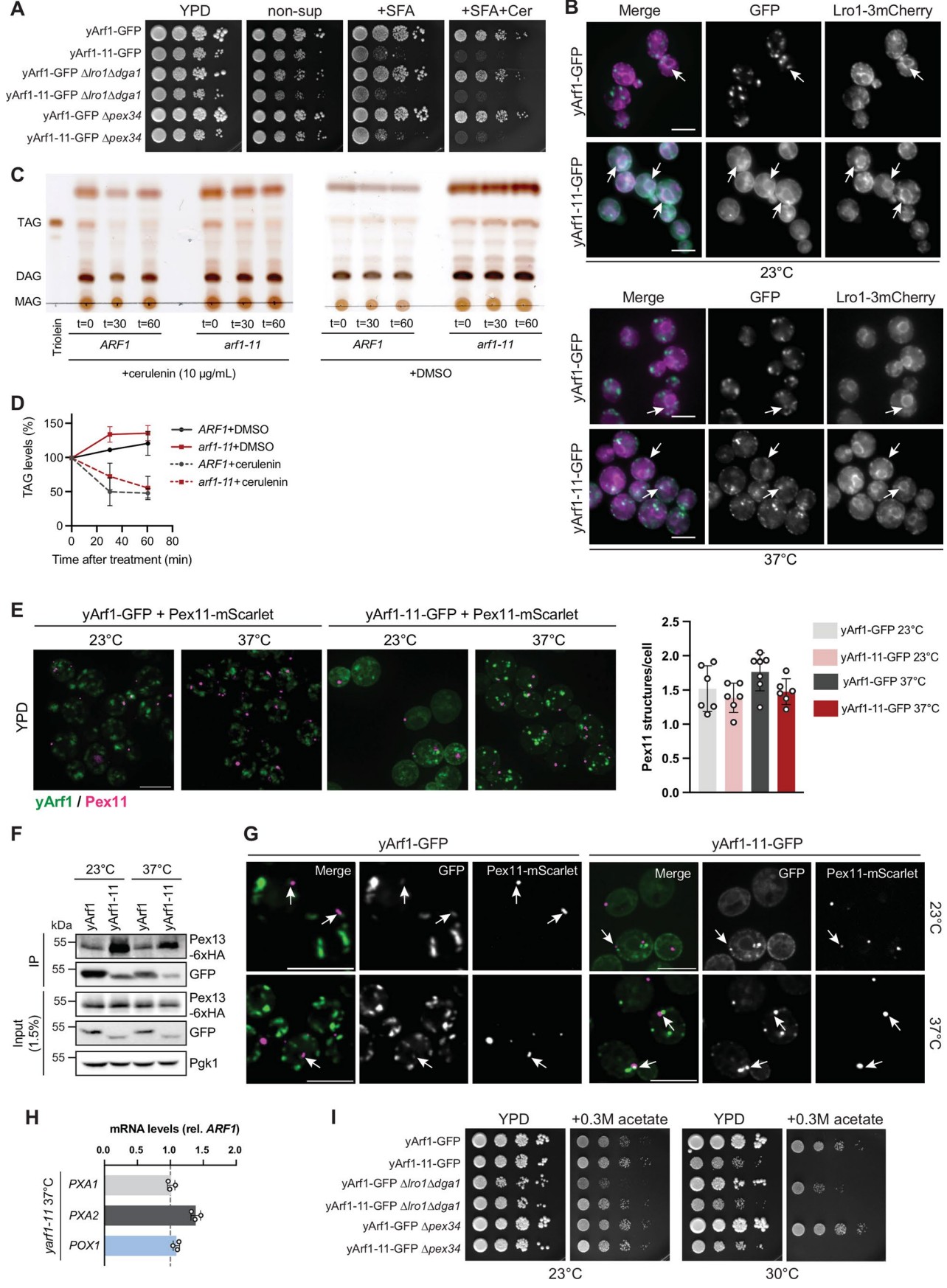

**Extended Data Fig. 5 | See next page for caption.**

**Extended Data Fig. 5 | yArf1 regulates LD-associated functions and ß-oxidation.** (**A**) Growth test of y*ARF1* and y*arf1-11* strains, deprived of the *LRO1/ DGA1* (Δ*lro1* Δ*dga1*) or of *PEX34* (Δ*pex34*), on rich media (YPD), rich media lacking saturated fatty acids (SFA) and containing cerulenin (Cer) (non-sup), containing SFA or SFA and cerulenin at 23 °C. (**B**) Co-localization of yArf1-GFP and yArf1-11-GFP with Lro1-3xmCherry grown at 23 °C or shifted to 37 °C for 30 min. Arrows indicate yArf1/yArf1-11 co-localizing or juxtaposed to Lro1. Scale bar: 5 μm. (**C**, **D**) *In vivo* TAG mobilization assay performed on y*ARF1* and y*arf1-11* strains. Cellular TAG levels evidenced by thin-liquid chromatography extracted from cells grown in the presence of cerulenin or DMSO (**C**) were measured (**D**). Mean and standard deviation from at least n = 3 biological replicates are shown. (**E**) Peroxisome numbers evaluated by microscopy using Pex11-mScarlet in the y*ARF1*- and y*arf1-11-GFP* strains grown at 23 °C or shifted to 37 °C. Peroxisome numbers per cell were quantified in each strain and conditions. Mean and

standard deviation are shown; yArf1+Pex11-mScarlet 23 °C = 289 cells, yArf1+Pex11-mScarlet 37 °C = 431 cells, yArf1-11+Pex11-mScarlet 23 °C = 200 cells, yArf1-11+Pex11-mScarlet 37 °C = 304 cells from n = 3 biological replicates. Scale bar: 5 μm. (**F**) Co-immunoprecipitation of yArf1-GFP and yArf1-11-GFP with Pex13-6xHA. Strains were grown at 23 °C or shifted to 37 °C. (**G**) High-resolution microscopy of y*ARF1*- and y*arf1-11*-GFP strains expressing Pex11-mScarlet grown at 23 °C or shifted to 37 °C. Arrows indicate Arf1 partially co-localizing with, or juxtaposed to peroxisomes. Scale bars: 5 μm. (**H**) mRNA levels measurement of *POX1*, *PXA1* and *PXA2* in y*arf1-11* compared to y*ARF1* grown at 37 °C by qRT-PCR. Fold changes were measured by the $2^{-\Delta\Delta Ct}$ method. Mean and standard deviation from n = 3 biological replicates are shown. (**I**) Growth test of y*ARF1*, y*arf1-11*, Δ*lro1* Δ*dga1* or Δ*pex34* strains, on YPD or media supplemented with 0.3 M sodium acetate at 23 °C and 30 °C. Source numerical data and unprocessed blots are available in source data.

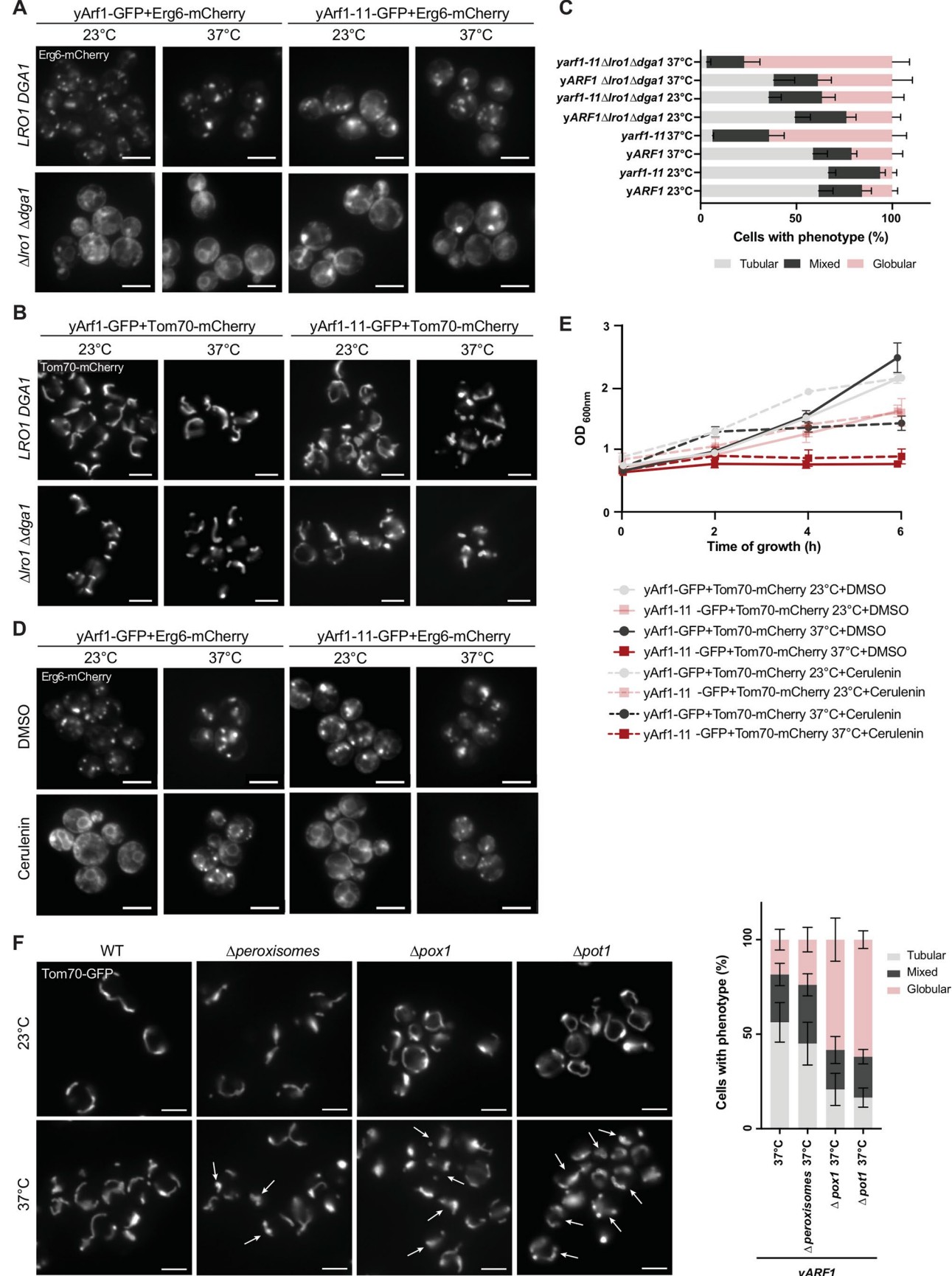

**Extended Data Fig. 6 | See next page for caption.**

**Extended Data Fig. 6 | Acetyl-CoA transfer loss leads to mitochondria fragmentation.** (**A**) Lipid droplets morphologies and Erg6 localization imaged in the y*ARF1* and y*arf1-11* parental strains (*LRO1 DGA1*) or in strains deprived of *LRO1 DGA1* (Δ*lro1*Δ*dga1*) grown at 23 °C or shifted to 37 °C. Scale bar: 5 μm.
(**B**) Mitochondrial morphologies imaged in the parental or Δ*lro1*Δ*dga1* y*ARF1* and y*arf1-11* strains grown at 23 °C or shifted to 37 °C. Scale bar: 5 μm.
(**C**) Quantification of the mitochondrial phenotypes observed in (**B**). Mean and standard deviation are shown; y*ARF1* 23 °C = 1213 cells, y*arf1-11* 23 °C = 948 cells, y*ARF1* Δ*lro1*Δ*dga1* 23 °C = 804 cells, y*arf1-11* Δ*lro1*Δ*dga1* 23 °C = 1832 cells, y*ARF1* 37 °C = 937 cells, y*arf1-11* 37 °C = 732 cells, y*ARF1* Δ*lro1*Δ*dga1* 37 °C = 640 cells, y*arf1-11* Δ*lro1*Δ*dga1* 37 °C = 988 cells from n = 4 biological replicates.
(**D**) Lipid droplets morphologies and Erg6 localization imaged in the y*ARF1* and y*arf1-11* strains grown at 23 °C or shifted to 37 °C and treated with either DMSO

or the fatty acid synthesis inhibitor cerulenin for 6 h. Scale bar: 5 μm. (**E**) Cell growth monitoring of y*ARF1* and y*arf1-11*-GFP strains treated with DMSO or with cerulenin at 23 °C and shifted to 37 °C as seen in (**D**). Mean and standard deviation from n = 3 biological replicates are shown. (**F**) Mitochondrial morphologies were imaged in the parental WT cells in which peroxisomal biogenesis was abolished *(Δpex3 Δpex19)*, or cells lacking enzymes of the ß-oxidation Δ*pox1* or Δ*pot1*. All strains were grown at 23 °C or shifted to 37 °C, and Tom70-GFP was used as a mitochondrial marker. Mitochondrial phenotypes observed were quantified and mean and standard deviation are shown. y*ARF1* 37 °C = 166 cells, y*ARF1* Δ*peroxisomes* 37 °C = 180 cells, y*ARF1* Δ*pox1* 37 °C = 574 cells, y*ARF1* Δ*pot1* 37 °C = 539 cells from n = 3 biological replicates. Scale bar: 5 μm. Source numerical data are available in source data.

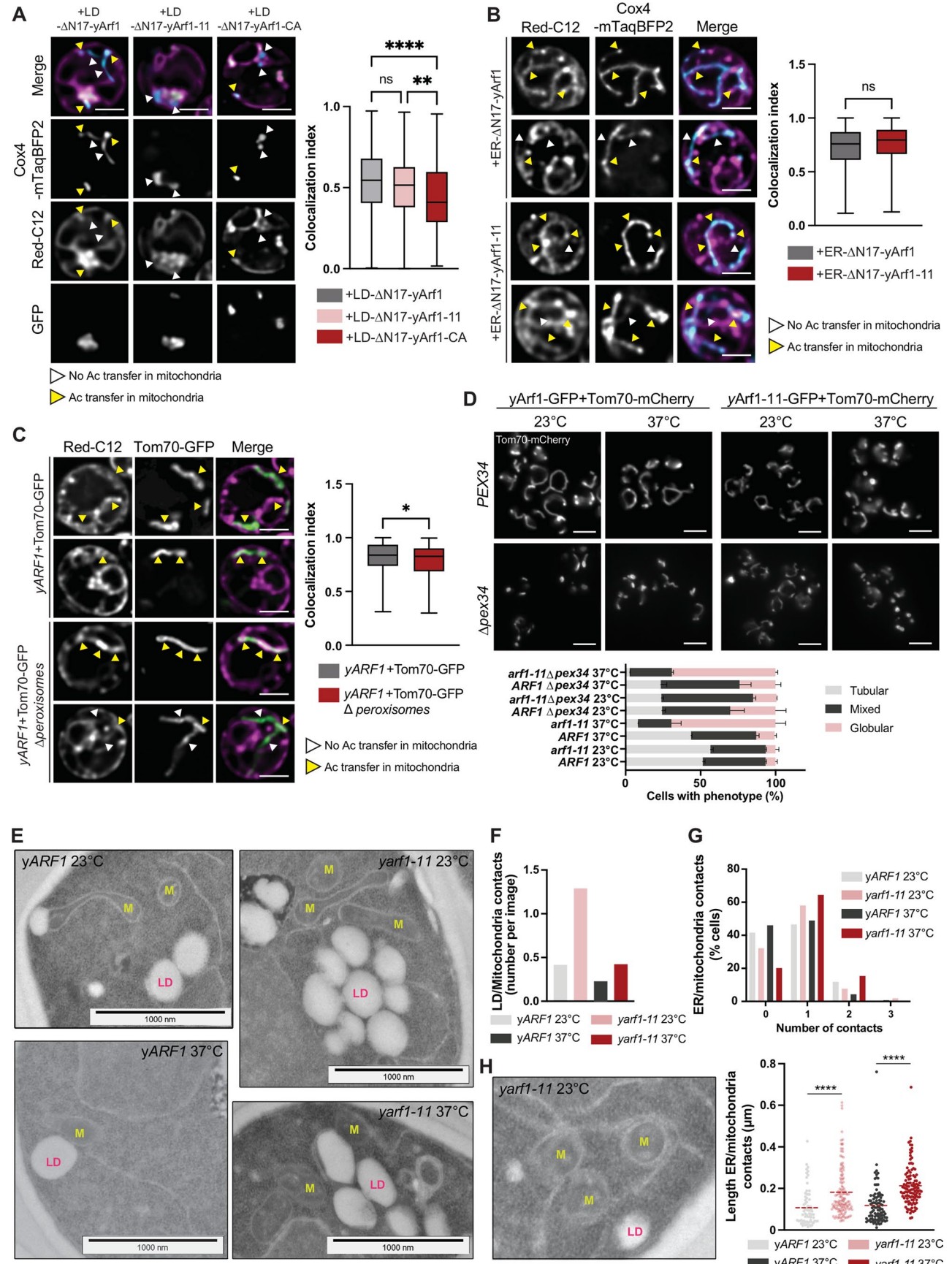

**Extended Data Fig. 7 | See next page for caption.**

**Extended Data Fig. 7 | Acetyl-CoA transfer is modulated by yArf1 localization.** (**A**, **B**) Red-C12 transfer to mitochondria in yΔarf1 + LD-anchored (**A**) or ER-anchored strains (**B**) grown at 37 °C. Cox4-mTaqBFP2 was used as mitochondrial marker. mTaqBFP2 colocalization over Red-C12 measured using Mander's index. Two-way ANOVA test using Turkey's multiple comparison test, ****$p$ = 0.00000215; **$p$ = 0.001. Mean and min to max are shown, box ranges from first (25th percentiles) to third quartile (75th percentiles); (**A**) LD-yArf1-GFP = 187 cells, LD-yArf1-11-GFP = 167 cells, LD-yArf1-CA-GFP = 169 cells and (**B**) ER-yArf1 = 224 cells, ER-yArf1-11 = 234 cells from n = 3 biological replicates. Scale bar: 2 μm. (**C**) Red-C12 transfer to mitochondria in y*ARF1* and *Δperoxisomes* strains using Tom70-GFP as a mitochondrial marker. GFP colocalization over Red-C12 measured using Mander's index. Student unpaired two-tailed $t$-test. *$p$ = 0.0315. Mean and min to max are shown, box ranges from first (25th percentiles) to third quartile (75th percentiles); y*ARF1* = 200 cells, y*ARF1* Δ*peroxisomes* = 185 cells from n = 3 biological replicates. Scale bar: 2 μm. (**D**) Mitochondrial morphology in WT (*PEX34*) or Δ*pex34* strains grown at 23 °C or shifted to 37 °C. Mean and standard deviation are shown; y*ARF1* 23 °C = 689 cells, y*ARF1* 37 °C = 698 cells, y*arf1-11* 23 °C = 743 cells, y*arf1-11* 37 °C = 517 cells, y*ARF1*Δ*pex34* 23 °C = 1740 cells, y*ARF1*Δ*pex34* 37 °C = 1592 cells, y*arf1-11*Δ*pex34* 23 °C = 1546 cells, y*arf1-11*Δ*pex34* 37 °C = 1622 cells from n = 3 biological replicates. Scale bar: 5 μm. (**E**) TEM of y*ARF1* and y*arf1-11* strains grown at 37 °C for 30 min. LD: lipid droplet; M: mitochondria. Scale bars: 1000 nm. (**F**) LD-mitochondria contacts quantified based on TEM images (**E**). n = 130 cells (y*ARF1* 23 °C), n = 136 cells (y*ARF1* 37 °C), n = 138 cells (y*arf1-11* 23 °C), n = 156 cells (y*arf1-11* 37 °C). (**G**, **H**) ER-mitochondria contacts quantification (**G**) and length (**H**) based on (**E**). Image in (**H**) shows an example of ER-mitochondria contact. Means are shown. Unpaired two-tailed $T$-test, y*ARF1 vs* y*arf1-11* 23 °C ****$p$ = 0.00000638; y*ARF1 vs* y*arf1-11* 37 °C ****$p$ = 0.0000000000072. n = 101 cells (y*ARF1* 23 °C), n = 139 cells (y*ARF1* 37 °C), n = 155 cells (y*arf1-11* 23 °C), n = 104 cells (y*arf1-11* 37 °C). Source numerical data are available in source data.

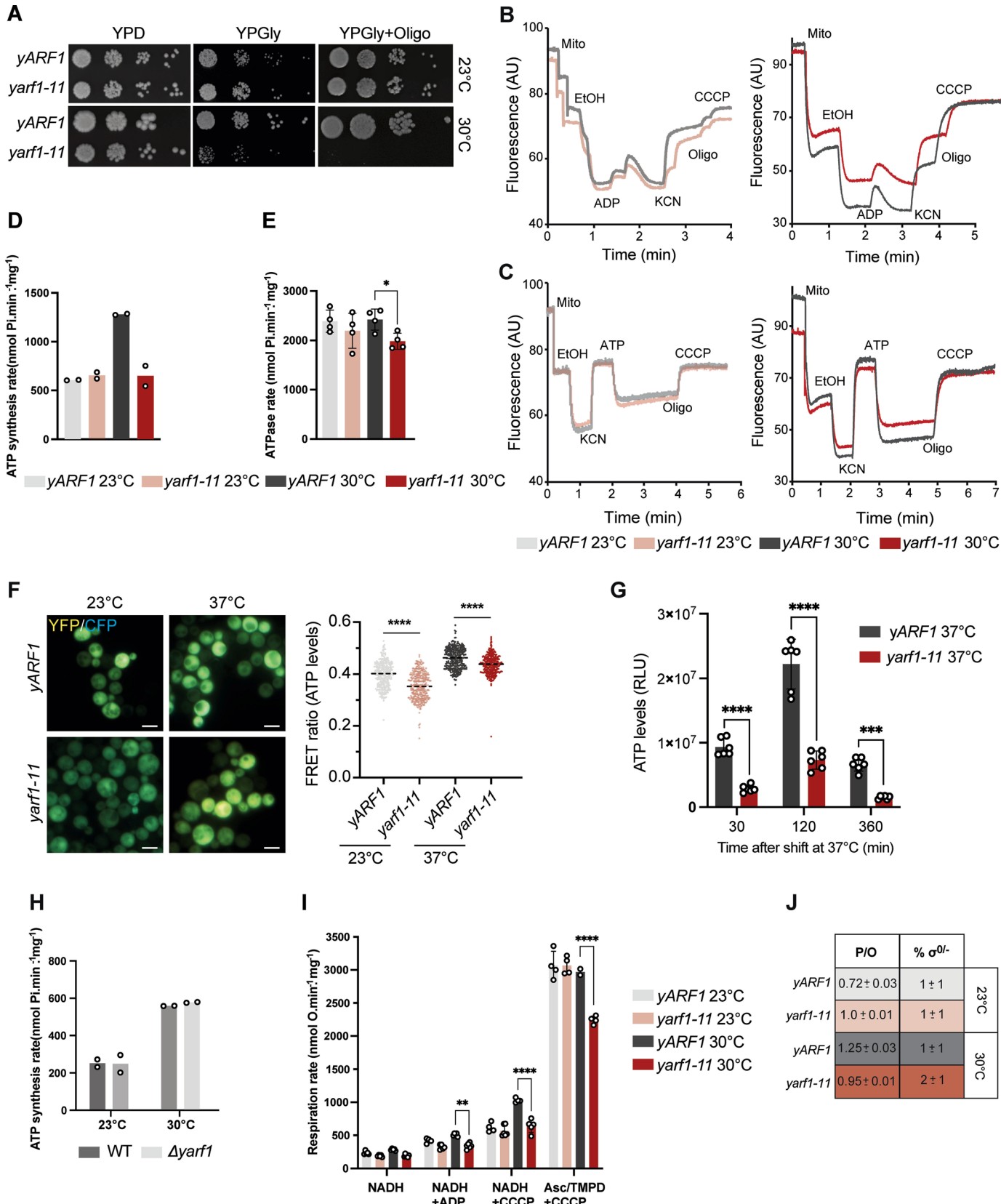

**Extended Data Fig. 8 | See next page for caption.**

**Extended Data Fig. 8 | OXPHOS activity and ATP synthesis are impaired in y*arf1-11*.** (**A**) Growth test of y*ARF1* and y*arf1-11* strain on rich media (YPD), on respiratory media (YPGly) or with oligomycine (YPGly+Oligo) at 23 °C and 30 °C. (**B**, **C**) Membrane potential measured on isolated mitochondria by Rhodamine-123 at 23 °C or 30 °C in y*ARF1* and y*arf1-11* strains in the presence of external ADP (**B**) or to test ATP-driven proton pumping (**C**). (**D**, **E**) ATP synthesis (**D**) and ATPase rate (**E**) measured on isolated mitochondria from y*ARF1* and y*arf1-11* strains grown at 23 °C or 30 °C. Means are shown from n = 2 biological replicates and 3 technical replicates and (**E**) n = 4 biological replicates. (**F**) Intracellular ATP levels quantified in y*ARF1* and y*arf1-11* strains grown at 23 °C or shifted to 37 °C. FRET ratios correspond to relative ATP levels. Median and individual values are shown. Unpaired two-tailed *T*-test, y*ARF1 vs yarf1-11* 23 °C ****$p = 0.000000000000001$; y*ARF1 vs yarf1-11* 37 °C ****$p = 0.0000000000372$; y*ARF1* 23 °C = 332 cells, y*ARF1* 37 °C = 332 cells, y*arf1-11* 23 °C = 362 cells, y*arf1-11*

37 °C = 409 cells from n = 4 biological replicates. Scale bar: 5 μm. (**G**) Relative ATP levels (RLU) measured by Luciferase assay in y*ARF1* and y*arf1-11* strains. Two-way ANOVA using Turkey's multiple comparison test, 30 min ****$p = 0.00000295$; 120 min ****$p = 0.000000000000028$; ***$p = 0.0001$. Mean and standard deviation are shown from n = 6 biological replicates. (**H**) ATP synthesis rate measured on isolated mitochondria from WT and Δy*arf1* strains grown at 23 °C or 30 °C. Means are shown from n = 2 biological replicates and 3 technical replicates. (**I**) Respiration rate measured on isolated mitochondria from y*ARF1* and y*arf1-11* strains grown at 23 °C or 30 °C. Two-way ANOVA using Sidak's multiple comparison test. NADH + CCCP y*ARF1 vs yarf1-11* 30 °C ****$p = 0.00000000097$, Asc/TMPD + CCCP y*ARF1 vs yarf1-11* 30 °C ****$p = 0.000000000014$, **$p = 0.0186$. Median and standard deviation are shown from n = 4 biological replicates. (**J**) Respiration coupled to oxygen respiration (P/O) and percentage of Rho 0 and Rho-minus were measured. Source numerical data are available in source data.

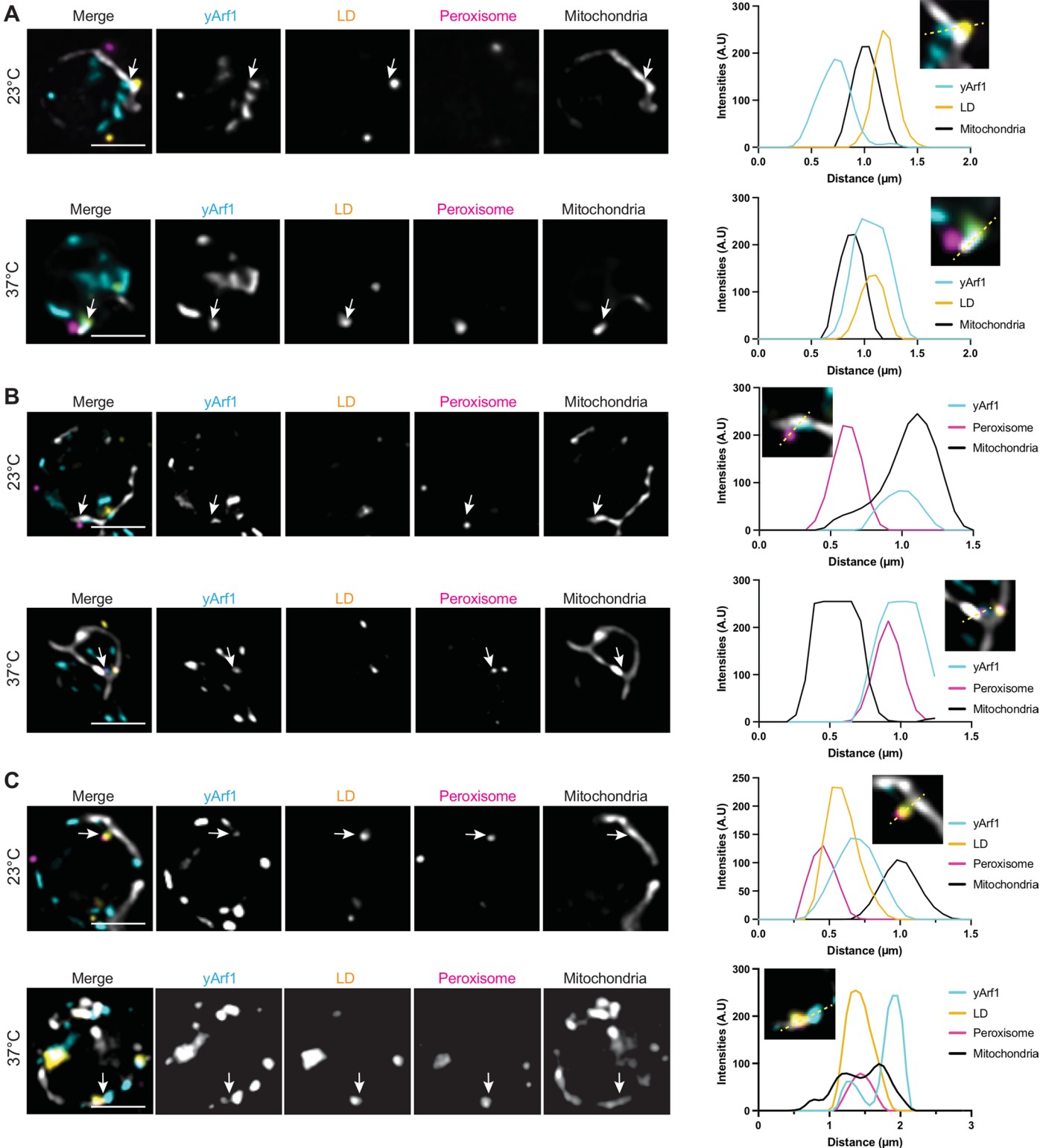

**Extended Data Fig. 9 | yArf1 is present at organellar contact sites.**
(**A**–**C**) High-resolution microscopy of yArf1-CFP strain expressing Erg6-YFP as LD marker, Pex3-mCherry as peroxisomal marker and MitoTracker Deep Red FM to stain for mitochondria. Cells were grown at 23 °C or shifted to 37 °C for 30 min. Localizations of LD and yArf1 (**A**), peroxisomes and yArf1 (**B**), or LD-peroxisomes and yArf1 (**C**) at the vicinity of mitochondria were established by following individual fluorescent intensities of each markers on a 1.5-3 μm distance. Representative images are shown and fluorescent intensities measured along the dotted lines. Single planes of 0.2 μm thickness are shown. Scale bar: 2 μm.

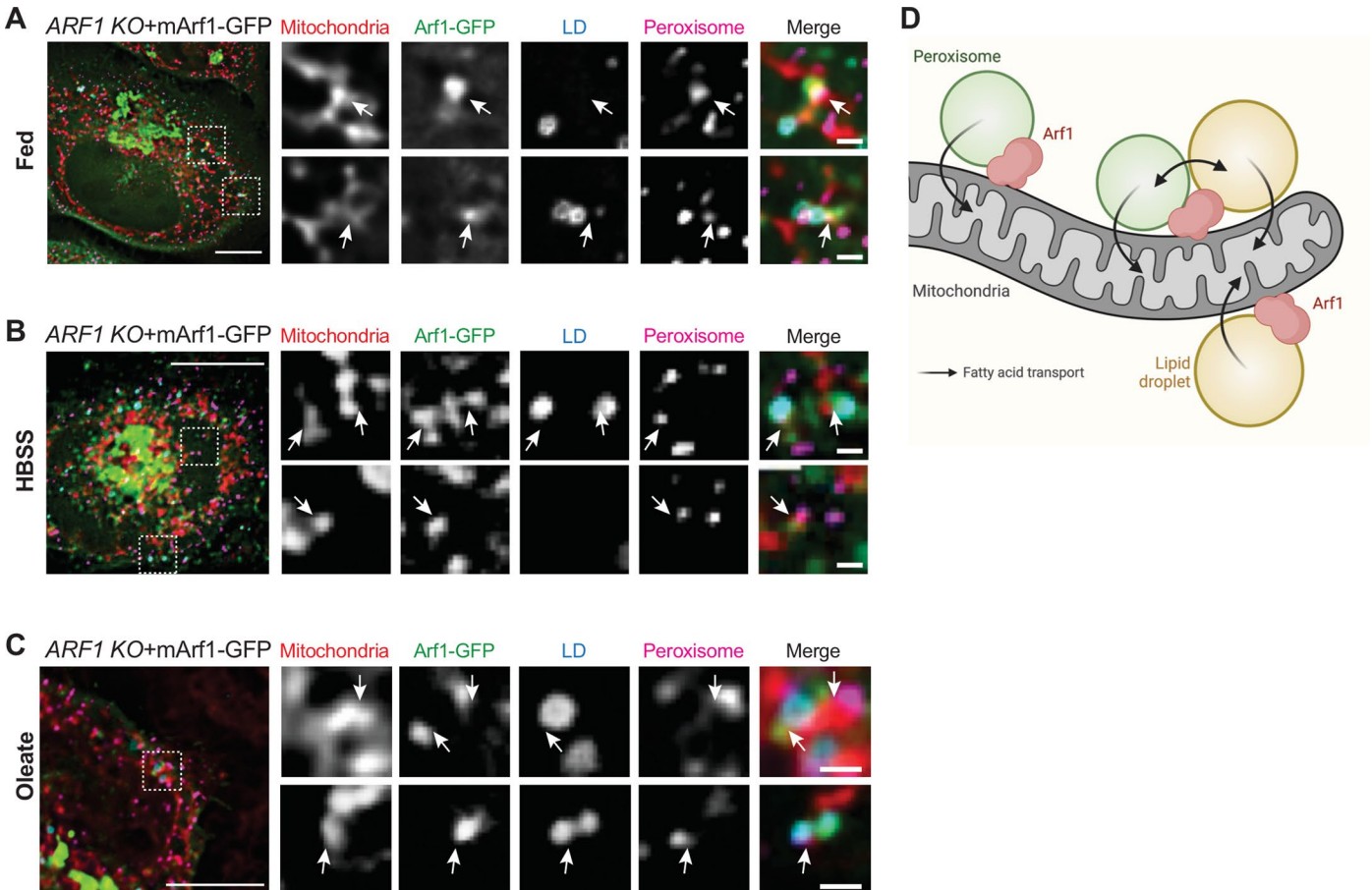

**Extended Data Fig. 10 | mArf1 is present at organellar contact sites.**
(**A**–**C**) mArf1-GFP localization in *ARF1 KO* cells grown in complete media
(**A**), shifted for 14 h in HBSS (**B**) or in the presence of oleate (**C**). Arrows indicate the
presence of mArf1 at mitochondria-LD contact sites, mitochondria-peroxisomes
contact sites or mitochondria-LD-peroxisomes contact sites. Scale bar: 10 μm,
Scale bar inlays: 1 μm. (**D**) Schematic of Arf1 localization based on images taken in
(**A**–**C**) and in Extended Data Fig. 9A–C. Created with Biorender.com.

| | |
|---|---|

# Reporting Summary

## Statistics

For all statistical analyses, confirm that the following items are present in the figure legend, table legend, main text, or Methods section.

| n/a | Confirmed | |
|---|---|---|
| ☐ | ☒ | The exact sample size (*n*) for each experimental group/condition, given as a discrete number and unit of measurement |
| ☐ | ☒ | A statement on whether measurements were taken from distinct samples or whether the same sample was measured repeatedly |
| ☐ | ☒ | The statistical test(s) used AND whether they are one- or two-sided<br>*Only common tests should be described solely by name; describe more complex techniques in the Methods section.* |
| ☒ | ☐ | A description of all covariates tested |
| ☒ | ☐ | A description of any assumptions or corrections, such as tests of normality and adjustment for multiple comparisons |
| ☐ | ☒ | A full description of the statistical parameters including central tendency (e.g. means) or other basic estimates (e.g. regression coefficient) AND variation (e.g. standard deviation) or associated estimates of uncertainty (e.g. confidence intervals) |
| ☐ | ☒ | For null hypothesis testing, the test statistic (e.g. *F*, *t*, *r*) with confidence intervals, effect sizes, degrees of freedom and *P* value noted<br>*Give P values as exact values whenever suitable.* |
| ☒ | ☐ | For Bayesian analysis, information on the choice of priors and Markov chain Monte Carlo settings |
| ☒ | ☐ | For hierarchical and complex designs, identification of the appropriate level for tests and full reporting of outcomes |
| ☐ | ☒ | Estimates of effect sizes (e.g. Cohen's *d*, Pearson's *r*), indicating how they were calculated |

*Our web collection on statistics for biologists contains articles on many of the points above.*

## Software and code

Policy information about availability of computer code

| | |
|---|---|
| Data collection | Software microscopy - ZEN 2.6 (Blue edition; widefield microscope); Live Acquisition 2.5 (FEI MORE); FV3000 (FV31S-SW Version 2.5.1) system software (Olympus Fluoview 3000); CellSens (Olympus SpinSR)<br>Western Blots: captured on a FUSION FX Vilber Lourmat using FusionCapt Advance |
| Data analysis | Microscopy: Fiji 2.3 and plugins therein; Omero webclient; Deconvolution performed with Huygens pro software 21.10<br>Statistical analysis data from Western Blots and microscopy - GraphPad Prism 9 and Excel (for MAC 16.72). |

For manuscripts utilizing custom algorithms or software that are central to the research but not yet described in published literature, software must be made available to editors and reviewers. We strongly encourage code deposition in a community repository (e.g. GitHub). See the Nature Portfolio guidelines for submitting code & software for further information.

## Data

Policy information about availability of data

All manuscripts must include a data availability statement. This statement should provide the following information, where applicable:
- Accession codes, unique identifiers, or web links for publicly available datasets
- A description of any restrictions on data availability
- For clinical datasets or third party data, please ensure that the statement adheres to our policy

The authors declare that the main data supporting the findings of this study are available within the article and its Supplementary Information files. All other data

supporting the findings of this study are available from corresponding author upon request. The lipidomics data will be made accessible through open servers of hte University of Basel.

## Human research participants

Policy information about studies involving human research participants and Sex and Gender in Research.

| | |
|---|---|
| Reporting on sex and gender | Not applicable |
| Population characteristics | Not applicable |
| Recruitment | Not applicable |
| Ethics oversight | Not applicable |

Note that full information on the approval of the study protocol must also be provided in the manuscript.

# Field-specific reporting

Please select the one below that is the best fit for your research. If you are not sure, read the appropriate sections before making your selection.

☒ Life sciences          ☐ Behavioural & social sciences          ☐ Ecological, evolutionary & environmental sciences

For a reference copy of the document with all sections, see nature.com/documents/nr-reporting-summary-flat.pdf

# Life sciences study design

All studies must disclose on these points even when the disclosure is negative.

| | |
|---|---|
| Sample size | Sample size was chosen in each case according to experimental design.  All cell biology experiments were performed at least in 3 independent biological repeats. Exact numbers are given in the figure legends. |
| Data exclusions | No data was excluded from any analyses reported in this study. |
| Replication | At least 3 independent experiments were performed per condition. All experiments were shown to be reproducible. |
| Randomization | No randomization was used. All the samples were prepared with known composition and contained appropriate controls, which mitigates the influence of co-variates. |
| Blinding | Blinding was not used in this study as phenotypes were obvious. |

# Behavioural & social sciences study design

All studies must disclose on these points even when the disclosure is negative.

| | |
|---|---|
| Study description | Briefly describe the study type including whether data are quantitative, qualitative, or mixed-methods (e.g. qualitative cross-sectional, quantitative experimental, mixed-methods case study). |
| Research sample | State the research sample (e.g. Harvard university undergraduates, villagers in rural India) and provide relevant demographic information (e.g. age, sex) and indicate whether the sample is representative. Provide a rationale for the study sample chosen. For studies involving existing datasets, please describe the dataset and source. |
| Sampling strategy | Describe the sampling procedure (e.g. random, snowball, stratified, convenience). Describe the statistical methods that were used to predetermine sample size OR if no sample-size calculation was performed, describe how sample sizes were chosen and provide a rationale for why these sample sizes are sufficient. For qualitative data, please indicate whether data saturation was considered, and what criteria were used to decide that no further sampling was needed. |
| Data collection | Provide details about the data collection procedure, including the instruments or devices used to record the data (e.g. pen and paper, computer, eye tracker, video or audio equipment) whether anyone was present besides the participant(s) and the researcher, and whether the researcher was blind to experimental condition and/or the study hypothesis during data collection. |
| Timing | Indicate the start and stop dates of data collection. If there is a gap between collection periods, state the dates for each sample cohort. |

| Data exclusions | *If no data were excluded from the analyses, state so OR if data were excluded, provide the exact number of exclusions and the rationale behind them, indicating whether exclusion criteria were pre-established.* |
|---|---|
| Non-participation | *State how many participants dropped out/declined participation and the reason(s) given OR provide response rate OR state that no participants dropped out/declined participation.* |
| Randomization | *If participants were not allocated into experimental groups, state so OR describe how participants were allocated to groups, and if allocation was not random, describe how covariates were controlled.* |

# Ecological, evolutionary & environmental sciences study design

All studies must disclose on these points even when the disclosure is negative.

| Study description | *Briefly describe the study. For quantitative data include treatment factors and interactions, design structure (e.g. factorial, nested, hierarchical), nature and number of experimental units and replicates.* |
|---|---|
| Research sample | *Describe the research sample (e.g. a group of tagged Passer domesticus, all Stenocereus thurberi within Organ Pipe Cactus National Monument), and provide a rationale for the sample choice. When relevant, describe the organism taxa, source, sex, age range and any manipulations. State what population the sample is meant to represent when applicable. For studies involving existing datasets, describe the data and its source.* |
| Sampling strategy | *Note the sampling procedure. Describe the statistical methods that were used to predetermine sample size OR if no sample-size calculation was performed, describe how sample sizes were chosen and provide a rationale for why these sample sizes are sufficient.* |
| Data collection | *Describe the data collection procedure, including who recorded the data and how.* |
| Timing and spatial scale | *Indicate the start and stop dates of data collection, noting the frequency and periodicity of sampling and providing a rationale for these choices. If there is a gap between collection periods, state the dates for each sample cohort. Specify the spatial scale from which the data are taken* |
| Data exclusions | *If no data were excluded from the analyses, state so OR if data were excluded, describe the exclusions and the rationale behind them, indicating whether exclusion criteria were pre-established.* |
| Reproducibility | *Describe the measures taken to verify the reproducibility of experimental findings. For each experiment, note whether any attempts to repeat the experiment failed OR state that all attempts to repeat the experiment were successful.* |
| Randomization | *Describe how samples/organisms/participants were allocated into groups. If allocation was not random, describe how covariates were controlled. If this is not relevant to your study, explain why.* |
| Blinding | *Describe the extent of blinding used during data acquisition and analysis. If blinding was not possible, describe why OR explain why blinding was not relevant to your study.* |

Did the study involve field work? ☐ Yes ☒ No

# Reporting for specific materials, systems and methods

We require information from authors about some types of materials, experimental systems and methods used in many studies. Here, indicate whether each material, system or method listed is relevant to your study. If you are not sure if a list item applies to your research, read the appropriate section before selecting a response.

## Materials & experimental systems

| n/a | Involved in the study |
|---|---|
| ☐ | ☒ Antibodies |
| ☐ | ☒ Eukaryotic cell lines |
| ☒ | ☐ Palaeontology and archaeology |
| ☒ | ☐ Animals and other organisms |
| ☒ | ☐ Clinical data |
| ☒ | ☐ Dual use research of concern |

## Methods

| n/a | Involved in the study |
|---|---|
| ☒ | ☐ ChIP-seq |
| ☒ | ☐ Flow cytometry |
| ☒ | ☐ MRI-based neuroimaging |

## Antibodies

| Antibodies used | HRP-conjugated anti-mouse, 1:10,000;  Invitrogen 31430, anti-HA primary antibody (1:5,000, Eurogentec 16B12) , or anti-Pgk1 primary antibody (1:5,000, Invitrogen clone 22C5D8), beta-COP (1:500; CM1; hybridoma supernatant; gift from Dr. Felix Wieland, |
|---|---|

Heidelberg University, Heidelberg, Germany), monoclonal mouse anti-CLIMP63 (1:1,000, G1/296 gift from the Hauri lab), GM130 antibody (1:1,000, Cell Signalling 12480S), TOM20 antibody (1:200, Santa Cruz sc-17764), anti-Arf1 (1:2,500, Abnova MAB10011), anti-actin (1:100,000, Sigma-Aldrich MAB1501), HRP-conjugated secondary antibody (1:10,000; anti-rabbit, Sigma-Aldrich A0545 or anti-mouse, Sigma-Aldrich A0168), anti-GFP antibody (1:100, 6556 Abcam), goat anti-rabbit secondary antibody coupled with 10 nm gold (1:100; BBI solutions EM.GAR10/2); Secondary Alexa-Fluor 568 (1:500; anti-mouse A10037 and anti-rabbit A10042, Invitrogen)

Validation | All primary antibodies, except anti-ß-COP and anti-CLIMP63 , are commercially available and validated by the manufacturer. In addition, most of the antibodies have been used by other research groups previously as referenced in the associated manuscript. Anti-ß-COP and anti-CLIMP63 were validated by the Wieland and Hauri labs, respectively, and have been widely used by the scientific community.

# Eukaryotic cell lines

Policy information about cell lines and Sex and Gender in Research

Cell line source(s) | HeLa alpha cell lines obtained from ATCC and HeLa ARF1 KO cells were kind gift from Prof. Martin Spiess and are further referenced here : Pennauer M, Buczak K, Prescianotto-Baschong C, Spiess M. Shared and specific functions of Arfs 1-5 at the Golgi revealed by systematic knockouts. J Cell Biol. 2022 Jan 3;221(1):e202106100. doi: 10.1083/jcb.202106100. Epub 2021 Nov 8. PMID: 34749397; PMCID: PMC8579194.

Authentication | Standard cell lines, authenticated by ATCC. Recently (2021), the cell lines' identities were authenticated by STR analysis by Microsynth AG (Balgach Switzerland).

Mycoplasma contamination | We routinely test our cell lines for mycoplasma contamination. All cell lines were negative.

Commonly misidentified lines (See ICLAC register) | No commonly misidentified cell lines were used

# Palaeontology and Archaeology

Specimen provenance | *Provide provenance information for specimens and describe permits that were obtained for the work (including the name of the issuing authority, the date of issue, and any identifying information). Permits should encompass collection and, where applicable, export.*

Specimen deposition | *Indicate where the specimens have been deposited to permit free access by other researchers.*

Dating methods | *If new dates are provided, describe how they were obtained (e.g. collection, storage, sample pretreatment and measurement), where they were obtained (i.e. lab name), the calibration program and the protocol for quality assurance OR state that no new dates are provided.*

☐ Tick this box to confirm that the raw and calibrated dates are available in the paper or in Supplementary Information.

Ethics oversight | *Identify the organization(s) that approved or provided guidance on the study protocol, OR state that no ethical approval or guidance was required and explain why not.*

Note that full information on the approval of the study protocol must also be provided in the manuscript.

# Clinical data

Policy information about clinical studies
All manuscripts should comply with the ICMJE guidelines for publication of clinical research and a completed CONSORT checklist must be included with all submissions.

Clinical trial registration | *Provide the trial registration number from ClinicalTrials.gov or an equivalent agency.*

Study protocol | *Note where the full trial protocol can be accessed OR if not available, explain why.*

Data collection | *Describe the settings and locales of data collection, noting the time periods of recruitment and data collection.*

Outcomes | *Describe how you pre-defined primary and secondary outcome measures and how you assessed these measures.*

# Dual use research of concern

Policy information about dual use research of concern

## Hazards

Could the accidental, deliberate or reckless misuse of agents or technologies generated in the work, or the application of information presented in the manuscript, pose a threat to:

| No | Yes | |
|----|-----|---|
| ☒ | ☐ | Public health |
| ☒ | ☐ | National security |
| ☒ | ☐ | Crops and/or livestock |
| ☒ | ☐ | Ecosystems |
| ☒ | ☐ | Any other significant area |

## Experiments of concern

Does the work involve any of these experiments of concern:

| No | Yes | |
|----|-----|---|
| ☒ | ☐ | Demonstrate how to render a vaccine ineffective |
| ☒ | ☐ | Confer resistance to therapeutically useful antibiotics or antiviral agents |
| ☒ | ☐ | Enhance the virulence of a pathogen or render a nonpathogen virulent |
| ☒ | ☐ | Increase transmissibility of a pathogen |
| ☒ | ☐ | Alter the host range of a pathogen |
| ☒ | ☐ | Enable evasion of diagnostic/detection modalities |
| ☒ | ☐ | Enable the weaponization of a biological agent or toxin |
| ☒ | ☐ | Any other potentially harmful combination of experiments and agents |

# ChIP-seq

## Data deposition

☐ Confirm that both raw and final processed data have been deposited in a public database such as GEO.

☐ Confirm that you have deposited or provided access to graph files (e.g. BED files) for the called peaks.

| | |
|---|---|
| **Data access links** *May remain private before publication.* | *For "Initial submission" or "Revised version" documents, provide reviewer access links. For your "Final submission" document, provide a link to the deposited data.* |
| **Files in database submission** | *Provide a list of all files available in the database submission.* |
| **Genome browser session** (e.g. UCSC) | *Provide a link to an anonymized genome browser session for "Initial submission" and "Revised version" documents only, to enable peer review. Write "no longer applicable" for "Final submission" documents.* |

## Methodology

| | |
|---|---|
| **Replicates** | *Describe the experimental replicates, specifying number, type and replicate agreement.* |
| **Sequencing depth** | *Describe the sequencing depth for each experiment, providing the total number of reads, uniquely mapped reads, length of reads and whether they were paired- or single-end.* |
| **Antibodies** | *Describe the antibodies used for the ChIP-seq experiments; as applicable, provide supplier name, catalog number, clone name, and lot number.* |
| **Peak calling parameters** | *Specify the command line program and parameters used for read mapping and peak calling, including the ChIP, control and index files used.* |
| **Data quality** | *Describe the methods used to ensure data quality in full detail, including how many peaks are at FDR 5% and above 5-fold enrichment.* |
| **Software** | *Describe the software used to collect and analyze the ChIP-seq data. For custom code that has been deposited into a community repository, provide accession details.* |

# Flow Cytometry

## Plots

Confirm that:

☒ The axis labels state the marker and fluorochrome used (e.g. CD4-FITC).

☒ The axis scales are clearly visible. Include numbers along axes only for bottom left plot of group (a 'group' is an analysis of identical markers).

☒ All plots are contour plots with outliers or pseudocolor plots.

☒ A numerical value for number of cells or percentage (with statistics) is provided.

## Methodology

| | |
|---|---|
| Sample preparation | ARF1 KO HeLa alpha cells transfected with either ARF1-EGFP, ARF1-11-EGFP or empty EGFP vector were seeded in 12-well plates at a density of 5500 cells/well. Every 24h for 6 consecutive days, cells from one well for each cell line were trypsinized and resuspended in PBS complemented with 2% FCS. |
| Instrument | BD LSR Fortessa Analyzer |
| Software | FlowJo Version 10.9.0 |
| Cell population abundance | GFP intensity of 100.000 cells were measured per sample (single cells were 97-98% of the initial cell population). |
| Gating strategy | To discard the cell debri from the initial cell population FSC/SSC gating was applied. Single cells were determined by SSC-A/SSC-W and by FSC-A/FSC-W gating. From the single cell population GFP+ cells were gated by GFP-A/SSC-A and the GFP+ cells were counted in each sample. |

☒ Tick this box to confirm that a figure exemplifying the gating strategy is provided in the Supplementary Information.

# Magnetic resonance imaging

## Experimental design

| | |
|---|---|
| Design type | *Indicate task or resting state; event-related or block design.* |
| Design specifications | *Specify the number of blocks, trials or experimental units per session and/or subject, and specify the length of each trial or block (if trials are blocked) and interval between trials.* |
| Behavioral performance measures | *State number and/or type of variables recorded (e.g. correct button press, response time) and what statistics were used to establish that the subjects were performing the task as expected (e.g. mean, range, and/or standard deviation across subjects).* |

## Acquisition

| | |
|---|---|
| Imaging type(s) | *Specify: functional, structural, diffusion, perfusion.* |
| Field strength | *Specify in Tesla* |
| Sequence & imaging parameters | *Specify the pulse sequence type (gradient echo, spin echo, etc.), imaging type (EPI, spiral, etc.), field of view, matrix size, slice thickness, orientation and TE/TR/flip angle.* |
| Area of acquisition | *State whether a whole brain scan was used OR define the area of acquisition, describing how the region was determined.* |

Diffusion MRI      ☐ Used      ☐ Not used

## Preprocessing

| | |
|---|---|
| Preprocessing software | *Provide detail on software version and revision number and on specific parameters (model/functions, brain extraction, segmentation, smoothing kernel size, etc.).* |
| Normalization | *If data were normalized/standardized, describe the approach(es): specify linear or non-linear and define image types used for transformation OR indicate that data were not normalized and explain rationale for lack of normalization.* |
| Normalization template | *Describe the template used for normalization/transformation, specifying subject space or group standardized space (e.g. original Talairach, MNI305, ICBM152) OR indicate that the data were not normalized.* |
| Noise and artifact removal | *Describe your procedure(s) for artifact and structured noise removal, specifying motion parameters, tissue signals and physiological signals (heart rate, respiration).* |

| Volume censoring | *Define your software and/or method and criteria for volume censoring, and state the extent of such censoring.* |

## Statistical modeling & inference

| Model type and settings | *Specify type (mass univariate, multivariate, RSA, predictive, etc.) and describe essential details of the model at the first and second levels (e.g. fixed, random or mixed effects; drift or auto-correlation).* |

| Effect(s) tested | *Define precise effect in terms of the task or stimulus conditions instead of psychological concepts and indicate whether ANOVA or factorial designs were used.* |

Specify type of analysis:  ☐ Whole brain  ☐ ROI-based  ☐ Both

| Statistic type for inference<br>(See Eklund et al. 2016) | *Specify voxel-wise or cluster-wise and report all relevant parameters for cluster-wise methods.* |

| Correction | *Describe the type of correction and how it is obtained for multiple comparisons (e.g. FWE, FDR, permutation or Monte Carlo).* |

## Models & analysis

| n/a | Involved in the study |
|---|---|
| ☐ | ☐ Functional and/or effective connectivity |
| ☐ | ☐ Graph analysis |
| ☐ | ☐ Multivariate modeling or predictive analysis |

| Functional and/or effective connectivity | *Report the measures of dependence used and the model details (e.g. Pearson correlation, partial correlation, mutual information).* |

| Graph analysis | *Report the dependent variable and connectivity measure, specifying weighted graph or binarized graph, subject- or group-level, and the global and/or node summaries used (e.g. clustering coefficient, efficiency, etc.).* |

| Multivariate modeling and predictive analysis | *Specify independent variables, features extraction and dimension reduction, model, training and evaluation metrics.* |

