## [Peer Review File · Nature Cell Biology]

Peer Review Information

Journal: Nature Cell Biology

Manuscript Title: Arf1 coordinates fatty acid metabolism and mitochondrial homeostasis

Corresponding author name(s): Professor Anne Spang

Editorial Notes:

Reviewer Comments & Decisions:

Decision Letter, initial version:
--

Dear Anne,

Thank you for submitting your manuscript "The small GTPase Arf1 regulates ATP synthesis and mitochondria homeostasis by modulating fatty acid metabolism" to Nature Cell Biology. Thank you also very much for your patience while the manuscript was under review. It has now been seen by 3 referees, who are experts in trafficking, yeast, GTPases (Referee #1); lipid trafficking, yeast, organelle contacts (Referee #2); and mitochondria-ER contacts, lipid metabolism (Referee #3), and whose comments are pasted below. In light of their advice, we regret that we cannot offer to publish the study in Nature Cell Biology.

As you will see, although the reviewers found this work interesting, they raised a number of concerns that question the strength of the data and of the conclusions that can be drawn, including with the strength of the data and the degree of examination of how Arf1 may function at mitochondria-LD contacts in a WT setting. In light of the points they raise, we find the present data-set too preliminary to pursue at this stage. Despite our interest in the results, we agree with the reviewers that mechanistic insight explaining the role of Arf1 in a WT setting would be needed, and thoroughly addressing their comments about the phenotypes, localization data, alternative models and interpretations would be important. Substantial additional experimentation would be needed, and we feel that this is greater than what is reasonable for us to request in a standard revision period.

[Of note, one reviewer suggested - and asked us to relay - that it would likely be helpful to the reader if you could please expand in the text, cite, and discuss published primary literature on mammalian PLIN5 and MIGA2, which have been shown to be involved in forming the LD-mitochondria organelle interface in mammalian cells. This is a very minor point for your consideration only. It did not contribute to the decision, but, as per the reviewer's request, we wanted to share it with you.]

I very much hope that you find the reviews helpful as you decide how to move forward with the study. We are very sorry that we could not be more positive on this occasion, but we thank you for the opportunity to consider this work and thank you again for your patience.

With kind regards,
Melina

Melina Casadio, PhD
Senior Editor, Nature Cell Biology
ORCID ID: <https://orcid.org/0000-0003-2389-2243>

Reviewers' comments:

Reviewer #1 (Remarks to the Author):

In this study, the authors sought to understand the role(s) of Arf1 in mitochondrial and lipid droplet function. They made extensive use of *arf1-11*, a heat-sensitive yeast mutant that displays altered mitochondrial dynamics and defective FA transport/metabolism. The authors conclude that Arf1 activity is important for FA mobilization from LDs and, consequently, mitochondrial morphology and function.

This model is intriguing, especially given that some of the observed *arf1-11* phenotypes closely mirror those of ARF1 cells with genetically- or pharmacologically-impaired fatty acid metabolism (e.g., globular mitochondria à la *arf1-11* mutants in ARF1 yeast cells that cannot synthesize FAs). However, it would be more compelling if it were accompanied by a plausible mechanism for how Arf1 normally regulates this process. What effector(s) might it be recruiting, and to where? Is COPI involved? To what extent (and how) might Arf1 normally manage contact sites among these organelles? At the very least, including a more detailed examination of published literature concerning roles for Arf1 in LD and mitochondrial function in the Discussion section would be helpful in this regard. In addition, although the effect of the *arf1-11* mutant is clear, the authors have not fully investigated whether Arf1 is required in wild-type cell physiology regarding mitochondrial dynamics and FA transport/metabolism.

Major points

1. In Figure 5C, ARF1 KO cells transfected with mArf1-11-GFP have LD numbers comparable to wild-type non-transfected cells, while non-transfected ARF1 KO cells (and ARF1 KO cells transfected with mArf1-GFP) have far fewer LDs. Do yeast *arf1Δ* cells also contain fewer LDs?
2. Similarly, what do LDs, mitochondria, and ER-mitochondria/LD-mitochondria contact sites look like in *arf1ΔARF2* yeast? TAG levels? ATP production?
3. Based on prior literature, one would guess that COPI is the critical Arf1 effector for lipid metabolism based on its established role with lipid droplets. Does COPI distribution change in *arf1-11* cells? If not, can the authors identify any Arf1 effectors that have altered localization and/or function in *arf1-11* cells (either at permissive or restrictive temperatures)?
4. Why were the membrane potential measurements performed at 23°C and 30°C, while other experiments (including other metabolism measurements) were done at 26°C and 37°C?

Minor point

5. The figure legends for Figure 5 do not match the figure.

Reviewer #2 (Remarks to the Author):

This study investigates how Arf1 regulates mitochondrial shape and function, which has been suggested by previous work from this group and others. It argues that Arf1 is required for mitochondrial fission and fusion and that Arf1 regulates beta oxidation by controlling the movement of fatty acids out of lipid droplets to sites of beta oxidation in mitochondria and, in *S. cerevisiae*, peroxisomes. These are fascinating new ideas about the functions of Arf1 beyond vesicular trafficking. However, it remains unclear how directly Arf1 is regulating mitochondrial dynamics or fatty acid oxidation and a good deal more mechanistic insight is necessary for this study to be appropriate for NCB. Here are the major issues.

1. It is not clear that Arf1 directly regulates mitochondrial fission and fusion. The study suggests that Arf1 on membranes near mitochondria somehow regulates mitochondrial dynamics, which is consistent with the finding that Arf1-GFP is next to about 80% of fission and fusion events (Fig 1 and movies). However, wild-type Arf1 is mostly on the cis- and trans-Golgi, which is not known to directly participate in mitochondrial fission or fusion. In addition, Arf1 does not need to be on the Golgi to regulate mitochondrial dynamics since they are still normal when Arf1 is on the ER (Fig. 3C-E). It is not clear how Arf1 could directly regulate mitochondrial dynamics from different organelles. Some mechanistic insight into how this occurs is necessary.
2. The study suggests that Arf1-11 is sequestered on lipid droplets (and other membranes) at non-permissive temperature, rendering it inactive. However, it is also possible that Arf1-11 on lipid droplets is toxic. If this is correct, Arf1-11 will disrupt mitochondrial dynamics even when wild-type Arf1 is present. This should be tested.

3. Much of the evidence that Arf1 regulates TAG mobilization out of lipid droplets is indirect. Rates of fatty acid mobilization should be directly measured. One way to do this is to treat cells with cerulenin and terbinafine (for example, see PMID: 16135509) and measure TAG and SE levels over time. This will make it possible directly determine rates of mobilization in cells expressing Arf1 or Arf1-11. It might be useful to attach a degron to Arf1 to rapidly remove it from cells to see whether changes in TAG mobilization are specific to Arf1-11.

4. For this study to be appropriate for NCB, some mechanistic insight into how Arf1 regulates neutral lipid mobilization is necessary. How does Arf1 regulate levels of Pxa2 and Pox1? Does down regulation of these proteins, by itself, reduced TAG mobilization? If Arf1 regulates TAG mobilization or fatty acid transport in other ways, how does it do so? A complete answer to these questions beyond the scope of this study, but some hint of mechanism is necessary to support the claim Arf1 is directly involved.

5. To test the idea that Arf1 on the surface of LDs regulates neutral lipid mobilization from LDs, rates of mobilization should be determined when Arf1 is artificially targeting to LDs or anchored away from the LDs.

6. The idea that rates of TAG mobilization significantly alter mitochondrial shape is not well supported. Fig S7C shows that cells expressing wild-type Arf1 have similar mitochondrial shapes whether they do or not produce TAG.

7. Minor: The data in Fig 5E should be presented as bar graphs as in Fig. 5D,F. Whether differences are statistically significant should be determined.

Reviewer #3 (Remarks to the Author):

The manuscript by Enkler et al. reports on mechanisms by which cells coordinate fatty acid distribution between different organelles. The paper presents a number of unexpected results, resolves a major inconsistency in the field and potentially unravels an interesting new mechanism in molecular metabolism. The manuscript is timely and of great interest to a broad readership in the cell biology community. Pending the clarifications requested below the novelty, originality and importance of the suggested mechanism would make the paper an excellent candidate for publication in NCB.

Summary of key results

Enkler et al. focus specifically on the role of Arf1, a small monomeric GTPase, in organizing FA transport between lipid droplets (LDs) and peroxisomes or LDs and mitochondria in yeast and in mammalian cells, respectively. Arf1 is classically associated with coordinating vesicular membrane traffic events at the Golgi complex, but has recently been shown to have additional importance for the regulation of mitochondrial fusion and fission. The underlying mechanisms are, however, not completely understood.

Arf1 is mostly localized on the Golgi complex both in yeast and mammalian cells, but a sub-fraction is targeted to the mitochondria where it is involved in maintaining mitochondrial morphology. The authors have now clarified that Arf1-11 (a yeast Arf1 mutant) is a gain of function or a hyperactive

Arf1 version, rather than a loss of function allele as previously assumed. This is an important finding because it reconciles conflicting results between yeast and mammalian cells. The new data clearly establish that loss of Arf1 function causes elongation of mitochondria (which was previously known from work in mammalian cells) and gain of function mutants shorten mitochondria.

The authors find that the hyperactive GTP-bound Arf1-11 is found less at the Golgi, but targets in yeast to the ER at permissive temperature and localizes to LDs at 37°C. In mammalian cells an Arf1 version containing the point mutations from yeast Arf1-11 is found mostly on LDs and mitochondria. The new localization of the mutant leads to fission of mitochondria. The authors suggest that this change in shape is not a direct effect of Arf1 on the fusion/fission machinery but is rather explained by defects in lipid and energy metabolism.

When Arf1-11 localizes to LDs the number of LDs increase and the cells contain higher amounts of triglycerides (TGs), which are stored in LDs. According to the authors, TGs accumulate because Arf1-11 blocks mobilization of FAs from LDs, and inhibits beta oxidation in peroxisomes of yeast cells or in mammalian mitochondria. In both cases Arf1-11 reduces acetyl-CoA levels, and decreases ATP production. How lower ATP levels lead to fission of mitochondria is not yet clear and is possibly a good starting point for a next study.

Comments on data and methodology

The manuscript is complex, but clearly written. The most convincing part are the data in figures 1 and 2 showing that Arf1 localizes to sites where mitochondria undergo fusion and fission. Especially the fission part is clear. It is also convincing that arf1-11 is GTP bound. It would have been helpful to add GTPase assays to show that the protein is GTPase deficient, but perhaps this was already done in the literature. Are mitochondria globular in Arf1 Q71L? This was not really clear to this reviewer. Can the authors please clarify these points?

The data in figure 3 is convincing, but with respect to the main message in the paper (focusing mostly on the role of Arf1 in mitochondrial fission) it is a negative result. Would it be possible to add a lipid droplet binding domain to γ Arf1? This should have similar effects as the Arf1-11 mutant and would strengthen the point that the localization to the LDs is important for the mitochondrial fission. Is there a specific domain in erg6 which the authors use as LD marker that can be moved to γ Arf1? This LD localized mutant should lead to globular mitochondria. This or an equivalent experiment should be added.

The experiments in figure 4 are technically elegant. It is, however, not easy to understand why the localization of Arf1-11 is so different between panel B and F. In theory this should be the same conditions. How often can the localization along the mitochondrial profile be observed? Some quantification would be helpful, e.g. % of cells with Arf1-11 on long mitochondria. Conceptually, it is not clear why the mammalian mitochondria do not undergo fission when Arf1-11 is expressed. The authors should clarify this point.

The EM images in figure 5 suggest that γ Arf1-11 cells contain much more internal membranes than

controls. It seems however that this is not reflected in the lipid analysis, and the membrane lipids stay unchanged. Are the sphingolipids increasing? The methods section indicates that sphingolipids were analyzed, but the data is missing from this version of the paper. Please add the data or remove the text from the methods.

The internal membranes appear quite white, which might indicate that they are rich in TG. The authors might want to point this out and compare the results to EM images from Pex30 complex knockout cells where similar observations were made.

In general, the figure legends in figure 5 seem to come from a different version of the paper, which complicates the assessment of the data. This should be corrected. The figure legends for C are completely missing and the units for the box plots are not clear. What does % of HeLa mean and how are the values normalized?

Panel E has to be presented in the same format as panel D and F (bar graphs with confidence intervals). It is not clear how the log₂ fold change was calculated, the relevant methods are incomprehensible (what is sample 1?). This is a key result in the paper and its presentation should be improved.

The quantification in figure H and I is confusing. The figure legend is difficult to understand, and it is not indicated how many images were analyzed. Please, make the data more accessible to the reader.

The yeast data in figures 6 and 7 point at a possible function of beta oxidation in maintaining mitochondrial morphology. Figure 6 indicates that fatty acid beta oxidation in peroxisomes is switched on after heat shock leading to the production of acetyl-CoA for ATP synthesis in mitochondria. In *arf1-11* cells this pathway is blocked. It is, however, not clear why mitochondria undergo fission under these conditions. The authors have to include more controls and show that mitochondria are round after heat shock in cells with knockouts for beta oxidation enzymes (e.g. *Pox1* deletion). This control is necessary, because it is not clear why ATP production would depend on respiration specifically under heat shock. Usually yeast cells ferment and produce all ATP by glycolysis. Why would heat shock make aerobic respiration essential, and why would this ATP pool be necessary for mitochondrial homeostasis?

The title of figure 7 is inconsistent with the data. According to the model in figure 6, acetyl-CoA and not the entire fatty acid should enter mitochondria of yeast. If the entire fatty acids enter yeast mitochondria as suggested in the title and the figure legend, the whole concept is incorrect. The images in figure 7E probably show that the bodipy dye remains attached to the acetyl CoA after the C12red fatty acids have been oxidized in the peroxisomes and therefore Bodipy-acetyl-CoA stains mitochondria red. If the whole fatty acid reaches the mitochondria, the mechanism that is suggested in figure 6 would be wrong.

In mammalian cells it is clear that the fatty acids enter into mitochondria from LDs as shown by previous work (Rambold et al. in *Dev Cell* 2015). It would be interesting and novel if Arf1 is necessary for the relevant transport step as suggested by the authors. Can wild-type Arf1-GFP be seen at the contact sites between C12red LDs and mitochondria. This would be an important result!

In its current form the conclusions from figure 6 and 7 (figures S6-S8) are not robust and the interpretations have to be corrected.

The current legend for figure 7 likely belongs to a different version of the paper. This should be corrected.

**Although we cannot publish your paper, it may be appropriate for another journal in the Nature Portfolio. If you wish to explore the journals and transfer your manuscript please use our manuscript transfer portal. If you transfer to Nature journals or the Communications journals, you will not have to re-supply manuscript metadata and files. This link can only be used once and remains active until used.

All Nature Portfolio journals are editorially independent, and the decision on your manuscript will be taken by their editors. For more information, please see our manuscript transfer FAQ page.

Note that any decision to opt in to In Review at the original journal is not sent to the receiving journal on transfer. You can opt in to In Review at receiving journals that support this service by choosing to modify your manuscript on transfer. In Review is available for primary research manuscript types only.

Author Rebuttal to Initial comments

We wish to thank the reviewers for useful and insightful comments, which helped to provide more mechanistical insights into the evolutionary conserved role of Arf1 in lipid metabolism and mitochondrial function. Please find the answers to the individual points below.

Reviewer #1 (Remarks to the Author):

In this study, the authors sought to understand the role(s) of Arf1 in mitochondrial and lipid droplet function. They made extensive use of *arf1-11*, a heat-sensitive yeast mutant that displays altered mitochondrial dynamics and defective FA transport/metabolism. The authors conclude that Arf1 activity is important for FA mobilization from LDs and, consequently, mitochondrial morphology and function.

This model is intriguing, especially given that some of the observed *arf1-11* phenotypes closely mirror those of ARF1 cells with genetically- or pharmacologically-impaired fatty acid metabolism (e.g., globular mitochondria in *arf1-11* mutants in ARF1 yeast cells that cannot synthesize FAs). However, it would be more compelling if it were accompanied by a plausible mechanism for how Arf1 normally regulates this process. What effector(s) might it be recruiting, and to where? Is COPI involved? To what extent (and how) might Arf1 normally manage contact sites among these organelles? At the very least, including a more detailed examination of published literature concerning roles for Arf1 in LD and mitochondrial function in the Discussion section would be helpful in this regard. In addition, although the effect of the *arf1-11* mutant is clear, the authors have not fully investigated whether Arf1 is required in wild-type cell physiology regarding mitochondrial dynamics and FA transport/metabolism.

We wish to thank the reviewer for the positive feedback and that s/he finds our model intriguing.

As the reviewer will see below, we investigated the mechanistic role of Arf1 in more detail during the revision of the manuscript. First of all, our data provide evidence that the role of Arf1 in lipid/acyl chain transfer is independent of coatamer, indicating that COPI is not involved. Second, we show that Arf1 interacts with Dga1, the last enzyme in TAG biogenesis, the triglyceride lipase Tgl4 and the acyl-CoA synthase Faa1 on LDs. Thus, the presence of the Arf1 on LDs may positively regulate TAG synthesis and downregulate TAG conversion into acyl-CoA, the substrate for import into peroxisomes in yeast. Third, Arf1 can also interact with the peroxisomal protein Pex13. However, this interaction should be in 'trans' because we found Arf1 only to be juxtaposed to peroxisomes and not co-localizing with them. These data would be consistent with Arf1 being at contacts between LDs and peroxisomes. Indeed, we found Arf1 localizing at LD-peroxisome, LD-mitochondria and LD-peroxisome-mitochondria contact sites both in yeast and in mammalian cells. Thus, our data are consistent with the model that Arf1 regulates lipid metabolism in yeast and mammalian cells.

Major points

1. In Figure 5C, ARF1 KO cells transfected with mArf1-11-GFP have LD numbers comparable to wild-type non-transfected cells, while non-transfected ARF1 KO cells (and ARF1 KO cells transfected with mArf1-GFP) have far fewer LDs. Do yeast *arf1Δ* cells also contain fewer LDs?

It was reported previously (Gaynor et al., MBoC 1998) that *arf1Δ* yeast cells contain more lipid droplets. We confirmed this data by TEM and LipidTox and showed that TAG levels are elevated in the *arf1Δ* and *arf1-11*. The reason for the difference in LD numbers between yeast and mammalian cells, might be due to transport of the lipids into peroxisomes in yeast and into mitochondria in mammalian cells. In yeast, import and β -oxidation in peroxisomes is perturbed when yArf1 is not functional or hyperactive. Therefore, the transport is blocked at two stages in yeast. In mammalian cells, in the absence of Arf1, efflux of TAGs into mitochondria might not be blocked, but the hyperactive mutant may have a negative effect.

2. Similarly, what do LDs, mitochondria, and ER-mitochondria/LD-mitochondria contact sites look like in *arf1Δ*ARF2 yeast? TAG levels? ATP production?

In *arf1Δ* cells, we did not observe an increase of contact sites, indicating that the presence of Arf1 activity is needed for contact site formation and/or maintenance. As pointed out above, we did observe an increase in TAGs but no defects in ATP production compared to the wild-type. These data are also now included into the manuscript. Taken together, our data indicate that Arf1 needs to cycle between the 'ON' and 'OFF' state to fulfil its function at least on LD, since the absence of Arf1 and the hyperactive mutant show the same phenotype in terms of LD number and increase of cellular TAG. These findings are also supported by the data obtained with Arf1 constitutively active and dominant negative versions.

3. Based on prior literature, one would guess that COPI is the critical Arf1 effector for lipid metabolism based on its established role with lipid droplets. Does COPI distribution change in *arf1-11* cells? If not, can the authors identify

any Arf1 effectors that have altered localization and/or function in *arf1-11* cells (either at permissive or restrictive temperatures)?

We determined the number of lipid droplets in three temperature sensitive mutants of coatamer subunits, all of which are defective in retrograde transport from the Golgi to the ER. We did not see a consistent increase in LD number in these mutants. Only *sec27-1* appeared to have an effect while *sec21-1* and *ret1-1* had none. Therefore, we conclude that the phenotype we observe in *arf1-11* is not strictly linked to coatamer function. Our findings are consistent with data in mammalian cells (Takashima et al. Cell Struct. Funct. 2011), in which the authors established that coatamer is required for lipid homeostasis but that knockdown does not yield defects in ATGL-mediated lipolysis.

We would also like to point out that in a previous study we showed that the function of Arf1 on mitochondria morphology was independent of coatamer in yeast and in *C. elegans* (Ackema et al., EMBO J. 2014).

We also provide now experiments in which we show that the localization of two COPI components (Sec21 and Sec27) are only slightly altered in *arf1-11*, indicating that the mutant does not greatly change COPI distribution.

In an attempt to identify potential Arf1 effectors, we concentrated on lipid droplet components involved in lipolysis and the conversion of TAG into acyl-CoA. We found the Arf1-11 interacts more strongly with Tgl4 and Faa1 than Arf1. These data suggest that those proteins could be potential effectors and that Arf1-11 may negatively regulate their function. Given these findings, we wondered whether Arf1 could also interact with TAG synthesis proteins. Indeed, Dga1 was pulled down with Arf1. Moreover, we found that Arf1-11 also interacted with the peroxisomal protein Pex13. This interaction takes place most likely in *trans* because using high resolution imaging, we provide now evidence that Arf1 is juxtaposed to peroxisomes but not directly located onto peroxisomes, at least not under the conditions tested. All these data are now included into the manuscript.

4. Why were the membrane potential measurements performed at 23°C and 30°C, while other experiments (including other metabolism measurements) were done at 26°C and 37°C?

We apologize for the lack of clarity. *arf1-11* does not grow on glycerol containing plates at 30°C, while on YPD plates (glucose containing plates) the mutant still grows well. Therefore, we felt we should not do the membrane potential measurements at 37°C but only at 30°C. Since *arf1-11* was described as a ts-mutant at 37°C (Yahara et al., 2002), we routinely perform experiments after shift to the restrictive temperature 37°C.

Minor point

5. The figure legends for Figure 5 do not match the figure.

We apologize for the mistake, which was fixed.

Reviewer #2 (Remarks to the Author):

This study investigates how Arf1 regulates mitochondrial shape and function, which has been suggested by previous work from this group and others. It argues that Arf1 is required for mitochondrial fission and fusion and that Arf1 regulates beta oxidation by controlling the movement of fatty acids out of lipid droplets to sites of beta oxidation in mitochondria and, in *S. cerevisiae*, peroxisomes. These are fascinating new ideas about the functions of Arf1 beyond vesicular trafficking. However, it remains unclear how directly Arf1 is regulating mitochondrial dynamics or fatty acid oxidation and a good deal more mechanistic insight is necessary for this study to be appropriate for NCB. Here are the major issues.

Thank you for the positive assessment of our findings. We now provide more mechanistic insights about how Arf1 regulates fatty acid oxidation and acyl chain transfer to mitochondria. More specifically, we now show that Arf1 on LD can interact with two enzymes involved in the conversion of TAG into acyl-CoA, the substrate that is imported into peroxisomes. Given that TAG accumulated in $\Delta arf1$ and *arf1-11* cells, we surmise that cycling of Arf1 between the active and inactive form is necessary for functional conversion mediated by Tgl4 and Faa1. In addition, Arf1 appears to interact in *trans* with the peroxisomal protein Pex13, suggesting a potential role in contact site formation/regulation between lipid droplets and peroxisomes. Moreover, we followed the reviewer's suggestion to immobilize Arf1 on different compartments to assess the effect on Red-C12 transfer and/or mitochondrial morphology. These results of these experiments were all consistent with the notion that Arf1 regulates the transfer of acyl chains from lipid droplets into peroxisomes and from there into mitochondria in yeast and from lipid droplets into mitochondria in mammalian cells. Finally, we show that wild-type Arf1 is present at contact sites between LDs and peroxisomes, LDs and mitochondria, peroxisomes and mitochondria, and at triple junctions involving all three organelles. We hope that these results and the other experiments we performed at this and the other reviewers' request will satisfy this reviewer's comment about mechanistic insights.

1. It is not clear that Arf1 directly regulates mitochondrial fission and fusion. The study suggests that Arf1 on

membranes near mitochondria somehow regulates mitochondrial dynamics, which is consistent with the finding that Arf1-GFP is next to about %80 of fission and fusion events (Fig 1 and movies). However, wild-type Arf1 is mostly on the cis- and trans-Golgi, which is not known to directly participate in mitochondrial fission or fusion. In addition, Arf1 does not need to be on the Golgi to regulation mitochondrial dynamics since they are still normal when Arf1 is on the ER (Fig. 3C-E). It is not clear how Arf1 could directly regulate mitochondrial dynamics from different organelles. Some mechanistic insight into how this occurs is necessary.

We do not only see Arf1-11 close to/on mitochondria, but also wild-type Arf1. The bulk of WT Arf1 is on the Golgi, and this is undisputed, but we also know that Arf1 is on endosomes, on COPI vesicles, on lipid droplets and interacts with peroxisomes from numerous studies from many labs. So, it would not be surprising that a part of Arf1 could also be on or close to mitochondria. We acknowledge that there is now a string of papers implicating Arf1 in mitochondrial fission through Golgi-derived vesicles in mammalian cells (the first being by Nagashima et al. Science 2020). However, there are also a number of papers calling for a more direct, or Golgi-independent, involvement of Arf1 in mitochondrial dynamics and function, such as Wang et al Nat Comm 2020 and Walch et al., Sci Rep. 2018., and also our own previously published data (Ackema et al., EMBO J. 2014).

To address the reviewer's point more directly and experimentally, we anchored Arf1 and Arf1-11 onto the ER, mitochondria and lipid droplets. Anchoring Arf1 onto the ER slightly increased mitochondrial dynamics as measure by fission and fusion events at 23°C and 37°C. More importantly, mitochondrial dynamics were rescued in *arf1-11* at 37°C. In contrast, immobilizing Arf1 or Arf1-11 on lipid droplets or mitochondria abolished mitochondrial dynamics. Thus, our data indicate that Arf1 is able to work in 'trans' on the ER to regulate mitochondrial dynamics. We assume that the continuous presence of Arf1 on mitochondria and LDs has a toxic effect, which is also supported by the phenotype of the constitutively-active and dominant-negative versions of Arf1. At least in mammalian cells, Arf1-11 is present on both LDs and mitochondria, consistent with a direct involvement of Arf1 in regulating mitochondrial fission and fusion.

Finally, we generated a strain in which we simultaneously can visualize Arf1, lipid droplets, peroxisomes and mitochondria. Our data indicate that Arf1 is present at contact sites between these organelles. Moreover, this finding is conserved from yeast to mammals. Thus, it is not only the hyperactive version of Arf1, which is involved in the metabolism and transfer of TAGs out of LDs down to mitochondria but also wild-type Arf1 can do this. Moreover, our data provide strong evidence that it is not Arf1 on the Golgi but rather Arf1 on LDs, on the ER and possibly also on mitochondria, which is required for the FA flux and for the maintenance of functional mitochondria in yeast and mammalian cells. The ER and mitochondria localization of Arf1, presumably at contact sites, would be able to regulate mitochondria fission and fusion. We would also like to point out that we have shown previously that Arf1's function in regulating mitochondrial dynamics is independent of its function in COPI mediated transport at the Golgi (Ackema et al., EMBO J. 2014).

Thus, we provided strong evidence that it is not primarily the Arf1 Golgi pool that regulates mitochondrial dynamics in our experimental systems.

2. The study suggests that Arf1-11 is sequestered on lipid droplets (and other membranes) at non-permissive temperature, rendering it inactive. However, it is also possible that Arf1-11 on lipid droplets is toxic. If this is correct, Arf1-11 will disrupt mitochondrial dynamics even when wild-type Arf1 is present. This should be tested.

We respectfully disagree with the reviewer that Arf1-11 is inactive, but rather more active than wild-type Arf1. Still, we liked the idea of putting WT-Arf1 on LDs to test toxicity.

To the first point: we actually provide evidence that Arf1-11 is a gain-of-function mutant form of Arf1. We showed already in the previous version of the manuscript that Arf-11 binds more strongly to the GGA2-GAT, which specifically binds the activated form of Arf1. Likewise, the constitutive active Arf1^{Q71L} was also enriched on the ER, while the dominant negative Arf1^{T31N} was not. Also, immobilization of Arf1 on the ER enhanced mitochondrial fission and fusion, similarly to Arf1-11 at 23 °C (the permissive temperature). Still Arf1-11 is not dominant over Arf1 and does not disrupt mitochondrial dynamics in the presence of wild-type Arf1. Thus, all experiments are in agreement that Arf1-11 is a hyperactive Arf1 mutant. We also wanted to do biochemical GTPase assays to further corroborate our data. However, we failed to purify Arf1-11 or Δ N17Arf-11 (w/o the N-terminal amphipathic helix) in sufficient purity and amount from *E. coli* to perform such assays, while Δ N17Arf1 was very well expressed and readily purified. Therefore, we could not measure the enzymatic activity of Arf1-11.

To the second point:

We anchored Arf1 onto lipid droplets by replacing the N-terminal 17 amino acid by the amphipathic helix of the PAT domain of Pln1/Pet10 (Gao et al., JCB 2017)). Sequestering Arf1 on LDs caused fragmentation of mitochondria, indicating that the continuous presence of Arf1 on LDs might indeed be toxic for the cell. Thank you for this suggestion! These data are now included in the manuscript.

See also comments to point 2 of reviewer 3

3. Much of the evidence that Arf1 regulates TAG mobilization out of lipid droplets is indirect. Rates of fatty acid mobilization should be directly measured. One way to do this is to treat cells with cerulenin and terbinafine (for example, see PMID: 16135509) and measure TAG and SE levels over time. This will make it possible directly determine rates of mobilization in cells expressing Arf1 or Arf1-11. It might be useful to attach a degron to Arf1 to rapidly remove it from cells to see whether changes in TAG mobilization are specific to Arf1-11.

We agree with the reviewer that we did not provide the mechanism on how Arf1 regulates export of TAGs from LDs in the initial submission. However, we do not see how the suggested experiment will provide us the answer. In *arf1-11* cells, the peroxisomal transporter Pxa2 and the first and rate limiting enzyme of the β -oxidation cascade Pox1 are down regulated. Mutants in either *POX1* or *PXA2* have already been shown that have defects in TAG immobilization (Thompson and Trinh, Biotechnol. Bioeng. 2014; Valle-Rodriguez et al., Appl. Energy, 2014). We feel that we would be unable to distinguish between effects from the lack of Pxa2 and Pox1 and a potentially more direct role of Arf1.

Still, to provide more insights into Arf1's role in TAG immobilization, we asked whether Arf1 would interact with enzymes involved in TAG synthesis and conversion on LDs. We found that Arf1 interacts with Dga1, the last enzyme in TAG biosynthesis, with the triglyceride lipase Tgl4 and the acyl-CoA synthase Faa1. Together with the findings that *arf1-11* and Δ *arf1* cells accumulate TAGs, our data suggest that Arf1 regulate FA efflux on two levels: Arf1 positively impacts TAG synthesis and negatively regulates FA efflux, and that Arf1's cycling between the ON- and OFF- state might be important for proper regulation of TAG mobilization.

4. For this study to be appropriate for NCB, some mechanistic insight into how Arf1 regulates neutral lipid mobilization is necessary. How does Arf1 regulate levels of Pxa2 and Pox1? Does down regulation of these proteins, by itself, reduced TAG mobilization? If Arf1 regulates TAG mobilization or fatty acid transport in other ways, how does it do so? A complete answer to these questions beyond the scope of this study, but some hint of mechanism is necessary to support the claim Arf1 is directly involved.

We provide now evidence that Pxa2 and Pox1 levels are regulated post-transcriptionally. It has already been shown previously that mutants in *PXA2* and *POX1* have TAG mobilization defects. Since both proteins are downregulated in *arf1-11* cells, it will be difficult to address whether there is another pathway contributing to the defect. See also the response to the point above.

It has been assumed that *POX1* and *PXA2* levels are controlled transcriptionally by Msn2/Msn4 (Rajvanshi et al. JBC, 2017) and *PXA1* and *PXA2* by Adr1 (Shani et al., PNAS 1995, Verleur et al, Eur J. Biochem, 1997, Harbison et al., Nature 2004). Therefore, we performed RT-qPCR to determine the RNA of *POX1* and *PXA2* in wild-type and *arf1-11* cells. We did not find any difference in RNA levels, indicating that the downregulation of Pox1 and Pxa2 occurs post-transcriptionally.

We also wanted to test whether Pox1 and Pxa2 would be degraded by the proteasome system in *arf1-11*. The most common way to inhibit proteasomal activity is by adding MG132. The use of this inhibitor in yeast is difficult because MG132 does not readily pass through the cell wall into the cell. People typically use special media involving 0.003% SDS, 0.1% proline and delete Pdr5 and Erg6 (Lee et al., J. Biol. Chem 1996; Marshall et al., 2016; Liu et al., Biotechniques 2018). However, given the circumstance that we want to look at metabolic enzymes and metabolism, this treatment appeared to be very harsh and not appropriate for our purpose. Therefore, we can only conclude that the downregulation of Pox1 and Pxa2 does not occur through the previously identified transcriptional pathways, but that it occurs post-transcriptionally and we speculate that it occurs by proteasomal degradation. We find it unlikely that pexophagy would be involved because we did not observe any decrease of Pex3 or Pex11 foci in *arf1-11* cells compared to *ARF1* cells.

Arf1 has been proposed to bind to peroxisomes (Passreiter et al., J. Cell Biol 1998; Lay et al., J. Biol. Chem 2005), and to interact with Pxa1 and Pxa2 (Snider et al., Nat. Chem. Biol. 2013). To get more mechanistic insights on how Arf1 affects peroxisomal function, we revisited this issue and performed super resolution microscopy of Arf1 and a peroxisomal marker in collaboration with Einat Zalckvar at the Weizmann Institute in Israel. We found that Arf1 is localized mostly juxtaposed to peroxisomes and not directly on peroxisomes, indicating that Arf1 might be mostly acting in *trans*, presumably from lipid droplets or the ER. We can also not exclude a contribution by mitochondria. Others had already reported that Arf1 can bind to peroxisomal proteins. We confirmed these results by showing that Arf1 and Pex13 could be co-immunoprecipitated. Finally, Arf1 is present at contact sites between LDs and peroxisomes in yeast and in mammalian cells.

Taken these findings together, we provide strong evidence that Arf1 has a regulatory function in the conversion of TAG into acyl-CoA and its import into peroxisomes.

5. To test the idea that Arf1 on the surface of LDs regulates neutral lipid mobilization from LDs, rates of mobilization should be determined when Arf1 is artificially targeting to LDs or anchored away from the LDs.

To address this point, we first anchored Arf1, Arf1Q71L and Arf1T31N onto LDs. Immobilization of Arf1 on LDs already drastically reduced Red-C12 efflux into mitochondria. The phenotype remained roughly the same for Arf1Q71L and Arf1T31N, indicating that the continuous presence of Arf1 has a negative effect. In contrast, when we anchored Arf1 and Arf1-11 onto the ER and measured Red-C12 arrival in mitochondria, the Red-C12 efflux was not perturbed. These data are consistent with a role of Arf1 negatively regulating Red-C12 efflux from LDs into mitochondria. We cannot exclude the possibility that at least some Red-C12 could be transferred from LDs to mitochondria directly. To investigate whether Red-C12 can bypass peroxisomes to get to mitochondria, we deleted both PEX3 and PEX19 which abolishes all peroxisomes (Hettema et al., EMBO J. 2000). Under these conditions, we still observed transfer of Red-C12 to mitochondria. Hence, at least some Red-C12 can be transferred to mitochondria in the absence of peroxisomes.

Our finding that Arf1 interacts with the LD proteins Dga1 (production of TAGs), Tgl4 (lipolysis) and Faa1 (acyl-CoA synthesis), is consistent with the idea that Arf1 on LDs negatively regulates the conversion of TAGs into acyl-CoA. Arf1 is on LDs is juxtaposed to peroxisomes. The down-regulation of Pxa1 and Pxa2 on peroxisome in *arf1-11* cells impairs import of acyl-CoA into peroxisome and the concomitant down-regulation of Pox1, the first and rate-limiting enzyme in β -oxidation inhibits the production of acetyl CoA. Thus, Arf1 appears to be strongly involved in the regulation of TAG metabolism at several levels.

6. The idea that rates of TAG mobilization significantly alter mitochondrial shape is not well supported. Fig S7C shows that cells expressing wild-type Arf1 have similar mitochondrial shapes whether they do or not produce TAG.

We agree with the reviewer that the TAG levels per se may not cause toxicity but that the continued presence of Arf1 may negatively regulate the conversion and efflux TAGs and thereby compromise mitochondrial activity and function. Our new data on the interaction of Arf1 with Tgl4 and Faa1, LD proteins required for the formation of acyl-CoA suggest that Arf1 might negatively regulate the activity of the proteins and this lack for transfer of acyl chains into peroxisomes is a major reason for the defects we observed. This pathway seems to be particularly important because in *arf1-11* cells also Pxa1 and Pxa2, the import machinery of acyl-CoA is down regulated. Our data suggest that the metabolism of neutral lipids is highly regulated and that this process involves Arf1 as a key component. Consistent with this notion is our finding that deletion of *POX1* by itself alters mitochondria shape. Of note, Pox1 is also downregulated in *arf1-11* cells.

7. Minor: The data in Fig 5E should be presented as bar graphs as in Fig. 5D,F. Whether differences are statistically significant should be determined.

Fixed

Reviewer #3 (Remarks to the Author):

The manuscript by Enkler et al. reports on mechanisms by which cells coordinate fatty acid distribution between different organelles. The paper presents a number of unexpected results, resolves a major inconsistency in the field and potentially unravels an interesting new mechanism in molecular metabolism. The manuscript is timely and of great interest to a broad readership in the cell biology community. Pending the clarifications requested below the novelty, originality and importance of the suggested mechanism would make the paper an excellent candidate for publication in NCB.

Thank you very much for this positive assessment!

Summary of key results

Enkler et al. focus specifically on the role of Arf1, a small monomeric GTPase, in organizing FA transport between lipid droplets (LDs) and peroxisomes or LDs and mitochondria in yeast and in mammalian cells, respectively. Arf1 is classically associated with coordinating vesicular membrane traffic events at the Golgi complex, but has recently been shown to have additional importance for the regulation of mitochondrial fusion and fission. The underlying mechanisms are, however, not completely understood.

Arf1 is mostly localized on the Golgi complex both in yeast and mammalian cells, but a sub-fraction is targeted to the mitochondria where it is involved in maintaining mitochondrial morphology. The authors have now clarified that Arf1-11 (a yeast Arf1 mutant) is a gain of function or a hyperactive Arf1 version, rather than a loss of function allele as previously assumed. This is an important finding because it reconciles conflicting results between yeast and mammalian cells. The new data clearly establish that loss of Arf1 function causes elongation of mitochondria (which was previously known from work in mammalian cells) and gain of function mutants shorten mitochondria.

The authors find that the hyperactive GTP-bound Arf1-11 is found less at the Golgi, but targets in yeast to the ER

at permissive temperature and localizes to LDs at 37°C. In mammalian cells an Arf1 version containing the point mutations from yeast Arf1-11 is found mostly on LDs and mitochondria. The new localization of the mutant leads to fission of mitochondria. The authors suggest that this change in shape is not a direct effect of Arf1 on the fusion/fission machinery but is rather explained by defects in lipid and energy metabolism.

When Arf1-11 localizes to LDs the number of LDs increase and the cells contain higher amounts of triglycerides (TGs), which are stored in LDs. According to the authors, TGs accumulate because Arf1-11 blocks mobilization of FAs from LDs, and inhibits beta oxidation in peroxisomes of yeast cells or in mammalian mitochondria. In both cases Arf1-11 reduces acetyl-CoA levels, and decreases ATP production. How lower ATP levels lead to fission of mitochondria is not yet clear and is possibly a good starting point for a next study.

Comments on data and methodology

The manuscript is complex, but clearly written. The most convincing part are the data in figures 1 and 2 showing that Arf1 localizes to sites where mitochondria undergo fusion and fission. Especially the fission part is clear. It is also convincing that arf1-11 is GTP bound. It would have been helpful to add GTPase assays to show that the protein is GTPase deficient, but perhaps this was already done in the literature. Are mitochondria globular in Arf1 Q71L? This was not really clear to this reviewer. Can the authors please clarify these points?

We agree with the reviewer that a GTPase assay would have been nice. Arf1 has however a very low intrinsic GTPase activity. Therefore, these GTPase assays require also the addition of GTPase activating proteins, and hence somewhat more complicated to set up. These assays are certainly not routinely done for yeast Arf proteins. On the other hand, the GGA-pulldown is a very simple and easy assay that is well established and which provides a very good assessment of the GTP-bound state and it does not require the purification of native Arf1. Even a number of companies sell this type of pulldown kits to check for Arf1 activity.

Still, we tried to accommodate the reviewers request and set out to purify Arf1-11 (or Δ N17-Arf1-11) from *E. coli*. Although wild-type Arf1 (in particular, Δ N17-Arf1) expression was good, we failed to purify significant amounts of Arf1-11 or Δ N17-Arf1-11, and hence were unable to perform in vitro-GTPase assays.

We do not think that Arf1-11 lacks GTPase activity entirely because then it should behave like a dominant/constitutive-active mutant, which it does not. In the presence of a wild-type copy of Arf1, there is no temperature-sensitivity observed. We assume that Arf1-11 has just a lower GTPase activity perhaps because it cannot bind as efficiently to its GAP as the wild-type. To figure out the precise mechanism would be a project on its own and beyond the point of this manuscript.

We expressed the constitutively-active (Arf1Q71L) and the dominant-negative (Arf1T31N) in wild-type yeast. While we observed no effect on mitochondria in Arf1Q71L expressing cells, mitochondria were fragmented in Arf1T31N cells. We take this as an indication that we need actually do need a cycle of Arf1 activation and inactivation for the observed phenotype in *arf1-11* cells. Consistent with this notion, sequestering Arf1 onto lipid droplets, caused mitochondrial fragmentation to a similar extent than Arf1-11 on LDs. Also, immobilizing Arf1 on the ER or mitochondria phenocopied the *arf1-11* phenotypes.

The data in figure 3 is convincing, but with respect to the main message in the paper (focusing mostly on the role of Arf1 in mitochondrial fission) it is a negative result. Would it be possible to add a lipid droplet binding domain to yArf1? This should have similar effects as the Arf1-11 mutant and would strengthen the point that the localization to the LDs is important for the mitochondrial fission. Is there a specific domain in *erg6* which the authors use as LD marker that can be moved to yArf1? This LD localized mutant should lead to globular mitochondria. This or an equivalent experiment should be added.

Thank you for this great suggestion! We sequestered yArf1 on LDs, and as the reviewer predicted, this caused mitochondrial fragmentation. We likewise sequestered yArf1 and yArf1-11 mitochondria. Under these conditions we also observed mostly globular mitochondria.

See also comments to point 2 of reviewer 2.

The experiments in figure 4 are technically elegant. It is, however, not easy to understand why the localization of Arf1-11 is so different between panel B and F. In theory this should be the same conditions. How often can the localization along the mitochondrial profile be observed? Some quantification would be helpful, e.g. % of cells with Arf1-11 on long mitochondria. Conceptually, it is not clear why the mammalian mitochondria do not undergo fission when Arf1-11 is expressed. The authors should clarify this point.

We get different phenotypes and we chose the localization patterns closest to the organelle tested for co-localization. We now provide a quantification of the phenotypes in Suppl. Data.

Image mitochondrial dynamics in mammalian cells with Arf1 and Arf1-11 in ARF KO + empty vector control

The EM images in figure 5 suggest that γ Arf1-11 cells contain much more internal membranes than controls. It seems however that this is not reflected in the lipid analysis, and the membrane lipids stay unchanged. Are the sphingolipids increasing? The methods section indicates that sphingolipids were analyzed, but the data is missing from this version of the paper. Please add the data or remove the text from the methods. The internal membranes appear quite white, which might indicate that they are rich in TG. The authors might want to point this out and compare the results to EM images from Pex30 complex knockout cells where similar observations were made.

We thank the reviewer for pointing out a mistake. In fact, we did not measure sphingolipid levels and we removed this section from the manuscript.

As for the Δ pex30 mutant and LD accumulation and internal membranes. Joshi et al. (Nat Comm. 2018) analyzed Δ pex30 mutants by EM and also determined TAG levels. In Figure 3, the authors show smaller LDs and that they appear to be tethered to the ER. We agree that there are more internal membranes in Δ pex30 cells; they are not whitish though. Also in the same paper, they reported that TAG levels were decreased in Δ pex30 cells, which is the opposite of what we observed in both *arf1-11* and Δ *arf1* cells. Because the phenotypes of Δ pex30 and Δ *arf1* or *arf1-11* are not the same, we did not mention the Δ pex30 phenotype at this point. However, if this reviewer feels strongly about the published data on Pex30, we are happy to discuss them.

in general, the figure legends in figure 5 seem to come from a different version of the paper, which complicates the assessment of the data. This should be corrected. The figure legends for C are completely missing and the units for the box plots are not clear. What does % of Hela mean and how are the values normalized? Panel E has to be presented in the same format as panel D and F (bar graphs with confidence intervals). It is not clear how the log2 fold change was calculated, the relevant methods are incomprehensible (what is sample 1?). This is a key result in the paper and its presentation should be improved. The quantification in figure H and I is confusing. The figure legend is difficult to understand, and it is not indicated how many images were analyzed. Please, make the data more accessible to the reader.

We apologize for this omission. This has been amended in the new version of the manuscript.

The yeast data in figures 6 and 7 point at a possible function of beta oxidation in maintaining mitochondrial morphology. Figure 6 indicates that fatty acid beta oxidation in peroxisomes is switched on after heat shock leading to the production of acetyl-CoA for ATP synthesis in mitochondria. In *arf1-11* cells this pathway is blocked. It is, however, not clear why mitochondria undergo fission under these conditions. The authors have to include more controls and show that mitochondria are round after heat shock in cells with knockouts for beta oxidation enzymes (e.g. *Pox1* deletion). This control is necessary, because it is not clear why ATP production would depend on respiration specifically under heat shock. Usually yeast cells ferment and produce all ATP by glycolysis. Why would heat shock make aerobic respiration essential, and why would this ATP pool be necessary for mitochondrial homeostasis?

We do not know whether β -oxidation in general is turned on after shift to 37°C. *Pxa1*, *Pxa2*, and *Fat1* levels are not increased after temperature shift, which would provide the substrate, but then the levels of *Pox1* and *Fox2* are. It is unlikely that this increase in *Pox1* and *Fox2* levels are transcriptionally regulated, as they did not appear to be upregulated in various gene-expression experiments inducing heat shock (SGD; most relevant publication Gasch et al 2000). In fact, we checked in total 37 dataset with varying conditions of heat shock available in SGD. In none of the instances did we observe an upregulation of *Pox1* and *Fox2* while *Pxa1* and *Pxa2* levels remained the same or went down.

However, heat stress causes ROS production, which actually reduces respiratory ATP production because of the oxidation of the thiols. Thus, we do not think that there is an up-regulation of respiratory ATP production, but rather a downregulation.

When we deleted peroxisomes (Δ pex3 Δ pex19) from cells, we did not observe a defect in mitochondrial morphology or Red-C12 transfer. In contrast when we deleted *POX1* or *POT1*, mitochondrial morphology was strongly affected. Thus, our data indicate that in the absence of peroxisomes some bypass mechanism must exist, while non-functional peroxisomes in terms of β -oxidation negatively impact mitochondrial morphology and function. These data are now included into the manuscript.

The title of figure 7 is inconsistent with the data. According to the model in figure 6, acetyl-CoA and not the entire fatty acid should enter mitochondria of yeast. If the entire fatty acids enter yeast mitochondria as suggested in the title and the figure legend, the whole concept is incorrect. The images in figure 7E probably show that the bodypy

dye remains attached to the acetyl CoA after the C12red fatty acids have been oxidized in the peroxisomes and therefore Bodipy-acetyl-CoA stains mitochondria red. If the whole fatty acid reaches the mitochondria, the mechanism that is suggested in figure 6 would be wrong.

We agree with the reviewer that it is most likely Bodipy-acetyl CoA that stains mitochondria. We could not find any data about this in the literature though. Other researcher also used Red-C12 and as far as we are aware we could not find any indication how Red-C12 is modified/metabolized on the way into mitochondria. Nevertheless, we are more careful in the usage of the term 'fatty acid' and aimed to be more precise in expressing what species is transferred.

We still wanted to explore the possibility that when we remove peroxisomes from cells whether Red-C12 could reach the mitochondria. To this effect we deleted *PEX3* and *PEX19*. In this mutant strain, some Red-C12 still reached mitochondria, implying there might be an alternative route. In mammalian cells, it has been shown that some peroxisomal matrix proteins reach mitochondria in the absence of peroxisome biogenesis.

In mammalian cells it is clear that the fatty acids enter into mitochondria from LDs as shown by previous work (Rambold et al. in Dev Cell 2015). It would be interesting and novel if Arf1 is necessary for the relevant transport step as suggested by the authors. Can wild-type Arf1-GFP be seen at the contact sites between C12red LDs and mitochondria. This would be an important result!

We thank the reviewer for the suggestion. We determined the localization of Arf1-GFP in yeast and mammalian cells in conjunction with LDs, peroxisomes and mitochondria. In both experimental systems we found Arf1 to be present at contact sites between LDs and mitochondria, LDs and peroxisomes, peroxisomes and mitochondria and also at contacts involving all three organelles. Thus, Arf1 is at the right place to regulate Red-C12 flux into mitochondria.

In its current form the conclusions from figure 6 and 7 (figures S6-S8) are not robust and the interpretations have to be corrected.

We provide now more data to bolster up our model that Arf1 plays a role in regulating FA metabolism/flux from LDs to peroxisomes and mitochondria and that this role is conserved from yeast to mammalian cells. We show that Arf1 is at the contact site of these organelles. We also show that yArf1 is able to interact with Dga1, Tgl4 and Faa1 on lipid droplets, indicating that Arf1 could potentially regulate TAG synthesis and breakdown on LDs. Moreover, consistent with previous findings, we establish that Arf1 can interact with peroxisomal proteins. However, we think that this interaction is most likely in *trans* because we found Arf1 to be localized mostly juxtaposed to peroxisomes and not on them. Furthermore, we provide now also stronger evidence that Arf1 needs to cycle between the GDP and GTP bound forms -and thereby being only transiently present on LDs- in order to fulfil its regulatory function on LDs as continued presence of Arf1 on LDs was detrimental and impaired mitochondrial morphology.

The current legend for figure 7 likely belongs to a different version of the paper. This should be corrected.

Sorry! The figure legend was corrected.

Decision Letter, first revision:

Dear Professor Spang,

Thank you for submitting your revised manuscript "The small GTPase Arf1 regulates ATP synthesis and mitochondria homeostasis by modulating fatty acid metabolism", to Nature Cell Biology. Thank you also for your patience with the re-review process, I am sorry our decision could not come sooner. The revision has now been seen by the original reviewers, whose comments are pasted below. After detailed editorial discussions of their advice, we regret that we cannot offer to publish the study in Nature Cell Biology.

As you will see, although Rev#1 was fully supportive of the revision, unfortunately Revs#2 and #3 shared persistent concerns -- in particular, Rev#2 was still not convinced that the data strongly support the model that Arf1 affects mitochondrial morphology through a role in the regulation of fatty acid metabolism and beta-oxidation. We found these remaining concerns significant. Unfortunately, we do not have enough reviewer support to move forward with publication. As we make every effort to limit all manuscripts to a single round of major revision and to limit the overall time spent in peer review, we regrettably feel we have to return the manuscript to you for submission elsewhere.

We are very sorry that we could not be more positive about the revision, but we thank you for the opportunity to consider this work. Please also let me know if you would like me to consult with one of my colleagues at Nature Communications or another sister journal about a potential transfer of the study.

With kind regards,
Melina

Melina Casadio, PhD
Senior Editor, Nature Cell Biology
ORCID ID: <https://orcid.org/0000-0003-2389-2243>

Reviewers' comments:

Reviewer #1 (Remarks to the Author):

The authors now provide extensive new data and additional interpretations and discussion. I believe they have greatly improved the manuscript by clarifying their observations and providing some additional mechanistic insights. I am supportive of publication.

Reviewer #2 (Remarks to the Author):

My major concerns about this study remain the same. The study makes a good case that loss of Arf1 function affects mitochondrial morphology and dynamics, which has also been suggested previously.

The revision has stronger evidence that Arf1 on mitochondria and lipid droplets, rather than the Arf1 on the Golgi or vesicles, is responsible for the effect of Arf1 on mitochondria. This is certainly interesting, but there is still not much mechanistic insight into how Arf1 affects mitochondria, and it is not clear Arf1 is directly involved. The study argues that Arf1 effects mitochondria by directly regulating TAG metabolism and beta-oxidation. This is an attractive idea, but the evidence remains unpersuasive.

1. The idea TAG metabolism and beta-oxidation play important roles in regulating mitochondria shape is not well supported by results here or in the literature. Fig. S8BCF shows that cells expressing wild-type yArf1 have modest changes in mitochondria shape when they lack TAG or cannot perform beta-oxidation. Similar results have been found in previous studies using yeast and mammalian cells depleted on TAG or with defects in beta-oxidation. This suggests that defects in TAG metabolism and beta-oxidation probably play only a minor part in the dramatic changes in mitochondrial shape that occur in arf1-11 cells.

2. The study still does not make a strong case that Arf1 plays a direct role in TAG metabolism. The revision now shows that Arf1 interacts with three TAG metabolism enzymes: Dga1, Faa1, and Tgl4. This an important advance, which the study suggests shows that Arf1 directly regulates these enzymes. Unfortunately, there is no demonstration that it does. There are well established assays for all three enzymes that should be used to determine whether Arf1 affects their activities.

3. If Arf1 plays a direct role in TAG mobilization from lipid droplets, then it should be possible to show this by directly measuring rates of TAG hydrolysis in cells. The classic way to do this quantitatively, as I mentioned in my previous review, is to block fatty acid synthesis with drugs like cerulenin, which result in the rapid hydrolysis of TAG. If Arf1 directly regulates Tgl4 (and perhaps other TAG lipases), the rate of TAG hydrolysis should change in cells depleted of Arf1 or following expression of Arf1-11.

4. More work is necessary to show that Arf1 directly regulates TAG synthesis on lipid droplets. Only one of the two major TAG-synthesizing enzymes in yeast, Dga1, is on lipid droplets (and in some growth conditions it is largely on the ER). Lro1 has been shown to be in the ER and contains a transmembrane domain that almost certainly excludes it from the surface of lipid droplets. The different localization of the two primary TAG-synthesizing enzymes should be used to test the hypothesis that Arf1 regulates TAG synthesis on lipid droplets. If Arf1 regulates TAG synthesis in the ER, the model should be changed.

5. The study suggests that Arf1 regulates or facilitates the transfer of fatty acids, acyl-CoA, or acetyl-CoA derived from TAG in lipid droplets to peroxisomes or mitochondria. BODIPY Red-C12 is used to directly measure the transport fatty acids and fatty acid metabolites in cells. However, without evidence that BODIPY Red-C12 can be incorporated into TAG, conjugated to CoA, or undergo beta-oxidation, it is hard to know how to interpret the results. This issue was raised by reviewer 3 and in response the authors wrote: "as far as we are aware we could not find any indication how Red-C12 is modified/metabolized on the way into mitochondria." This needs to be resolved. Since the strongest evidence that Arf1 plays roles in metabolite trafficking come from the experiments with Red-C12, it is critical to confirm that it is metabolized similarly to endogenous fatty acids.

Reviewer #3 (Remarks to the Author):

This is an important paper for the field, with considerable conceptual novelty as outlined below. The authors have addressed most of the technical issues that were raised by this reviewer. However, a central conceptual point is not resolved.

A major finding of this work is that Arf1 is important in controlling mitochondrial fission and fusion. This is by itself an important result because it consolidates various and sometimes contradictory reports in the field. The major conceptual novelty is the cell biological description of the mitochondrial fission phenotype: the authors unequivocally show that localization of Arf1 to lipid droplets leads to mitochondrial fission, whereas an Arf1 version that is permanently anchored to the ER causes the opposite phenotype elongating mitochondria. Additionally, when Arf1 is anchored on the LDs the cells produce significantly more triglycerides. Another important fact is that the role of Arf1 in controlling mitochondrial morphology is conserved between yeast and mammals.

The different spatial organization of fatty acid beta oxidation in yeast and mammalian cells complicates the rest of the paper: it is not clear how the peroxisomes fit into the whole picture. The simplest explanation might be that Arf1 has a role in fatty acid transport at contact sites between various organelles as alluded to by the authors. How beta oxidation in mitochondria of mammalian cells or in peroxisomes of yeast relates to mitochondrial shape changes is not resolved in this work. An overwhelming amount of data is presented which is not easily accessible to the reader and perhaps even clouds the main message of the paper. The authors seem to have issues with explaining their thought process, and the experiments do not tie into a cohesive narrative.

A minimal requirement for acceptance at Nature Cell Biology is that the concept is formulated more clearly. In fact the experiments in FigS9 indicate that the peroxisomes are dispensable for the fatty acid transport to mitochondria. The authors state in line 361 "Abolishing peroxisome biogenesis did not affect the mitochondria phenotype [51], and did not reduce the flow of Red-C12 to mitochondria (Figure S9C) suggesting that under these conditions cells might develop alternative contact sites for efficient FA transfer."

According to this reviewer these results indicate that Arf1 is involved in FA transfer between the ER and mitochondria, which is important for mitochondrial shape. Although Arf1 might indeed be involved in FA transfer from LDs to peroxisome in yeast, this activity might not be part of the mechanisms determining the mitochondrial phenotypes.

A minor point is that Figure 3D and E which is a central result in the paper should be better annotated in the figure legend. It is not clear how D and E relate and the scales in the two panels are not clear since the scale bar is not described in panel D.

**Although we cannot publish your paper, it may be appropriate for another journal in the Nature Portfolio. If you wish to explore the journals and transfer your manuscript please use our manuscript transfer portal. You will not have to re-supply manuscript metadata and files, but please note that this link can only be used once and remains active until used. For more information, please see

our manuscript transfer FAQ page.

Note that any decision to opt in to In Review at the original journal is not sent to the receiving journal on transfer. You can opt in to In Review at receiving journals that support this service by choosing to modify your manuscript on transfer. In Review is available for primary research manuscript types only.

Author Rebuttal, first revision:

University of Basel, Department Biozentrum, Spitalstrasse 41, 4056 Basel

Reviewers' comments:

Reviewer #1 (Remarks to the Author):

The authors now provide extensive new data and additional interpretations and discussion. I believe they have greatly improved the manuscript by clarifying their observations and providing some additional mechanistic insights. I am supportive of publication.

Thank you very much for the positive assessment of our work and for recommending our work for publication in Nature Cell Biology.

Reviewer #2 (Remarks to the Author):

My major concerns about this study remain the same. The study makes a good case that loss of Arf1 function affects mitochondrial morphology and dynamics, which has also been suggested previously. The revision has stronger evidence that Arf1 on mitochondria and lipid droplets, rather than the Arf1 on the Golgi or vesicles, is responsible for the effect of Arf1 on mitochondria. This is certainly interesting, but there is still not much mechanistic insight into how Arf1 affects mitochondria, and it is not clear Arf1 is directly involved. The study argues that Arf1 effects mitochondria by directly regulating TAG metabolism and beta-oxidation. This is an attractive idea, but the evidence remains unpersuasiv.

We thank the reviewer that s/he finds our study interesting. We would like to point out, as we did in the previous rebuttal letter, that the Arf1-11 mutant is hyperactive mutant and not a loss of function. We do think that this is an important point because it is through this hyperactive mutant that we could detect a role of Arf1 on LD.

The reviewer's criticism hinges on the demonstration of the mechanism that Arf1 directly controls TAG metabolism and β -oxidation. The question is on how direct does this have to be? Direct in the strict sense would be that we reconstitute all this in vitro and measure how Arf1 would regulate the individual enzyme activity. Otherwise, one could always claim that the effect of Arf1 on the Dga1, Faa1 and Tgl4 might be indirect. We actually never distinguish in the manuscript between direct and indirect. We only present our data on this issue. It is the reviewer's interpretation, which

may or may not be correct. We assume that it is undisputed that we provide ample evidence that Arf1 is on LDs and affects proteins on both LDs and peroxisomes.

1. The idea TAG metabolism and beta-oxidation play important roles in regulating mitochondria shape is not well supported by results here or in the literature. Fig. S8BCF shows that cells expressing wild-type yArf1 have modest changes in mitochondria shape when they lack TAG or cannot perform beta-oxidation. Similar results have been found in previous studies using yeast and mammalian cells depleted on TAG or with defects in beta-oxidation. This suggests that defects in TAG metabolism and beta-oxidation probably play only a minor part in the dramatic changes in mitochondrial shape that occur in *arf1-11* cells.

We apologize, but we disagree with this reviewer's statement. The changes in shape are significant, also in the presence of wild-type Arf1 situation (quantified in Fig. S8C). There is a statistically significant difference between the WT *ARF1* strain grown at 37°C and the *Δlro1Δdga1* mutant. We observed that 39% of cells showed globular mitochondria in the *ARF1 Δlro1Δdga1* mutant compared to 21% in the WT *ARF1*. In addition, Fig. S8F clearly shows that loss of β-oxidation (in a *Δpot1* or *Δpox1* strain) leads to increased globular mitochondria and. To further emphasize this, we now provide measurements of three independent experiments to Fig. S8F (graph shown here, which we are happy to include into the manuscript):

In contrast, removing peroxisomes altogether in a *Δpex3 Δpex19* mutant has almost no effect on mitochondrial shape. In agreement with these data, RedC12 was efficiently transferred into mitochondria (Fig. S9C) under conditions when peroxisomes are absent. However, removing either of the β-oxidation enzymes (*Δpot1* or *Δpox1*) led to drastic changes in mitochondria phenotype (Fig. S8F and the quantification of the data provided above). The lack of effect on mitochondria morphology in the *Δperoxisome* condition could also be due to the fact that the cells were grown in 2% glucose, which does not require peroxisome function and also no shuttling of FA/FA metabolites into mitochondria for function or cell survival.

Our data also suggest that at 37°C, FA β-oxidation is nevertheless somehow activated/needed in yeast (Fig. 6G-H). We cannot provide any explanation for this finding, but nothing is known in the literature either, except for one study in which the authors have shown that β-oxidation genes were upregulated after 3h of growth at 34°C (Musa et al., Aging 2018). What we do see on the other hand is an exacerbated phenotype on peroxisomal proteins, in FA storage and transfer to

mitochondria in the *arf1-11* mutant at 37°C. This shows that despite being in suboptimal conditions (bound to the thermo-sensitivity of the mutant), a hyperactive mutant of Arf1 influences FA storage, peroxisomal protein levels, FA transfer to mitochondria and subsequently respiratory chain activity. Showing that wild-type Arf1 can be at the same location suggests that Arf1 also influences the same processes.

The link between loss of functional peroxisomes and mitochondria morphology is well documented in the literature. To name a few, loss of mitochondrial function (decreased respiration and/or changes in mitochondrial morphology), is widely recognized as Zellweger syndrome disorders (ZWD) phenotypes (Shinde *et al.*, 2018; Argyriou *et al.*, 2019). Mutants of PEX19 (involved in peroxisome matrix protein import and peroxisome biogenesis) in a *Drosophila* model and PEX19-null patient fibroblasts showed swollen mitochondria (Bülow *et al.*, MBoC 2017). Based on a mouse model for mild Zellweger syndrome in which PEX1 bears the G844D mutation, skin fibroblast cell lines showed a normal level of peroxisomes even though these cells had higher levels of VLCFAs, lower oxidation, and had an important defect in matrix protein import, pointing towards non-functional peroxisomes. In these cells, mitochondria morphology was also affected (Argyriou *et al.*, Exp Eye Res 2018; Hiebler *et al.*, Mol Genet Metabo. 2014).

Unrelated to ZWD, it has also been shown that VPS13D can be recruited to peroxisomes and mitochondria by Miro1/2, and its absence leads to heterogenous loss of peroxisomes in cell population, which correlates with globular mitochondria (Baldwin *et al.*, JCB 2021). In the same paper, they could not phenocopy the globular mitochondria in cells knocked-out for peroxisomes “we do not find evidence of the severely rounded mitochondria characteristic of VPS13D KO in the PEX-KO HeLa cells completely lacking peroxisomes”. Additionally, Seung *et al.* showed that mutations in VPS13D affects mitochondria morphology in *Drosophila* nervous system and in fibroblasts from families showing recessive ataxia (Seung *et al.*, Ann Neurol 2018).

We do acknowledge that the link between loss of TAG metabolism or β -oxidation cannot always be linked to loss of mitochondria morphology, and these different findings are highly discussed in the field. However, this has to be accounted to differences of the cell-types and/or organisms investigated. For instance, liver in *PEX5*^{-/-} mice used as ZWD model mitochondria showed strong morphological alterations and loss of Complex 1 activity, whereas these alterations were less severe in heart. Mitochondria also had normal appearance in fibroblasts in primary cultures of *PEX5* knockout mice, or *in situ* in various organs; complex I activity was not altered in fibroblast cultures of these mice (either immortalized fibroblasts or primary cultures) (Baumgart *et al.*, American Journal of Pathology 2001). Similarly, loss of peroxisomes does not always lead to mitochondria phenotype. Some studies have shown that the loss of peroxisomes in mouse serotonergic neurons and *Drosophila* Malpighian tubules resulted in the enlargement of mitochondria (Rahim *et al.*, 2016; Bülow *et al.*, 2018). A recent study also reported that human patient-derived fibroblasts lacking Pex3 did not exhibit changes in mitochondrial morphology (Sugiura *et al.*, 2017) while deletion of PEX3 or PEX5 lead to mitochondria fragmentation in mouse embryonic fibroblasts (Tanaka *et al.*, JCS 2019). Another example is X-linked adrenoleukodystrophy, which is the most frequent peroxisomal disorder. It is caused by disruption

of the peroxisomal ABC transporter ABCD1 (Bezman et al. 2001), which is involved in the import of VLCFAs into peroxisomes (van Roermund et al. 2011). Indeed, mitochondrial structural alterations in adrenal gland cells were reported in 12-month-old X-ALD mice (McGuinness et al. 2003). Although these findings were not corroborated for skeletal muscle mitochondria (Oezen et al. 2005), more recent studies suggested that mitochondria may be compromised in neuronal tissues of ABCD1 knockout mice and X-ALD patients due to elevated levels in VLCFAs (Hein et al. 2008; Galino et al. 2011; Lopez-Erauskin et al. 2013).

Taken together, whether or not loss of peroxisomes affects mitochondria morphology appears to be dependent on the tissue/cell type under investigation. The phenotype is probably also connected to the metabolic state of the cell and the environment in the tissue context. For our yeast experiments, we used 2% glucose in the medium, a condition under which FA do not need to be metabolized by yeast cells as there is glucose in high abundance. This might explain why the mitochondria appear normal when peroxisomes cannot be formed at all. However, when peroxisome biogenesis is only partially affected, we observe major mitochondria morphology defects, indicating that the presence of non-properly functioning peroxisomes is worse than their absence altogether and that mitochondria are affected by perturbed peroxisome biogenesis. Importantly, our data also reveal that under conditions under which β -oxidation is required, mitochondria morphology is highly affected.

2. The study still does not make a strong case that Arf1 plays a direct role in TAG metabolism. The revision now shows that Arf1 interacts with three TAG metabolism enzymes: Dga1, Faa1, and Tgl4. This is an important advance, which the study suggests shows that Arf1 directly regulates these enzymes. Unfortunately, there is no demonstration that it does. There are well established assays for all three enzymes that should be used to determine whether Arf1 affects their activities.

The experiment that Reviewer 2 is now asking is new, and is meant to answer a statement that we do not have in our manuscript as we never claimed that Arf1 directly regulates these enzymes. We do not, however, exclude this possibility. We do agree with the reviewer that it might be indirect through interaction with (an)other factor(s).

There is a published protocol from the group of Günther Daum from 2002 using microsomes and radioactively labelled lipid. However, in our case the experimental setup is non-trivial and we expect to face major problems. In order to provide the requested data, we would need to purify Arf1, Dga1, Faa1 and Tgl4, reconstitute them into LDs and then measure the corresponding activities. While Tgl4 has been purified in the past by Günther Daum's group, and we can purify myristoylated Arf1, setting up these assays with reconstitution on LDs is complicated and not something that can be just easily done. We are not aware of any reconstitution that we could just simply emulate.

We do acknowledge that we write in the discussion section that Arf1 interacts with Dga1 (lines 471-473), and with Tgl4 and Faa1 (lines 478-479), but this is through IP and we all know that this

does not allow to distinguish between direct or indirect interactions. If the reviewer insists, we can be more explicit in stating that the IP does not allow us to distinguish between direct and indirect effects of Arf1 on Dga1, Faa1 and Tgl4 and that the influence of the activity of these enzymes is speculative (we should be allowed to speculate in the discussion.)

Finally, we would respectfully point out that this reviewer said in his/her previous review: 'How does Arf1 regulate levels of Pxa2 and Pox1? Does down regulation of these proteins, by itself, reduced TAG mobilization? If Arf1 regulates TAG mobilization or fatty acid transport in other ways, how does it do so? **A complete answer to these questions beyond the scope of this study**, but some hint of mechanism is necessary to support the claim Arf1 is directly involved.'

- We exclude transcriptional regulation of Pxa2 and Pox1 levels by Arf1, pointing to post-transcriptional regulation.
- It has been shown previously that lack of Pax2 and Pox1 reduces TAG mobilization, and we refer to this study in the manuscript.
- We find that in addition to the post-transcriptional regulation of Pax2 and Pox1 levels, Arf1 interacts as shown by co-IP with Dga1, Faa1 and Tgl4, indicating that there might be an additional way by which Arf1 regulates TAG levels.

We provide answers to all three questions raised by the reviewer and suggest a possible mechanism. We would like to emphasize again that we do not state direct interaction because we have not shown that. We agree with the reviewer, however, that this is a possibility.

3. If Arf1 plays a direct role in TAG mobilization from lipid droplets, then it should be possible to show this by directly measuring rates of TAG hydrolysis in cells. The classic way to do this quantitatively, as I mentioned in my previous review, is to block fatty acid synthesis with drugs like cerulenin, which result in the rapid hydrolysis of TAG. If Arf1 directly regulates Tgl4 (and perhaps other TAG lipases), the rate of TAG hydrolysis should change in cells depleted of Arf1 or following expression of Arf1-11.

Reviewer 2 already raised this point in the first revision (**point 3**), wanted us to measure the synergy of TAG mobilization and levels of peroxisomal protein in *arf1-11* (**point 4**), and see how Arf1 could regulate TAG mobilization by artificially anchoring Arf1 on LD (**point 5**).

In response to these points, we showed that peroxisomal proteins were already downregulated in *arf1-11* (**Fig. 6F-H**) and it had already been shown in the literature that the absence of these proteins impairs TAG flux (Ferreira, R., et al., *Metabolic engineering of *Saccharomyces cerevisiae* for overproduction of triacylglycerols*. *Metab Eng Commun*, 2018. **6**: p. 22-27). Reviewer 2 also acknowledged that a complete understanding of Arf1 role on TAG mobilization and its effect on peroxisomes and mitochondria was beyond the scope of this manuscript. So, there was nothing we could have learnt from these experiments. Finally, we also provided new data to test the hypothesis that Arf1 regulates TAG levels in LD by anchoring Arf1, Arf1-11 and Arf1-CA on LD as suggested by Reviewer 2. This led to an increase of cells bearing globular mitochondria irrespective of the construct tested (**Fig. 3C**).

To assess the **point 3** of the reviewer, there will be technical issues to consider. Although we think that testing TAG mobilization in *ARF1* and *arf1-11* strains after shift to 37°C for 30 min to 1h can be done, we are not sure that cerulenin treatment will be sufficient to see relevant differences within the time of our experimental setting. If one looks at **Fig4B, D, F, H** of the Athenstaedt and Daum paper (Athenstaedt and Daum, JBC 2005), we see that TAG mobilization assessment can be challenging before 2h of cerulenin treatment (deletion of *TGL4* led within the first hour of cerulenin treatment only to an about 15% decrease of TAG and there are no error bars in the figure). But within this time frame, there is not much difference even in the *ARF1* deletion strain. We postulate a negative influence not a complete inhibition. So, we are unlikely to see something meaningful.

We do find Reviewer's 2 idea to tag Arf1 with a degron and monitor TAG mobilization appealing. However, with our experience on AID degrons, deletion of a protein takes from 30 min to 240 min, which would then again compromise the relevance of the results. Moreover, this approach has the same caveat as using the mutants, if there is just inhibition by Arf1, we are unlikely to see much of an effect. In addition, we respectfully point out again that *arf1-11* is a hyperactive mutant allele. Finally, if the Arf1 degron would lead to a decrease of TAGs, this might also be due to the lack of the positive effect on Dga1. At any rate, we would not be able to distinguish the effects between decreased Dga1 activity or increased Tgls activity.

Nevertheless, we performed the TAG mobilization experiment as suggested by the reviewer in wild-type and *arf1-11* mutants. We observe a small, but consistent reduction in the TAG hydrolysis rate in *arf1-11* cells after 30 min shift to the restrictive temperature 37°C, similar to what had been reported by Athenstaedt and Daum. As indicated by the growth curve, we cannot go much beyond 60 min. The data represent two independent experiments. The 3rd experiment is in preparation. We can include these data into the manuscript.

4. More work is necessary to show that Arf1 directly regulates TAG synthesis on lipid droplets. Only one of the two major TAG-synthesizing enzymes in yeast, Dga1, is on lipid droplets (and in some growth conditions it is largely on the ER). Lro1 has been shown to be in the ER and contains a transmembrane domain that almost certainly excludes it from the surface of lipid droplets. The different localization of the two primary TAG-synthesizing enzymes should be used to test the hypothesis that Arf1 regulates TAG synthesis on lipid droplets. If Arf1 regulates TAG synthesis in the ER, the model should be changed.

First, we are surprised about this new point, which is not related to the data that were added to the revised manuscript in response to previous reviewers' comments.

Second, we would like to point out to the reviewer that we use in the manuscript *Δlro1Δdga1* double mutants. Therefore, we did take a potential role of Lro1 into account. We assume that Arf1 might be able to regulate TAG synthesis at the ER and on LD droplets because Arf1-11 is localized on the ER and not on LDs at 23 °C and on LDs but not on the ER at 37°C. Yet, under both conditions we observe an increase of TAG in cells. Thus, Arf1-11 localization on the ER and on LDs increases TAG synthesis. We did not want to make a very strong point out of this and the reviewer had not pointed this out in the previous round, but if s/he wants, we are happy to change our model and include the TAG synthesis at the ER.

In principle, we can test the co-localization of Arf1 and Arf1-11 with Dga1 and Lro1. Lro1 marks the entire ER under normal growth conditions (Barbosa et al 2019). Thus, we will of course observe co-localization with Arf1-11 at 23°C, but what does that mean in terms of mechanism?

So, in the end this might be more relevant to look for Dga1 localization, but there again co-localization (between Arf1-11 and Dga1 on nascent LD at 37°C) does not mean interaction. Moreover, the reviewer was not satisfied by our co-IP data in which we pulled down Arf1 and Dga1 together and suggested that Arf1 acts on Dga1. We do not see how co-localization studies would be better evidence.

Still, to appease the reviewer, we endogenously tagged Dga1 and determined the co-localization with Arf1 and Arf1-11 at 23 °C and after shift to 37°C. Two examples for each temperature are shown below. As expected, Arf1 occasionally co-localizes or is juxtaposed to Dga1, but most of the signal is on the Golgi, consistent with the other data we presented in the manuscript. Again, as expected, Arf1-11 co-localizes with the ER pool of Dga1 at 23°C and the LD pool of Dga1 at 37°C.

We also did a pilot in which we grow Arf1 cells in oleate containing medium to induce LDs, under these conditions we observe an increase of co-localization of Arf1 and Dga1 on LDs, consistent with the data presented in the manuscript.

We can of course include these data into the manuscript and we are willing to do that. Yet, we did not learn anything from these experiments that we did not know before. So, it is just adding more data. Nevertheless, these data can be added, if the reviewer wishes to have them included.

We have tried to endogenously tag Lro1; however, without much success at this point. We keep trying and use different strategies. However, as outlined above, we do expect that ER-localized Lro1 will co-localize with Arf1-11 at 23°C as they are both in the ER.

5. The study suggests that Arf1 regulates or facilitates the transfer of fatty acids, acyl-CoA, or acetyl-CoA derived from TAG in lipid droplets to peroxisomes or mitochondria. BODIPY Red-C12 is used to directly measure the transport fatty acids and fatty acid metabolites in cells. However, without evidence that BODIPY Red-C12 can be incorporated into TAG, conjugated to CoA, or undergo beta-oxidation, it is hard to know how to interpret the results. This issue was raised by reviewer 3 and in response the authors wrote: “as far as we are aware we could not find any indication how Red-C12 is modified/metabolized on the way into mitochondria.” This needs to be resolved. Since the strongest evidence that Arf1 plays roles in metabolite trafficking come from the experiments with Red-C12, it is critical to confirm that it is metabolized similarly to endogenous fatty acids.

As we pointed out previously, numerous labs have been using in BODIPY RedC12 and other BODIPY FAs in the past. They have been shown to incorporate into LD-specific neutral lipids (for examples please see: Rambold et al., 2015; Herms et al., 2013, Kassan et al., 2013, Thumser and Storch, 2007, Wang et al., 2010). We did find now two publications with studies in mammalian cells, in which RedC12 or NBDC12 have been shown to be metabolized in mitochondria under starvation and which could be inhibited by etomoxir (inhibitor of the mitochondrial FA importer CPT-I) or DEUP (pan-lipase inhibitor), respectively (Rambold et al., 2015. Chang et al., 2019). Thus, these results show that RedC12 can be metabolized in mitochondria, mostly likely by β -oxidation in mammalian cells. We agree with the reviewer that it remains unknown to which precise product RedC12 is metabolized, but this goes beyond this manuscript. We will include a sentence about the findings by Rambold et al. and Chang et al. into the revised manuscript.

Reviewer #3 (Remarks to the Author):

This is an important paper for the field, with considerable conceptual novelty as outlined below. The authors have addressed most of the technical issues that were raised by this reviewer. However, a central conceptual point is not resolved.

A major finding of this work is that Arf1 is important in controlling mitochondrial fission and fusion. This is by itself an important result because it consolidates various and sometimes contradictory reports in the field. The major conceptual novelty is the cell biological description of the mitochondrial fission phenotype: the authors unequivocally show that localization of Arf1 to lipid

droplets leads to mitochondrial fission, whereas an Arf1 version that is permanently anchored to the ER causes the opposite phenotype elongating mitochondria. Additionally, when Arf1 is anchored on the LDs the cells produce significantly more triglycerides. Another important fact is that the role of Arf1 in controlling mitochondrial morphology is conserved between yeast and mammals.

Thank you for the positive assessment of our findings!

The different spatial organization of fatty acid beta oxidation in yeast and mammalian cells complicates the rest of the paper: it is not clear how the peroxisomes fit into the whole picture. The simplest explanation might be that Arf1 has a role in fatty acid transport at contact sites between various organelles as alluded to by the authors. How beta oxidation in mitochondria of mammalian cells or in peroxisomes of yeast relates to mitochondrial shape changes is not resolved in this work. An overwhelming amount of data is presented which is not easily accessible to the reader and perhaps even clouds the main message of the paper. The authors seem to have issues with explaining their thought process, and the experiments do not tie into a cohesive narrative.

We will modify the manuscript for clarity. We apologize for the enormous amounts of data that are in the manuscript. We absolutely agree with the reviewer, there are too many data in this manuscript. However, it was not our choice. We addressed all the points that were raised by this and the other reviewers; and reviewer 2 asks now for even more data.

A minimal requirement for acceptance at Nature Cell Biology is that the concept is formulated more clearly. In fact the experiments in FigS9 indicate that the peroxisomes are dispensable for the fatty acid transport to mitochondria. The authors state in line 361 “Abolishing peroxisome biogenesis did not affect the mitochondria phenotype [51], and did not reduce the flow of Red-C12 to mitochondria (Figure S9C) suggesting that under these conditions cells might develop alternative contact sites for efficient FA transfer.”

According to this reviewer these results indicate that Arf1 is involved in FA transfer between the ER and mitochondria, which is important for mitochondrial shape. Although Arf1 might indeed be involved in FA transfer from LDs to peroxisome in yeast, this activity might not be part of the mechanisms determining the mitochondrial phenotypes.

We will revise the manuscript for clarity about the concept, the proposed mechanisms and the model.

Loss of peroxisomes does not always lead to mitochondria phenotype. Some studies have shown that the loss of peroxisomes in mouse serotogenic neurons and *Drosophila* Malpighian tubules resulted in the enlargement of mitochondria (Rahim et al., 2016; Bülow et al., 2018). A recent study also reported that human patient-derived fibroblasts lacking Pex3 did not exhibit changes in mitochondrial morphology (Sugiura et al., 2017) while deletion of PEX3 or PEX5 lead to mitochondria fragmentation in mouse embryonic fibroblasts (Tanaka et al., JCS 2019).

We have no evidence that the ER-mitochondria contacts are the major place for FA exchange: if this would be the case, we should not observe a drop of in RedC12 transfer in LD-anchored Arf1 (**Fig. S9A**) and an increase in globular mitochondria (**Fig. 3C**).

Although it is widely known that ER-mitochondria contact sites are required to transport phospholipids, we could not find any report showing that neutral lipids (TAG in our case) can be transferred from the ER to the mitochondria. This would require specific DAG transporters on both the ER and mitochondria. If such mechanisms existed, we would see efficient TAG transfer in mitochondria of *arf1-11* strain grown at 37°C since in these conditions ER-mitochondria contacts lengths are significantly increased as compared to *ARF1* (**Fig. S9H**). Unfortunately, this is not the case, pointing to the possibility that the effect of ER-anchored Arf1-11 on mitochondria is most likely due to the absence of Arf1-11 on LD, as we actually show (**Fig. 3E-F** and **Fig. S9B**). All this again points to Arf1 being on LD.

Another point is that in the absence of efficient transfer of DAG into LD, they rather accumulate in the ER as many laboratories have reported (Discussed in Walther, Chung and Farese Ann Rev Cell Dev Biol 2017; Li et al., BMC Biol 2020; Choudhary et al., Curr. Biol 2019).

Please see also the response to point 1 of reviewer 2.

A minor point is that Figure 3D and E which is a central result in the paper should be better annotated in the figure legend. It is not clear how D and E relate and the scales in the two panels are not clear since the scale bar is not described in panel D.

This has now been changed.

Decision Letter, second revision:

Dear Professor Spang,

Thank you for your email asking us to reconsider our decision on your manuscript, "The small GTPase Arf1 regulates ATP synthesis and mitochondria homeostasis by modulating fatty acid metabolism". Thank you so much for your patience again while we were discussing your appeal materials.

I have now discussed your manuscript, the referees' comments, and your rebuttal, in detail with my colleagues, and we would be willing to reconsider a revised manuscript provided that you add the data discussed in the rebuttal, and that nothing similar is accepted for publication at Nature Cell Biology or published elsewhere in the meantime. We hope submission of the revision within 2 weeks would be possible.

Although our referees were very well placed to evaluate this work, if deemed necessary we may choose to involve an additional referee in the event of resubmission.

In addition, please pay close attention to our guidelines on statistical and methodological reporting (listed below) as failure to do so may delay the reconsideration of the revised manuscript. In particular please provide:

- a Supplementary Figure including unprocessed images of all gels/blots in the form of a multi-page pdf file. Please ensure that blots/gels are labeled and the sections presented in the figures are clearly indicated.
- a Supplementary Table including all numerical source data in Excel format, with data for different figures provided as different sheets within a single Excel file. The file should include source data giving rise to graphical representations and statistical descriptions in the paper and for all instances where the figures present representative experiments of multiple independent repeats, the source data of all repeats should be provided.

On resubmission please provide the completed Editorial Policy Checklist (found here <https://www.nature.com/documents/nr-editorial-policy-checklist.pdf>), and Reporting Summary (found here <https://www.nature.com/documents/nr-reporting-summary.pdf>). This is essential for reconsideration of the manuscript and these documents will be available to editors and referees in the event of peer review. For more information see below. Please also ensure that the presentation of statistical information in the revised submission complies with Nature Cell Biology's statistical guidelines (see below).

Please use the link below to submit the complete manuscript files and include a point-by-point response to the complete reviewer comments, verbatim as provided in their reports.

[Redacted]

Please let us know how you wish to proceed and when we can expect your revised manuscript. Thank

you again so much for your patience.

With kind regards,

Melina

Melina Casadio, PhD
Senior Editor, Nature Cell Biology
ORCID ID: <https://orcid.org/0000-0003-2389-2243>

GUIDELINES FOR EXPERIMENTAL AND STATISTICAL REPORTING

REPORTING REQUIREMENTS – To improve the quality of methods and statistics reporting in our papers we have recently revised the reporting checklist we introduced in 2013. We are now asking all life sciences authors to complete two items: an Editorial Policy Checklist (found here <https://www.nature.com/documents/nr-editorial-policy-checklist.pdf>) that verifies compliance with all required editorial policies and a reporting summary (found here <https://www.nature.com/documents/nr-reporting-summary.pdf>) that collects information on experimental design and reagents. These documents are available to referees to aid the evaluation of the manuscript. Please note that these forms are dynamic 'smart pdfs' and must therefore be downloaded and completed in Adobe Reader. We will then flatten them for ease of use by the reviewers. If you would like to reference the guidance text as you complete the template, please access these flattened versions at <http://www.nature.com/authors/policies/availability.html>.

Author Rebuttal, third revision:

Reviewers' comments:

Reviewer #1 (Remarks to the Author):

The authors now provide extensive new data and additional interpretations and discussion. I believe they have greatly improved the manuscript by clarifying their observations and providing some additional mechanistic insights. I am supportive of publication.

Thank you very much for the positive assessment of our work and for recommending our work for publication in Nature Cell Biology.

Reviewer #2 (Remarks to the Author):

My major concerns about this study remain the same. The study makes a good case that loss of Arf1 function affects mitochondrial morphology and dynamics, which has also been suggested previously. The revision has stronger evidence that Arf1 on mitochondria and lipid droplets, rather than the Arf1 on the Golgi or vesicles, is responsible for the effect of Arf1 on mitochondria. This is certainly interesting, but there is still not much mechanistic insight into how Arf1 affects mitochondria, and it is not clear Arf1 is directly involved. The study argues that Arf1 effects mitochondria by directly regulating TAG metabolism and beta-oxidation. This is an attractive idea, but the evidence remains unpersuasive.

We thank the reviewer that s/he finds our study interesting. We would like to point out, as we did in the previous rebuttal letter, that the Arf1-11 mutant is hyperactive mutant and not a loss of function. We do think that this is an important point because it is through this hyperactive mutant that we could detect a role of Arf1 on LD.

The reviewer's criticism hinges on the demonstration of the mechanism that Arf1 directly controls TAG metabolism and β -oxidation. The question is on how direct does this have to be? Direct in the strict

sense would be that we reconstitute all this in vitro and measure how Arf1 would regulate the individual enzyme activity. Otherwise, one could always claim that the effect of Arf1 on the Dga1, Faa1 and Tgl4 might be indirect. We actually never distinguished between direct and indirect in the manuscript. We only present our data on this issue. It is the reviewer's interpretation, which may or may not be correct. We assume that it is undisputed that we provide ample evidence that Arf1 is on LDs and affects proteins on both LDs and peroxisomes. We state now specifically in the manuscript that the interaction could be direct or indirect.

1. The idea TAG metabolism and beta-oxidation play important roles in regulating mitochondria shape is not well supported by results here or in the literature. Fig. S8BCF shows that cells expressing wild-type yArf1 have modest changes in mitochondria shape when they lack TAG or cannot perform beta-oxidation. Similar results have been found in previous studies using yeast and mammalian cells depleted on TAG or with defects in beta-oxidation. This suggests that defects in TAG metabolism and beta-oxidation probably play only a minor part in the dramatic changes in mitochondrial shape that occur in arf1-11 cells.

We apologize, but we disagree with this reviewer's statement. The changes in shape are significant, also in the presence of wild-type Arf1 situation (quantified in Fig. S8C). There is a statistically significant difference between the WT ARF1 strain grown at 37°C and the $\Delta lro1\Delta dga1$ mutant. We observed that 39% of cells showed globular mitochondria in the ARF1 $\Delta lro1\Delta dga1$ mutant compared to 21% in the WT ARF1. In addition, Fig. S8F clearly shows that loss of β -oxidation (in a $\Delta pot1$ or $\Delta pox1$ strain) leads to increased globular mitochondria and. To further emphasize this, we now provide the quantification of three independent experiments also in Fig. S8F.

In contrast, removing peroxisomes altogether in a $\Delta pex3 \Delta pex19$ mutant has almost no effect on mitochondrial shape. In agreement with these data, RedC12 was efficiently transferred into mitochondria (Fig. S9C) under conditions when peroxisomes are absent. However, removing either of the β -oxidation enzymes ($\Delta pot1$ or $\Delta pox1$) led to drastic changes in mitochondria phenotype (Fig. S8F and the quantification of the data provided above). The lack of effect on mitochondria morphology in the Δ peroxisome condition could also be due to the fact that the cells were grown in 2% glucose, which does not require peroxisome function and also no shuttling of FA/FA metabolites into mitochondria for function or cell survival.

Our data also suggest that at 37°C, FA β -oxidation is nevertheless somehow activated/needed in yeast (Fig. 6G-H). We cannot provide any explanation for this finding, but nothing is known in the literature either, except for one study in which the authors have shown that β -oxidation genes were upregulated after 3h of growth at 34°C (Musa et al., Aging 2018). What we do see on the other hand is an exacerbated phenotype on peroxisomal proteins, in FA storage and transfer to mitochondria in the *arf1-11* mutant at 37°C. This shows that a hyperactive mutant of Arf1 influences FA storage, peroxisomal protein levels, FA transfer to mitochondria and subsequently respiratory chain activity. Moreover, Arf1 is present on LDs, at contact sites involving LDs, peroxisomes and/or mitochondria, suggesting that Arf1 does have a function there.

The link between loss of functional peroxisomes and mitochondria morphology is well documented in the literature. To name a few, loss of mitochondrial function (decreased respiration and/or changes in mitochondrial morphology), is widely recognized as Zellweger syndrome disorders (ZWD) phenotypes (Shinde et al, 2018; Argyriou et al, 2019). Mutants of PEX19 (involved in peroxisome matrix protein import and peroxisome biogenesis) in a *Drosophila* model and PEX19-null patient fibroblasts showed swollen mitochondria (Bülow et al., MBoC 2017). Based on a mouse model for mild Zellweger syndrome in which PEX1 bears the G844D mutation, skin fibroblast cell lines showed a normal level of peroxisomes even though these cells had higher levels of VLCFAs, lower oxidation, and had an important defect in matrix protein import, pointing towards non-functional peroxisomes. In these cells, mitochondria morphology was also affected (Argyriou et al., Exp Eye Res 2018; Hiebler et al., Mol Genet Metab. 2014).

Unrelated to ZWD, it has also been shown that VPS13D can be recruited to peroxisomes and mitochondria by Miro1/2, and its absence leads to heterogenous loss of peroxisomes in cell population, which correlates with globular mitochondria (Baldwin et al., JCB 2021). In the same paper, they could not phenocopy the globular mitochondria in cells knocked-out for peroxisomes “we do not find evidence of the severely rounded mitochondria characteristic of VPS13D KO in the PEX-KO HeLa cells completely lacking peroxisomes”. Additionally, Seung et al. showed that mutations in VPS13D affects mitochondria morphology in *Drosophila* nervous system and in fibroblasts from families showing recessive ataxia (Seung et al., Ann Neurol 2018).

We do acknowledge that loss of TAG metabolism or β -oxidation cannot always be linked to loss of mitochondria morphology, and these different findings are highly discussed in the field. However, this has to be accounted to differences of the cell-types and/or organisms investigated. For instance, liver in PEX5^{-/-} mice used as ZWD model mitochondria showed strong morphological alterations and loss of Complex 1 activity, whereas these alterations were less severe in heart. Mitochondria also had normal appearance in fibroblasts in primary cultures of PEX5 knockout mice, or in situ in various organs;

complex I activity was not altered in fibroblast cultures of these mice (either immortalized fibroblasts or primary cultures) (Baumgart et al., American Journal of Pathology 2001). Similarly, loss of peroxisomes does not always lead to mitochondria phenotype. Some studies have shown that the loss of peroxisomes in mouse serotogenic neurons and *Drosophila* Malpighian tubules resulted in the enlargement of mitochondria (Rahim et al., 2016; Bülow et al., 2018). A recent study also reported that human patient-derived fibroblasts lacking Pex3 did not exhibit changes in mitochondrial morphology (Sugiura et al., 2017) while deletion of PEX3 or PEX5 lead to mitochondria fragmentation in mouse embryonic fibroblasts (Tanaka et al., JCS 2019). Another example is X-linked adrenoleukodystrophy, which is the most frequent peroxisomal disorder. It is caused by disruption of the peroxisomal ABC transporter ABCD1 (Bezman et al. 2001), which is involved in the import of VLCFAs into peroxisomes (van Roermund et al. 2011). Indeed, mitochondrial structural alterations in adrenal gland cells were reported in 12-month-old X-ALD mice (McGuinness et al. 2003). Although these findings were not corroborated for skeletal muscle mitochondria (Oezen et al. 2005), more recent studies suggested that mitochondria may be compromised in neuronal tissues of ABCD1 knockout mice and X-ALD patients due to elevated levels in VLCFAs (Hein et al. 2008; Galino et al. 2011; Lopez-Erauskin et al. 2013).

Taken together, whether or not loss of peroxisomes affects mitochondria morphology appears to be dependent on the tissue/cell type under investigation. The phenotype is probably also connected to the metabolic state of the cell and the environment in the tissue context. For our yeast experiments, we used 2% glucose in the medium, a condition under which FA do not need to be metabolized by yeast cells as there is glucose in high abundance. This might explain why the mitochondria appear normal when peroxisomes cannot be formed at all. However, when peroxisome biogenesis is only partially affected, we observe major mitochondria morphology defects, indicating that the presence of non-properly functioning peroxisomes is worse than their absence altogether and that mitochondria are affected by perturbed peroxisome biogenesis.

Importantly, our data also reveal that under conditions under which β -oxidation is required, mitochondria morphology is highly affected.

We offer now also the possibility in the manuscript that neutral lipids could also reach mitochondria via the ER, although this has not been observed to date. We suggest that the complete absence of peroxisomes might induce a salvage pathway, which would not be activated otherwise. This hypothesis would reconcile the different effects and phenotypes we observe for loss of peroxisomes versus abolished β -oxidation, a condition under which non-functional peroxisomes are present. In other words, we speculate that complete absence has less dramatic effects than a non-functional organelle.

2. The study still does not make a strong case that Arf1 plays a direct role in TAG metabolism. The revision now shows that Arf1 interacts with three TAG metabolism enzymes: Dga1, Faa1, and Tgl4. This

an important advance, which the study suggests shows that Arf1 directly regulates these enzymes. Unfortunately, there is no demonstration that it does. There are well established assays for all three enzymes that should be used to determine whether Arf1 affects their activities.

The experiment that Reviewer 2 is now asking is new, and is meant to answer a statement that we do not have in our manuscript as we never claimed that Arf1 directly regulates these enzymes. We do not, however, exclude this possibility. We do agree with the reviewer that it might be indirect through interaction with (an)other factor(s).

There is a published protocol from the group of Günther Daum from 2002 using microsomes and radioactively labelled lipid. However, in our case the experimental setup is non-trivial and we expect to face major problems. In order to provide the requested data, we would need to purify Arf1, Dga1, Faa1 and Tgl4, reconstitute them into LDs and then measure the corresponding activities. While Tgl4 has been purified in the past by Günther Daum's group, and we can purify myristoylated Arf1, setting up these assays with reconstitution on LDs is complicated and not something that can be just easily done. We are not aware of any reconstitution that we could just simply emulate.

We do acknowledge that we wrote in the discussion section that Arf1 interacts with Dga1 (lines 471-473), and with Tgl4 and Faa1 (lines 478-479), but we also state that we detect this interaction by co-immune precipitation. This kind of experiment does not allow the conclusion of direct interaction; the interaction could also be indirect. We made this clearer now in the discussion.

Finally, we would respectfully point out that this reviewer said in his/her previous review: 'How does Arf1 regulate levels of Pxa2 and Pox1? Does down regulation of these proteins, by itself, reduced TAG mobilization? If Arf1 regulates TAG mobilization or fatty acid transport in other ways, how does it do so? **A complete answer to these questions beyond the scope of this study**, but some hint of mechanism is necessary to support the claim Arf1 is directly involved.'

- We exclude transcriptional regulation of Pxa2 and Pox1 levels by Arf1, pointing to post-transcriptional regulation.
- It has been shown previously that lack of Pxa2 and Pox1 reduces TAG mobilization, and we refer to this study in the manuscript. Moreover, we have performed TAG mobilization assays with Arf1 and *arf1-11* mutant cells after shift to 37°C. *arf1-11* mutant cells show TAG mobilization defect. These data are included into the manuscript.

- We find that in addition to the post-transcriptional regulation of Pxa2 and Pox1 levels, Arf1 interacts, as shown by co-IP, with Dga1, Faa1 and Tgl4, indicating that there might be an additional way by which Arf1 regulates TAG levels.

We provide answers to all three questions raised by the reviewer and suggest a possible mechanism. We would like to emphasize again that we do not state direct interaction because we have not shown that. We agree with the reviewer, however, that this is a possibility.

3. If Arf1 plays a direct role in TAG mobilization from lipid droplets, then it should be possible to show this by directly measuring rates of TAG hydrolysis in cells. The classic way to do this quantitatively, as I mentioned in my previous review, is to block fatty acid synthesis with drugs like cerulenin, which result in the rapid hydrolysis of TAG. If Arf1 directly regulates Tgl4 (and perhaps other TAG lipases), the rate of TAG hydrolysis should change in cells depleted of Arf1 or following expression of Arf1-11.

In response to these points, we showed that peroxisomal proteins were already downregulated in *arf1-11* (Fig. 6F-H) and it had already been shown in the literature that the absence of these proteins impairs TAG flux (Ferreira, R., et al., Metabolic engineering of *Saccharomyces cerevisiae* for overproduction of triacylglycerols. *Metab Eng Commun*, 2018. 6: p. 22-27). Reviewer 2 also acknowledged that a complete understanding of Arf1 role on TAG mobilization and its effect on peroxisomes and mitochondria was beyond the scope of this manuscript. So, there was nothing we could have learnt from these experiments. Finally, we also provided new data to test the hypothesis that Arf1 regulates TAG levels in LD by anchoring Arf1, Arf1-11 and Arf1-CA on LD as suggested by Reviewer 2 (an appreciated suggestion!). This led to an increase of cells bearing globular mitochondria irrespective of the construct tested (Fig. 3C).

We were reluctant to perform the TAG mobilization experiment because Athenstaedt and Daum had performed TAG mobilization experiments in a $\Delta tgl4$ strain paper (Athenstaedt and Daum, *JBC* 2005). In Fig. 4 they report that deletion of TGL4 led only to an about 15% decrease of TAG within the first hour of cerulenin treatment and there are no error bars in the figure. We were wary that since in *arf1-11* cells, Tgl4 is not absent and because we can only shift *arf1-11* cells for maximal 1 hr to the restrictive temperature before the complication of dying cells kicks in. Since Reviewer 2 insisted on this experiment, we performed it. We observe a small effect on TAG mobilization in *arf1-11* cells. These data are now included into the manuscript.

We do find Reviewer's 2 idea to tag Arf1 with a degron and monitor TAG mobilization appealing. However, with our experience on AID degrons, deletion of a protein takes from 30 min to 240 min, which would then again compromise the relevance of the results. Moreover, this approach has the same caveat as using the mutants, if there is just inhibition by Arf1, we are unlikely to see much of an effect. Finally, if the Arf1 degron would lead to a decrease of TAGs, this might also be due to the lack of the positive effect on Dga1. At any rate, we would not be able to distinguish the effects between decreased Dga1 activity or increased Tgls activity.

We hope the reviewer is satisfied with the data, we provide on *arf1-11*.

4. More work is necessary to show that Arf1 directly regulates TAG synthesis on lipid droplets. Only one of the two major TAG-synthesizing enzymes in yeast, Dga1, is on lipid droplets (and in some growth conditions it is largely on the ER). Lro1 has been shown to be in the ER and contains a transmembrane domain that almost certainly excludes it from the surface of lipid droplets. The different localization of the two primary TAG-synthesizing enzymes should be used to test the hypothesis that Arf1 regulates TAG synthesis on lipid droplets. If Arf1 regulates TAG synthesis in the ER, the model should be changed.

First, we are surprised about this new point the reviewer raised, which was absent from the first round of reviews and is not related to the data that were added to the revised manuscript in response to previous reviewers' comments.

Second, we would like to point out to the reviewer that we use in the manuscript $\Delta lro1\Delta dga1$ double mutants. Therefore, we did take a potential role of Lro1 into account. We assume that Arf1 might be able to regulate TAG synthesis at the ER and on LD droplets because Arf1-11 is localized on the ER and not on LDs at 23 °C and on LDs but not on the ER at 37°C. Yet, under both conditions, we observed an increase of TAGs in cells. Thus, Arf1-11 localization on the ER and on LDs increases TAG synthesis. We did not want to make a very strong point out of this and the reviewer had not pointed this out in the previous round. We added a sentence about the possibility that Arf1 might also regulate TAG synthesis in the ER in the discussion.

We did not see how the localization of the TAG synthases should conclusively tell us whether Arf1 regulates TAG synthesis. However, we performed the experiments anyhow. Lro1 localizes to the ER (Barbosa et al 2019). This localization was not changed in *arf1-11* at the permissive or restrictive

temperature. As expected, at 23°C Arf1-11 co-localized with Lro1 and at 37°C it did not. Similarly, Arf1-11 co-localized with the ER-pool of Dga1 at 23°C and with the LD pool at 37°C. In all cases, Arf1 was mostly juxtaposed to the ER or LDs. Occasionally, we observed co-localization of Arf1 with Dga1 on LDs. We find that these data are not instructive whether or not Arf1 can stimulate TAG synthesis in the ER. Since the reviewer requested them, we included them into the manuscript.

5. The study suggests that Arf1 regulates or facilitates the transfer of fatty acids, acyl-CoA, or acetyl-CoA derived from TAG in lipid droplets to peroxisomes or mitochondria. BODIPY Red-C12 is used to directly measure the transport fatty acids and fatty acid metabolites in cells. However, without evidence that BODIPY Red-C12 can be incorporated into TAG, conjugated to CoA, or undergo beta-oxidation, it is hard to know how to interpret the results. This issue was raised by reviewer 3 and in response the authors wrote: “as far as we are aware we could not find any indication how Red-C12 is modified/metabolized on the way into mitochondria.” This needs to be resolved. Since the strongest evidence that Arf1 plays roles in metabolite trafficking come from the experiments with Red-C12, it is critical to confirm that it is metabolized similarly to endogenous fatty acids.

As we pointed out previously, numerous labs have been using in BODIPY RedC12 and other BODIPY FAs in the past. They have been shown to incorporate into LD-specific neutral lipids (for examples please see: Rambold et al., 2015; Herms et al., 2013, Kassar et al., 2013, Thumser and Storch, 2007, Wang et al., 2010). Thank you for making us digging deeper into the literature. We did find now two publications with studies in mammalian cells, in which RedC12 or NBDC12 have been shown to be metabolized in mitochondria under starvation and which could be inhibited by etomoxir (inhibitor of the mitochondrial FA importer CPT-I) or DEUP (pan-lipase inhibitor), respectively (Rambold et al., 2015. Chang et al., 2019). Thus, these results show that RedC12 can be metabolized in mitochondria, mostly likely by β -oxidation in mammalian cells. We agree with the reviewer that it remains unknown to which precise product RedC12 is metabolized, but this goes beyond this manuscript. We included the findings by Rambold et al., 2015 and Chang et al., 2019 into the revised manuscript.

Reviewer #3 (Remarks to the Author):

This is an important paper for the field, with considerable conceptual novelty as outlined below. The authors have addressed most of the technical issues that were raised by this reviewer. However, a central conceptual point is not resolved.

A major finding of this work is that Arf1 is important in controlling mitochondrial fission and fusion. This is by itself an important result because it consolidates various and sometimes contradictory reports in the field. The major conceptual novelty is the cell biological description of the mitochondrial fission phenotype: the authors unequivocally show that localization of Arf1 to lipid droplets leads to mitochondrial fission, whereas an Arf1 version that is permanently anchored to the ER causes the opposite phenotype elongating mitochondria. Additionally, when Arf1 is anchored on the LDs the cells produce significantly more triglycerides. Another important fact is that the role of Arf1 in controlling mitochondrial morphology is conserved between yeast and mammals.

Thank you for the positive assessment of our findings!

The different spatial organization of fatty acid beta oxidation in yeast and mammalian cells complicates the rest of the paper: it is not clear how the peroxisomes fit into the whole picture. The simplest explanation might be that Arf1 has a role in fatty acid transport at contact sites between various organelles as alluded to by the authors. How beta oxidation in mitochondria of mammalian cells or in peroxisomes of yeast relates to mitochondrial shape changes is not resolved in this work. An overwhelming amount of data is presented which is not easily accessible to the reader and perhaps even clouds the main message of the paper. The authors seem to have issues with explaining their thought process, and the experiments do not tie into a cohesive narrative.

We apologize if our writing of the manuscript was not clear enough. We modified the manuscript, streamlined the narrative and removed some superfluous information.

We equally apologize for the enormous amounts of data that are in the manuscript. We absolutely agree with the reviewer, there are too many data in this manuscript. However, it was not our choice. We addressed all the points that were raised by this and the other reviewers; and reviewer 2 asked in this round for even more data.

We agree with the reviewer that Arf1 could 'just' regulate fatty acid transport between various organelles. We are not aware of any study that has shown this before or that Arf1 even is present at these contact sites. We are a bit puzzled by this reviewer's comment that we do not resolve how β -oxidation relates to mitochondrial morphology. There is a lot of conflicting literature on this issue. In yeast, however, we show that blocking β -oxidation leads to mitochondrial fragmentation. We agree with

the reviewer that we did not show this for mammalian cells. However, in mammalian cells this is much more complicated because β -oxidation can take place in both peroxisomes and mitochondria. Nevertheless, we show that FA import factors and the rate-limiting protein in β -oxidation down-regulated in *yarf1-11* cells. Moreover, we provide strong evidence that FA flow into mitochondria is compromised in *arf1-11* cells in yeast and mammalian cells.

A minimal requirement for acceptance at Nature Cell Biology is that the concept is formulated more clearly. In fact the experiments in FigS9 indicate that the peroxisomes are dispensable for the fatty acid transport to mitochondria. The authors state in line 361 “Abolishing peroxisome biogenesis did not affect the mitochondria phenotype [51], and did not reduce the flow of Red-C12 to mitochondria (Figure S9C) suggesting that under these conditions cells might develop alternative contact sites for efficient FA transfer.”

According to this reviewer these results indicate that Arf1 is involved in FA transfer between the ER and mitochondria, which is important for mitochondrial shape. Although Arf1 might indeed be involved in FA transfer from LDs to peroxisome in yeast, this activity might not be part of the mechanisms determining the mitochondrial phenotypes.

As pointed out above, we revised the manuscript for clarity about the concept and the proposed mechanisms. We discuss now also an alternative pathway from LDs to ER to mitochondria to explain the Δ peroxisome phenotype.

Loss of peroxisomes does not always lead to mitochondria phenotype. Some studies have shown that the loss of peroxisomes in mouse serotonergic neurons and *Drosophila* Malpighian tubules resulted in the enlargement of mitochondria (Rahim et al., 2016; Bülow et al., 2018). A recent study also reported that human patient-derived fibroblasts lacking Pex3 did not exhibit changes in mitochondrial morphology (Sugiura et al., 2017) while deletion of PEX3 or PEX5 lead to mitochondria fragmentation in mouse embryonic fibroblasts (Tanaka et al., JCS 2019). We have no evidence that the ER-mitochondria contacts are the major place for FA exchange: if this would be the case, we should not observe a drop of in RedC12 transfer in LD-anchored Arf1 (Fig. S9A) and an increase in globular mitochondria (Fig. 3C). Still, we discuss now this possibility in the manuscript.

Although it is widely known that ER-mitochondria contact sites are required to transport phospholipids, we could not find any report showing that neutral lipids (TAG in our case) can be transferred from the ER to the mitochondria. This would require specific DAG transporters on both the ER and mitochondria.

If such mechanisms existed, we should have observed TAG transfer into mitochondria in *arf1-11* strain at 37°C, since under these conditions ER-mitochondria contact length is significantly increased as compared to ARF1. However, this is not the case (Fig. S9H). It is conceivable that since we only shift our yeast cells for 30 to 60 min to 37°C, the machinery that would allow transfer of neutral lipids from the ER to mitochondria is not in place yet, and would require prolonged stress. It will be interesting to explore this direction in the future.

Another point is that in the absence of efficient transfer of DAG into LD, DAG rather accumulate in the ER as many laboratories have reported (Discussed in Walther, Chung and Farese Ann Rev Cell Dev Biol 2017; Li et al., BMC Biol 2020; Choudhary et al., Curr. Biol 2019). These findings, however, do not exclude a role for Arf1 in TAG synthesis in the ER, and that a fraction of TAG might be transferred to mitochondria.

Please see also the response to point 1 of reviewer 2.

A minor point is that Figure 3D and E which is a central result in the paper should be better annotated in the figure legend. It is not clear how D and E relate and the scales in the two panels are not clear since the scale bar is not described in panel D.

This has now been changed.

Decision Letter, third revision:

Our ref: NCB-A47456C

1st May 2023

Dear Dr. Spang,

Thank you for submitting your revised manuscript "The small GTPase Arf1 regulates ATP synthesis and mitochondria homeostasis by modulating fatty acid metabolism" (NCB-A47456C) and thank you for your patience while we considered the revision. It has now been seen by two of the original referees -- Reviewer #2 was not convinced by the revision and did not provide detailed comments to us; however, we discussed all of Rev#2's points with Rev#3 in detail, and this referee also provided comments below. Rev#3 finds that the paper has improved in revision, and, after detailed editorial

discussions of the overall reviewer feedback, we agree that the current phenotypic data linking Arf1 to TAG levels are strong. We appreciate the edits made to the text to clarify what is shown and what is still unclear - in our view, this could be complemented by additional edits, e.g., at the end of page 10 when drawing conclusions on Arf1's function in the regulation of TAG synthesis and metabolism, FA flux into mitochondria, peroxisome function to clarify that this function may be direct or indirect.

Overall, at this stage, we agree with Rev#3 that the study should be published without further mechanistic definition, and we'll be happy in principle to publish it in Nature Cell Biology, pending minor revisions to comply with our editorial and formatting guidelines and minor edits to the text as per the above.

We are now performing detailed checks on your paper and will send you a checklist detailing our editorial and formatting requirements in about 1-2 weeks. Please do not upload the final materials and make any revisions until you receive this additional information from us. Thank you in advance for your patience.

Thank you again for your interest in Nature Cell Biology. Please do not hesitate to contact me if you have any questions.

Sincerely,

Melina

Melina Casadio, PhD
Senior Editor, Nature Cell Biology
ORCID ID: <https://orcid.org/0000-0003-2389-2243>

Reviewer #3 (Remarks to the Author):

After the second round of revisions, the paper has significantly improved in clarity, and I support publication.

The main discoveries come out clearly: Arf1 regulates fusion and fission of mitochondria, which is already reported but not well understood. Arf1-11 moves onto LDs at restrictive temperature and this leads to an increase in TAGs and induces mitochondrial fission. An Arf1-11 version that is anchored onto the ER prevents the fission. These data are strong and convincing.

The authors also make a good attempt to explain the observation mechanistically and conclude that beta oxidation in peroxisomes in yeast or in mitochondria in mammalian cells is perturbed when Arf1-11 is locked on the LDs which probably leads to fission. The mechanistic explanation for a direct function in increasing TAGs is perhaps less strong than the other parts but the data that is presented is still interesting. In my opinion, the paper will stimulate a lot of discussion in the field. While the exact mechanisms are not worked out yet the work deserves publication in Nature Cell Biology because of the important conceptual advance highlighting non-classical functions of Arf1 in cellular organization.

Decision Letter, final checks:

Our ref: NCB-A47456C

8th May 2023

Dear Dr. Spang,

Thank you for your patience as we've prepared the guidelines for final submission of your Nature Cell Biology manuscript, "The small GTPase Arf1 regulates ATP synthesis and mitochondria homeostasis by modulating fatty acid metabolism" (NCB-A47456C). Please carefully follow the step-by-step instructions provided in the attached file, and add a response in each row of the table to indicate the changes that you have made. Please also check and comment on any additional marked-up edits we have proposed within the text. Ensuring that each point is addressed will help to ensure that your revised manuscript can be swiftly handed over to our production team.

In recognition of the time and expertise our reviewers provide to Nature Cell Biology's editorial process, we would like to formally acknowledge their contribution to the external peer review of your manuscript entitled "The small GTPase Arf1 regulates ATP synthesis and mitochondria homeostasis by modulating fatty acid metabolism". For those reviewers who give their assent, we will be publishing their names alongside the published article.

Nature Cell Biology offers a Transparent Peer Review option for new original research manuscripts submitted after December 1st, 2019. As part of this initiative, we encourage our authors to support increased transparency into the peer review process by agreeing to have the reviewer comments, author rebuttal letters, and editorial decision letters published as a Supplementary item. When you submit your final files please clearly state in your cover letter whether or not you would like to participate in this initiative. Please note that failure to state your preference will result in delays in accepting your manuscript for publication.

Cover suggestions

As you prepare your final files we encourage you to consider whether you have any images or illustrations that may be appropriate for use on the cover of Nature Cell Biology.

Covers should be both aesthetically appealing and scientifically relevant, and should be supplied at the

best quality available. Due to the prominence of these images, we do not generally select images featuring faces, children, text, graphs, schematic drawings, or collages on our covers.

Nature Cell Biology has now transitioned to a unified Rights Collection system which will allow our Author Services team to quickly and easily collect the rights and permissions required to publish your work. Approximately 10 days after your paper is formally accepted, you will receive an email in providing you with a link to complete the grant of rights. If your paper is eligible for Open Access, our Author Services team will also be in touch regarding any additional information that may be required to arrange payment for your article.

Please note that *Nature Cell Biology* is a Transformative Journal (TJ). Authors may publish their research with us through the traditional subscription access route or make their paper immediately open access through payment of an article-processing charge (APC). Authors will not be required to make a final decision about access to their article until it has been accepted. Find out more about Transformative Journals

Please use the following link for uploading these materials:
[Redacted]

Best regards,

Kendra Donahue
Staff
Nature Cell Biology

On behalf of

Melina Casadio, PhD
Senior Editor, Nature Cell Biology
ORCID ID: <https://orcid.org/0000-0003-2389-2243>

Reviewer #2:
None

Reviewer #3:
Remarks to the Author:

After the second round of revisions, the paper has significantly improved in clarity, and I support publication.

The main discoveries come out clearly: Arf1 regulates fusion and fission of mitochondria, which is already reported but not well understood. Arf1-11 moves onto LDs at restrictive temperature and this leads to an increase in TAGs and induces mitochondrial fission. An Arf1-11 version that is anchored onto the ER prevents the fission. These data are strong and convincing.

The authors also make a good attempt to explain the observation mechanistically and conclude that beta oxidation in peroxisomes in yeast or in mitochondria in mammalian cells is perturbed when Arf1-11 is locked on the LDs which probably leads to fission. The mechanistic explanation for a direct function in increasing TAGs is perhaps less strong than the other parts but the data that is presented is still interesting. In my opinion, the paper will stimulate a lot of discussion in the field. While the exact mechanisms are not worked out yet the work deserves publication in Nature Cell Biology because of the important conceptual advance highlighting non-classical functions of Arf1 in cellular organization.

Author Rebuttal, fourth revision:

We thank the reviewers for assessing the manuscript again.

Reviewer #2:

None

Reviewer #3:

Remarks to the Author:

After the second round of revisions, the paper has significantly improved in clarity, and I support publication.

Thank you very much for your positive assessment.

The main discoveries come out clearly: Arf1 regulates fusion and fission of mitochondria, which is already reported but not well understood. Arf1-11 moves onto LDs at restrictive temperature and this leads to an increase in TAGs and induces mitochondrial fission. An Arf1-11 version that is anchored onto the ER prevents the fission. These data are strong and convincing.

Thank you!

The authors also make a good attempt to explain the observation mechanistically and conclude that beta oxidation in peroxisomes in yeast or in mitochondria in mammalian cells is perturbed when Arf1-11 is locked on the LDs which probably leads to fission. The mechanistic explanation for a direct function in increasing TAGs is perhaps less strong than the other parts but the data that is presented is still interesting. In my opinion, the paper will stimulate a lot of discussion in the field. While the exact mechanisms are not worked out yet the work deserves publication in Nature Cell Biology because of the important conceptual advance highlighting non-classical functions of Arf1 in cellular organization.

Thank you!

In addition, we changed the manuscript as requested by the senior editor Melina Casadio. We emphasized more that we still do not know whether Arf1 acts directly or indirectly in regulating the flow of FA metabolites from LDs to mitochondria.

Finally, we amended the manuscript to comply with Nature Cell Biology's journal style as requested. For details, please see the authors checklist.

Final Decision Letter:

Dear Dr Spang,

I am pleased to inform you that your manuscript, "Arf1 coordinates fatty acid metabolism and mitochondrial homeostasis", has now been accepted for publication in Nature Cell Biology.

Please note that *Nature Cell Biology* is a Transformative Journal (TJ). Authors may publish their research with us through the traditional subscription access route or make their paper immediately open access through payment of an article-processing charge (APC). Authors will not be required to make a final decision about access to their article until it has been accepted. Find out more about Transformative Journals

Authors may need to take specific actions to achieve compliance with funder and institutional open access mandates. If your research is supported by a funder that requires

immediate open access (e.g. according to Plan S principles) then you should select the gold OA route, and we will direct you to the compliant route where possible. For authors selecting the subscription publication route, the journal's standard licensing terms will need to be accepted, including self-archiving policies. Those licensing terms will supersede any other terms that the author or any third party may assert apply to any version of the manuscript.

If you have not already done so, we strongly recommend that you upload the step-by-step protocols used in this manuscript to the Protocol Exchange (www.nature.com/protocolexchange), an open online resource established by Nature Protocols that allows researchers to share their detailed experimental know-how. All uploaded protocols are made freely available, assigned DOIs for ease of citation and are fully searchable through nature.com. Protocols and Nature Portfolio journal papers in which they are used can be linked to one another, and this link is clearly and prominently visible in the online versions of both papers. Authors who performed the specific experiments can act as primary authors for the Protocol as they will be best placed to share the methodology details, but the Corresponding Author of the present research paper should be included as one of the authors. By uploading your Protocols to Protocol Exchange, you are enabling researchers to more readily reproduce or adapt the methodology you use, as well as increasing the visibility of your protocols and papers. You can also establish a dedicated page to collect your lab Protocols. Further information can be found at www.nature.com/protocolexchange/about

With kind regards,

Melina Casadio, PhD
Senior Editor, Nature Cell Biology
ORCID ID: <https://orcid.org/0000-0003-2389-2243>
